# Practical Sharpness-Aware Minimization Cannot Converge All the Way to Optima

**Dongkuk Si**     **Chulhee Yun**

Korea Advanced Institute of Science and Technology (KAIST)

{dongkuksi, chulhee.yun}@kaist.ac.kr

## Abstract

Sharpness-Aware Minimization (SAM) is an optimizer that takes a descent step based on the gradient at a perturbation $\boldsymbol{y}_t = \boldsymbol{x}_t + \rho \frac{\nabla f(\boldsymbol{x}_t)}{\|\nabla f(\boldsymbol{x}_t)\|}$ of the current point $\boldsymbol{x}_t$. Existing studies prove convergence of SAM for smooth functions, but they do so by assuming decaying perturbation size $\rho$ and/or no gradient normalization in $\boldsymbol{y}_t$, which is detached from practice. To address this gap, we study deterministic/stochastic versions of SAM with practical configurations (i.e., constant $\rho$ and gradient normalization in $\boldsymbol{y}_t$) and explore their convergence properties on smooth functions with (non)convexity assumptions. Perhaps surprisingly, in many scenarios, we find out that SAM has *limited* capability to converge to global minima or stationary points. For smooth strongly convex functions, we show that while deterministic SAM enjoys tight global convergence rates of $\tilde{\Theta}(\frac{1}{T^2})$, the convergence bound of stochastic SAM suffers an *inevitable* additive term $\mathcal{O}(\rho^2)$, indicating convergence only up to *neighborhoods* of optima. In fact, such $\mathcal{O}(\rho^2)$ factors arise for stochastic SAM in all the settings we consider, and also for deterministic SAM in nonconvex cases; importantly, we prove by examples that such terms are *unavoidable*. Our results highlight vastly different characteristics of SAM with vs. without decaying perturbation size or gradient normalization, and suggest that the intuitions gained from one version may not apply to the other.

## 1   Introduction

Modern neural networks are armed with a large number of layers and parameters, having a risk to overfit to training data. In order to make accurate predictions on unseen data, generalization performance has been considered as the most important factor in deep learning models. Based on the widely accepted belief that geometric properties of loss landscape are correlated with generalization performance, studies have proved theoretical and empirical results regarding the relation between *sharpness* measures and generalization [14, 17, 19, 21, 29]. Here, *sharpness* of a loss at a point generally refers to the degree to which the loss varies in the small neighborhood of the point.

Motivated by prior studies that show flat minima have better generalization performance [19, 21], Foret et al. [16] propose an optimization method referred to as *Sharpness-Aware Minimization (SAM)*. A single iteration of SAM consists of one ascent step (perturbation) and one descent step. Starting from the current iterate $\boldsymbol{x}_t$, SAM first takes an ascent step $\boldsymbol{y}_t = \boldsymbol{x}_t + \rho \frac{\nabla f(\boldsymbol{x}_t)}{\|\nabla f(\boldsymbol{x}_t)\|}$ to (approximately) maximize the loss value in the $\rho$-neighborhood of $\boldsymbol{x}_t$, and then uses the gradient at $\boldsymbol{y}_t$ to update $\boldsymbol{x}_t$ to $\boldsymbol{x}_{t+1}$. This two-step procedure gives SAM a special characteristic: the tendency to search for a flat minimum, i.e., a minimum whose neighboring points also return low loss value. Empirical results [3, 9, 16, 20, 25] show that SAM demonstrates an exceptional ability to perform well on different models and tasks with high generalization performance. Following the success of SAM,

Table 1: Convergence of SAM with constant perturbation size $\rho$ after $T$ steps. **C1**, **C2**, **C3**, and **C4** indicate $\beta$-smoothness, $\mu$-strong convexity, convexity, and Lipschitzness, respectively. **C5** indicates the bounded variance of the gradient oracle. See Section 2.3 for definitions of **C1–C5**.

| Optimizer | Function Class | Convergence Upper/Lower Bounds | Reference |
|---|---|---|---|
| Deterministic SAM | **C1,C2** | $\min_{t \in \{0,\ldots,T\}} f(\boldsymbol{x}_t) - f^* = \tilde{\mathcal{O}}\left(\exp(-T) + \frac{1}{T^2}\right)$ | Theorem 3.1 |
| Deterministic SAM | **C1,C2** | $\min_{t \in \{0,\ldots,T\}} f(\boldsymbol{x}_t) - f^* = \Omega\left(\frac{1}{T^2}\right)$ | Theorem 3.2 |
| Deterministic SAM | **C1,C3** | $\frac{1}{T}\sum_{t=0}^{T-1}\|\nabla f(\boldsymbol{x}_t)\|^2 = \mathcal{O}\left(\frac{1}{T} + \frac{1}{\sqrt{T}}\right)$ | Theorem 3.3 |
| Deterministic SAM | **C1** | $\frac{1}{T}\sum_{t=0}^{T-1}\|\nabla f(\boldsymbol{x}_t)\|^2 \leq \mathcal{O}\left(\frac{1}{T}\right) + \beta^2\rho^2$ | Theorem 3.4 |
| Stochastic SAM | **C1,C2,C5** | $\mathbb{E}f(\boldsymbol{x}_T) - f^* \leq \tilde{\mathcal{O}}\left(\exp(-T) + \frac{[\sigma^2 - \beta^2\rho^2]_+}{T}\right) + \frac{2\beta^2\rho^2}{\mu}$ | Theorem 4.1 |
| Stochastic SAM | **C1,C3,C5** | $\frac{1}{T}\sum_{t=0}^{T-1}\mathbb{E}\|\nabla f(\boldsymbol{x}_t)\|^2 \leq \mathcal{O}\left(\frac{1}{T} + \frac{\sqrt{[\sigma^2 - \beta^2\rho^2]_+}}{\sqrt{T}}\right) + 4\beta^2\rho^2$ | Theorem 4.3 |
| Stochastic $n$-SAM | **C1,C5** | $\frac{1}{T}\sum_{t=0}^{T-1}\mathbb{E}\|\nabla f(\boldsymbol{x}_t)\|^2 \leq \mathcal{O}\left(\frac{1}{T} + \frac{1}{\sqrt{T}}\right) + \beta^2\rho^2$ | Theorem 4.5 |
| Stochastic $m$-SAM | **C1,C4,C5** | $\frac{1}{T}\sum_{t=0}^{T-1}\mathbb{E}[(\|\nabla f(\boldsymbol{x}_t)\| - \beta\rho)^2] \leq \mathcal{O}\left(\frac{1}{\sqrt{T}}\right) + 5\beta^2\rho^2$ | Theorem 4.6 |

numerous extensions of SAM have been proposed in the literature [6, 15, 18, 23, 24, 26–28, 30–32, 34, 35].

On the theoretical side, various studies have demonstrated different characteristics of SAM [1, 2, 4, 5, 12, 13, 22, 33]. However, comprehending the global convergence properties of practical SAM on a theoretical level still remains elusive. In fact, several recent results [2, 18, 27, 31, 35] attempt to prove the convergence guarantees for SAM and its variants. While these results provide convergence guarantees of SAM on smooth functions, they are somewhat detached to the practical implementations of SAM and its variants. They either (1) assume *decaying* or *sufficiently small* perturbation size $\rho$ [18, 27, 31, 35], whereas $\rho$ is set to constant in practice; or (2) assume ascent steps *without gradient normalization* [2], whereas practical implementations of SAM use normalization when calculating the ascent step $\boldsymbol{y}_t$.

## 1.1 Summary of Our Contributions

To address the aforementioned limitations of existing results, we investigate convergence properties of SAM using *gradient normalization* in ascent steps and arbitrary *constant perturbation size* $\rho$. We note that to the best of our knowledge, the convergence analysis of SAM have not been carried out under the two practical implementation choices. Our analyses mainly focus on smooth functions, with different levels of convexity assumptions ranging from strong convexity to nonconvexity. We summarize our contributions below; a summary of our convergence results (upper and lower bounds) can also be found in Table 1.

- For deterministic SAM, we prove convergence to global minima of smooth strongly convex functions, and show the tightness of convergence rate in terms of $T$. Furthermore, we establish the convergence of SAM to stationary points of smooth convex functions. For smooth nonconvex functions, we prove that SAM guarantees convergence to stationary points up to an additive factor $\mathcal{O}(\rho^2)$. We provide a worst-case example that always suffers a matching squared gradient norm $\Omega(\rho^2)$, showing that the additive factor is unavoidable and tight in terms of $\rho$.

- For stochastic settings, we analyze two versions of stochastic SAM ($n$-SAM and $m$-SAM) on smooth strongly convex, smooth convex, and smooth (Lipschitz) nonconvex functions. We provide convergence guarantees to global minima or stationary points up to additive factors $\mathcal{O}(\rho^2)$. In case of $m$-SAM, we demonstrate that these factors are inevitable and tight in terms of $\rho$, by providing worst-case examples where SAM fails to converge properly.

## 1.2 Related Works: Convergence Analysis of SAM

Recent results [18, 27, 31, 35] prove convergence of stochastic SAM and its variants to stationary points for smooth nonconvex functions. However, these convergence analyses are limited to only *smooth nonconvex* functions and do not address convex functions. Also, as already pointed out, all the proofs require one crucial assumption detached from practice: *decaying* (or *sufficiently small*) perturbation size $\rho$. If $\rho$ becomes sufficiently small, then the difference between $f^{\text{SAM}}(\boldsymbol{x}) = \max_{\|\boldsymbol{\epsilon}\| \leq \rho} f(\boldsymbol{x} + \boldsymbol{\epsilon})$ and $f(\boldsymbol{x})$ becomes negligible, which means that they undergo approximately the same updates. Due to this reason, especially in later iterates, SAM with negligible perturbation size

would become almost identical to gradient descent (GD), resulting in a significant difference in the convergence behavior compared to constant $\rho$.

As for existing proofs that do not require decaying $\rho$, Andriushchenko and Flammarion [2] prove convergence guarantees for *deterministic* SAM with *constant* $\rho$ on smooth nonconvex, smooth Polyak-Łojasiewicz, smooth convex, and smooth strongly convex functions. Moreover, the authors prove convergence guarantees of *stochastic* SAM, this time with *decaying* $\rho$, on smooth nonconvex and smooth Polyak-Łojasiewicz functions. However, their analyses are also crucially different from the practical implementations of SAM because their proofs are for a variant of SAM *without gradient normalization* in ascent steps, with updates in the form of $\boldsymbol{x}_{t+1} = \boldsymbol{x}_t - \eta\nabla f(\boldsymbol{x}_t + \rho\nabla f(\boldsymbol{x}_t))$. This form can be considered as vanilla SAM with an "effective" perturbation size $\rho\|\nabla f(\boldsymbol{x}_t)\|$, which indicates that even with constant $\rho$, the effective perturbation size will become smaller and smaller as the algorithm converges towards a stationary point. As in the case of decaying $\rho$ discussed above, this can make a huge difference in the convergence behavior.

Figure 1 illustrates a comparison between SAM with and without gradient normalization, highlighting the disparity between their convergence points. As we predicted above, it is evident that they indeed reach entirely distinct global minima, exhibiting different convergence characteristics.

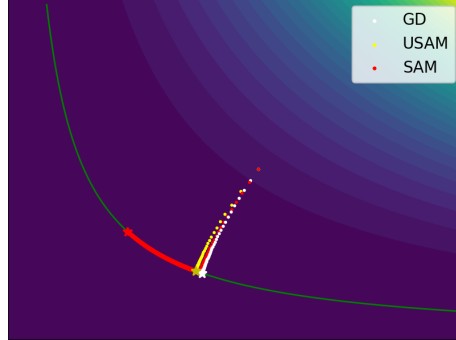

Figure 1: Trajectory plot for a function $f(x, y) = (xy - 1)^2$. The green line indicates global minima of $f$. SAM and USAM indicates SAM with and without gradient normalization, respectively, and GD indicates gradient descent. When USAM approaches the green line, it converges to a nearby global minimum, which is similar to GD. In contrast, SAM travels along the green line, towards the flattest minimum $(1, 1)$.

Discussions on the related works on the relationship between sharpness and generalization, as well as other theoretical properties of SAM, are provided in Appendix A.

### 1.3 Notation

The euclidean norm of a vector $\boldsymbol{v}$ is denoted as $\|\boldsymbol{v}\|$. We let $\mathbb{R}_+^* \triangleq \{x \in \mathbb{R} \mid x > 0\}$, and use $[\cdot]_+$ to define $\max\{\cdot, 0\}$. We use $\mathcal{O}(\cdot)$ and $\Omega(\cdot)$ to hide numerical constants in our upper and lower bounds, respectively. We use $\tilde{\mathcal{O}}(\cdot)$ to additionally hide logarithmic factors. When discussing rates, we sometimes use these symbols to hide everything except $T$, the number of SAM iterations. When discussing additive factors that arise in our convergence bounds, we use these symbols to hide all other variables except the perturbation size $\rho$. For an objective function $f$ and initialization $\boldsymbol{x}_0$ of SAM, we define $f^* \triangleq \inf_{\boldsymbol{x}} f(\boldsymbol{x})$ and $\Delta \triangleq f(\boldsymbol{x}_0) - f^*$.

## 2 Sharpness-Aware Minimization: Preliminaries and Intuitions

Before presenting the main results, we first introduce deterministic and stochastic SAM, and present definitions of function classes considered in this paper. We next introduce *virtual gradient map* and *virtual loss* that shed light on our intuitive understanding of SAM's convergence behavior.

### 2.1 Sharpness-Aware Minimization (SAM)

We focus on minimizing a function $f : \mathbb{R}^d \to \mathbb{R}$, where the optimization variable is represented by $\boldsymbol{x} \in \mathbb{R}^d$, using the Sharpness-Aware Minimization (SAM) optimizer. Instead of minimizing the vanilla objective function $f(\boldsymbol{x})$, SAM [16] aims to minimize the *SAM objective* $f^{\mathrm{SAM}}(\boldsymbol{x})$, where

$$\min_{\boldsymbol{x}} f^{\mathrm{SAM}}(\boldsymbol{x}) = \min_{\boldsymbol{x}} \max_{\|\boldsymbol{\epsilon}\| \le \rho} f(\boldsymbol{x} + \boldsymbol{\epsilon}). \tag{1}$$

For $\rho \in \mathbb{R}_+^*$, the SAM loss $f^{\mathrm{SAM}}(\boldsymbol{x})$ outputs the worst function value in a $\rho$-ball $\{\boldsymbol{w} \in \mathbb{R}^d \mid \|\boldsymbol{x} - \boldsymbol{w}\| \le \rho\}$ around $\boldsymbol{x}$. Assuming "sufficiently small" $\rho$, the inner maximization of (1) can be solved by Taylor-approximating the objective:

$$\arg \max_{\|\boldsymbol{\epsilon}\| \le \rho} f(\boldsymbol{x} + \boldsymbol{\epsilon}) \approx \arg \max_{\|\boldsymbol{\epsilon}\| \le \rho} f(\boldsymbol{x}) + \langle \boldsymbol{\epsilon}, \nabla f(\boldsymbol{x}) \rangle = \rho \frac{\nabla f(\boldsymbol{x})}{\|\nabla f(\boldsymbol{x})\|} \triangleq \hat{\boldsymbol{\epsilon}}(\boldsymbol{x}).$$

Using this approximate solution $\hat{\epsilon}(\boldsymbol{x})$, we can define the *approximate SAM objective* $\hat{f}^{\mathrm{SAM}}$ as $\hat{f}^{\mathrm{SAM}}(\boldsymbol{x}) \triangleq f(\boldsymbol{x} + \hat{\epsilon}(\boldsymbol{x}))$. In order to run gradient descent (GD) on $\hat{f}^{\mathrm{SAM}}(\boldsymbol{x})$, one needs to calculate its gradient; however, from the definition of $\hat{\epsilon}(\boldsymbol{x})$, we can realize that $\nabla\hat{f}^{\mathrm{SAM}}(\boldsymbol{x})$ has terms that involve the Hessian of $f$. Here, SAM makes another approximation, by ignoring the Hessian term:

$$\nabla\hat{f}^{\mathrm{SAM}}(\boldsymbol{x}) \approx \nabla f(\boldsymbol{x})|_{\boldsymbol{x}+\hat{\epsilon}(\boldsymbol{x})}.$$

From these approximations, one iteration of SAM is defined as a set of two-step update equations:

$$\begin{cases} \boldsymbol{y}_t = \boldsymbol{x}_t + \rho\frac{\nabla f(\boldsymbol{x}_t)}{\|\nabla f(\boldsymbol{x}_t)\|}, \\ \boldsymbol{x}_{t+1} = \boldsymbol{x}_t - \eta\nabla f(\boldsymbol{y}_t). \end{cases} \tag{2}$$

As seen in (2), we use $\boldsymbol{x}_t$ to denote the iterate of SAM at the $t$-th step. We use $T$ to denote the number of SAM iterations. We refer to the hyperparameter $\rho \in \mathbb{R}_+^*$ as the *perturbation size* and $\eta \in \mathbb{R}_+^*$ as the *step size*. Note that in (2), the perturbation size $\rho$ is a time-invariant constant; in practice, it is common to fix $\rho$ as a constant throughout training [3, 9, 16].

According to the SAM update in (2), the update cannot be defined when $\|\nabla f(\boldsymbol{x})\| = 0$. In practice, we add a small numerical constant (e.g., $10^{-12}$) to the denominator in order to prevent numerical instability. In this paper, we ignore this constant and treat $\frac{\nabla f(\boldsymbol{x}_t)}{\|\nabla f(\boldsymbol{x}_t)\|}$ as 0 whenever $\|\nabla f(\boldsymbol{x}_t)\| = 0$.

## 2.2 SAM under Stochastic Settings

To analyze stochastic SAM, we suppose that the objective is given as $f(\boldsymbol{x}) = \mathbb{E}_\xi[l(\boldsymbol{x};\xi)]$, where $\xi$ is a stochastic parameter (e.g., data sample) and $l(\boldsymbol{x};\xi)$ indicates the loss at point $\boldsymbol{x}$ with a random sample $\xi$. Based on the SAM update in (2), we can define stochastic SAM under this setting:

$$\begin{cases} \boldsymbol{y}_t = \boldsymbol{x}_t + \rho\frac{g(\boldsymbol{x}_t)}{\|g(\boldsymbol{x}_t)\|}, \\ \boldsymbol{x}_{t+1} = \boldsymbol{x}_t - \eta\tilde{g}(\boldsymbol{y}_t). \end{cases} \tag{3}$$

We define $g(\boldsymbol{x}) = \nabla_{\boldsymbol{x}}l(\boldsymbol{x};\xi)$ and $\tilde{g}(\boldsymbol{x}) = \nabla_{\boldsymbol{x}}l(\boldsymbol{x};\tilde{\xi})$. Here, $\xi$ and $\tilde{\xi}$ are stochastic parameters, queried from any distribution(s) which satisfies $\mathbb{E}_{\tilde{\xi}}l(\boldsymbol{x};\tilde{\xi}) = \mathbb{E}_\xi l(\boldsymbol{x};\xi) = f(\boldsymbol{x})$.

There are two popular variants of stochastic SAM, introduced in Andriushchenko and Flammarion [2]. Stochastic $n$-SAM algorithm refers to the update equation (3) when $\xi$ and $\tilde{\xi}$ are independent. In contrast, practical SAM algorithm in Foret et al. [16] employs stochastic $m$-SAM algorithm, which follows the update equation (3) where $\xi$ is equal to $\tilde{\xi}$. We will consider both versions in our theorems.

## 2.3 Function Classes

We state definitions and assumptions of function classes of interest, which are fairly standard.

**Definition 2.1** (Convexity). A function $f : \mathbb{R}^d \to \mathbb{R}$ is convex if, for any $\boldsymbol{x}, \boldsymbol{y} \in \mathbb{R}^d$ and $\lambda \in [0, 1]$, it satisfies $f(\lambda\boldsymbol{x} + (1 - \lambda)\boldsymbol{y}) \leq \lambda f(\boldsymbol{x}) + (1 - \lambda)f(\boldsymbol{y})$.

**Definition 2.2** ($\mu$-Strong Convexity). A differentiable function $f : \mathbb{R}^d \to \mathbb{R}$ is $\mu$-strongly convex if there exists $\mu > 0$ such that $f(\boldsymbol{x}) \geq f(\boldsymbol{y}) + \langle\nabla f(\boldsymbol{y}), \boldsymbol{x} - \boldsymbol{y}\rangle + \frac{\mu}{2}\|\boldsymbol{x} - \boldsymbol{y}\|^2$, for all $\boldsymbol{x}, \boldsymbol{y} \in \mathbb{R}^d$.

**Definition 2.3** ($L$-Lipschitz Continuity). A function $f : \mathbb{R}^d \to \mathbb{R}$ is $L$-Lipschitz continuous if there exists $L \geq 0$ such that $\|f(\boldsymbol{x}) - f(\boldsymbol{y})\| \leq L\|\boldsymbol{x} - \boldsymbol{y}\|$, for all $\boldsymbol{x}, \boldsymbol{y} \in \mathbb{R}^d$.

**Definition 2.4** ($\beta$-Smoothness). A differentiable function $f : \mathbb{R}^d \to \mathbb{R}$ is $\beta$-smooth if there exists $\beta \geq 0$ such that $\|\nabla f(\boldsymbol{x}) - \nabla f(\boldsymbol{y})\| \leq \beta\|\boldsymbol{x} - \boldsymbol{y}\|$, for all $\boldsymbol{x}, \boldsymbol{y} \in \mathbb{R}^d$.

Next we define an assumption considered in the analysis of stochastic SAM (3).

**Assumption 2.5** (Bounded Variance). The gradient oracle of a differentiable function $f : \mathbb{R}^d \to \mathbb{R}$ has bounded variance if there exists $\sigma \geq 0$ such that

$$\mathbb{E}_\xi\|\nabla f(\boldsymbol{x}) - \nabla l(\boldsymbol{x};\xi)\|^2 \leq \sigma^2, \quad \mathbb{E}_{\tilde{\xi}}\|\nabla f(\boldsymbol{x}) - \nabla l(\boldsymbol{x};\tilde{\xi})\|^2 \leq \sigma^2, \quad \forall\boldsymbol{x} \in \mathbb{R}^d.$$

## 2.4 SAM as GD on Virtual Loss

Convergence analysis of SAM is challenging due to the presence of normalized ascent steps. In order to provide intuitive explanations of SAM's convergence properties, we develop tools referred to as

*virtual gradient map* and *virtual loss* in this section. These tools can provide useful intuitions when we discuss our main results.

In order to define the new tools, we first need to define Clarke subdifferential [10], whose definition below is from Theorem 6.2.5 of Borwein and Lewis [7].

**Definition 2.6** (Clarke Subdifferential). Suppose that a function $f : \mathbb{R}^d \to \mathbb{R}$ is locally Lipschitz around a point $\boldsymbol{x}$ and differentiable on $\mathbb{R}^d \setminus W$ where $W$ is a set of measure zero. Then, the Clarke subdifferential of $f$ at $\boldsymbol{x}$ is $\partial f(\boldsymbol{x}) \triangleq \mathrm{cvxhull}\{\lim_{t \to \infty} \nabla f(\boldsymbol{x}_t) \mid \boldsymbol{x}_t \to \boldsymbol{x}, \boldsymbol{x}_t \notin W\}$, where $\mathrm{cvxhull}$ denotes the convex hull of a set.

Clarke subdifferential $\partial f(\boldsymbol{x})$ is a convex hull of all possible limits of $\nabla f(\boldsymbol{x}_t)$ as $\boldsymbol{x}_t$ approaches $\boldsymbol{x}$. It can be thought of as an extension of gradient to a nonsmooth function. It is also known that for a convex function $f$, the Clarke subdifferential of $f$ is equal to the subgradient of $f$.

Using the definition of Clarke differential, we now define virtual gradient map and virtual loss of $f$, which can be employed for understanding the convergence of SAM.

**Definition 2.7** (Virtual Gradient Map/Loss). For a differentiable function $f : \mathbb{R}^d \to \mathbb{R}$, we define the *virtual gradient map* $G_f : \mathbb{R}^d \to \mathbb{R}^d$ of $f$ to be $G_f(\boldsymbol{x}) \triangleq \nabla f\left(\boldsymbol{x} + \rho \frac{\nabla f(\boldsymbol{x})}{\|\nabla f(\boldsymbol{x})\|}\right)$. Additionally, if there exists a function $J_f : \mathbb{R}^d \to \mathbb{R}$ whose Clarke subdifferential is well-defined and $\partial J_f(\boldsymbol{x}) \ni G_f(\boldsymbol{x})$ for all $\boldsymbol{x}$, then we call $J_f$ a *virtual loss* of $f$.

If a virtual loss $J_f$ is well-defined for $f$, the update of SAM (2) on $f$ is equivalent to a (sub)gradient descent update on the virtual loss $J_f(\boldsymbol{x})$, which means, $\boldsymbol{x}_{t+1} = \boldsymbol{x}_t - \eta G_f(\boldsymbol{x}_t)$. The reason why we have to use Clarke subdifferential to define the virtual loss is because even for a differentiable and smooth function $f$, there are cases where the virtual gradient map $G_f$ is discontinuous and the virtual loss $J_f$ (if exists) is nonsmooth; see the discussion below Theorem 3.1.

Note that if the differentiable function $f$ is one-dimensional ($d = 1$), the virtual loss $J_f(x)$ is always *well-defined* because it can be obtained by simply (Lebesgue) integrating $G_f(x)$. However, in case of multi-dimensional functions ($d > 1$), there is no such guarantee, although $G_f$ is always well-defined. We emphasize that the virtual gradient map $G_f$ and virtual loss $J_f$ are mainly used for a better intuitive understanding of our (non-)convergence results. In formal proofs, we use them for analysis only if $J_f$ is well-defined.

Lastly, we note that Bartlett et al. [4] also employ a similar idea of virtual loss. In case of convex quadratic objective function $f(\boldsymbol{x})$, the authors define $\boldsymbol{u}_t = (-1)^t \boldsymbol{x}_t$ and formulate a (different) virtual loss $\tilde{J}_f(\boldsymbol{u})$ such that a SAM update on $f(\boldsymbol{x}_t)$ is equivalent to a GD update on $\tilde{J}_f(\boldsymbol{u}_t)$.

## 3 Convergence Analysis Under Deterministic Settings

In this section, we present the main results on the (non-)convergence of deterministic SAM with *constant* perturbation size $\rho$ and gradient normalization. We study four function classes: smooth strongly convex, smooth convex, smooth nonconvex, and nonsmooth Lipschitz convex functions.

### 3.1 Smooth and Strongly Convex Functions

For smooth strongly convex functions, we prove a global convergence guarantee for the best iterate.

**Theorem 3.1.** *Consider a $\beta$-smooth and $\mu$-strongly convex function $f$. If we run deterministic SAM starting at $\boldsymbol{x}_0$ with any perturbation size $\rho > 0$ and step size $\eta = \min\left\{\frac{1}{\mu T} \max\left\{1, \log\left(\frac{\mu^5 \Delta T^2}{\beta^6 \rho^2}\right)\right\}, \frac{1}{2\beta}\right\}$ to minimize $f$, we have*

$$\min_{t \in \{0,...,T\}} f(\boldsymbol{x}_t) - f^* = \tilde{\mathcal{O}}\left(\exp\left(-\frac{\mu T}{2\beta}\right)\Delta + \frac{\beta^6 \rho^2}{\mu^5 T^2}\right).$$

The proof of Theorem 3.1 can be found in Appendix B.2. As for relevent existing studies, Theorem 1 of Dai et al. [13] can be adapted to establish the convergence guarantee $f(\boldsymbol{x}_T) - f^* = \tilde{\mathcal{O}}\left(\frac{1}{T}\right)$, by selecting the step size $\eta = \min\left\{\frac{1}{\mu T} \max\left\{1, \log\left(\frac{\mu^2 \Delta T}{\beta^3 \rho^2}\right)\right\}, \frac{1}{\beta}\right\}$ and employing a similar proof technique as ours. We highlight that Theorem 3.1 achieves a faster convergence rate compared to the concurrent bound by Dai et al. [13]. Moreover, Theorem 11 of Andriushchenko and Flammarion [2] proves the convergence guarantee for this function class: $\|\boldsymbol{x}_T - \boldsymbol{x}^*\|^2 = \mathcal{O}(\exp(-T))$, but assuming

SAM *without gradient normalization*, and the boundedness of $\rho$. Here, we get a slower convergence rate of $\tilde{\mathcal{O}}\left(\exp(-T) + \frac{1}{T^2}\right)$, but with any $\rho > 0$ and with normalization.

By viewing SAM as GD on virtual loss, we can get an intuition why SAM cannot achieve exponential convergence. Consider a smooth and strongly convex function: $f(x) = \frac{1}{2}x^2$. One possible virtual loss of $f$ is $J_f(x) = \frac{1}{2}(x + \rho\mathrm{sign}(x))^2$, which is a *nonsmooth* and strongly convex function, for which the exponential convergence of GD is impossible [8].

Indeed, we can show that the sublinear convergence rate in Theorem 3.1 is not an artifact of our analysis. Interestingly, the $\tilde{\mathcal{O}}\left(\exp(-T) + \frac{1}{T^2}\right)$ rate given in Theorem 3.1 is *tight* in terms of $T$, up to logarithmic factors. Our next theorem provides a lower bound for smooth strongly convex functions.

**Theorem 3.2.** *Suppose $\frac{\beta}{\mu} \geq 2$. For any choice of perturbation size $\rho$, step size $\eta$, and initialization $x_0$, there exists a differentiable, $\beta$-smooth, and $\mu$-strongly convex function $f$ such that*

$$\min_{t \in \{0,...,T\}} f(x_t) - f^* = \Omega\left(\frac{\beta^3 \rho^2}{\mu^2 T^2}\right)$$

*holds for deterministic SAM iterates.*

For the proof of Theorem 3.2, refer to Appendix B.3. By comparing the rates in Theorems 3.1 and 3.2, we can see that the two bounds are tight in terms of $T$ and $\rho$. The bounds are a bit loose in terms of $\beta$ and $\mu$, but we believe that this may be partly due to our construction; in proving lower bounds, we used one-dimensional quadratic functions as worst-case examples. A more sophisticated construction may improve the tightness of the lower bound, which is left for future work.

### 3.2 Smooth and Convex Functions

For smooth and convex functions, proving convergence of the function value to global optimum becomes more challenging, due to the absence of strong convexity. In fact, we can instead prove that the gradient norm converges to zero.

**Theorem 3.3.** *Consider a $\beta$-smooth and convex function $f$. If we run deterministic SAM starting at $x_0$ with any perturbation size $\rho > 0$ and step size $\eta = \min\left\{\frac{\sqrt{2\Delta}}{\sqrt{\beta^3 \rho^2 T}}, \frac{1}{2\beta}\right\}$ to minimize $f$, we have*

$$\frac{1}{T}\sum_{t=0}^{T-1} \|\nabla f(x_t)\|^2 = \mathcal{O}\left(\frac{\beta\Delta}{T} + \frac{\sqrt{\Delta\beta^3\rho^2}}{\sqrt{T}}\right).$$

The proof of Theorem 3.3 is given in Appendix B.4. As for relevant existing studies, Theorem 11 of Andriushchenko and Flammarion [2] proves the convergence guarantee for this function class: $f(\bar{x}) - f^* = \mathcal{O}\left(\frac{1}{T}\right)$, where $\bar{x}$ indicates the averaged $x$ over $T$ iterates, while assuming SAM *without gradient normalization* and bounded $\rho$. Here, we prove a weaker result: a convergence rate of $\mathcal{O}(\frac{1}{\sqrt{T}})$ to *stationary points*, albeit for any $\rho > 0$ and with normalization.

One might expect that a reasonably good optimizer should converge to global minima of smooth convex functions. However, it turns out that both showing convergence to global minima and finding a non-convergence example are quite challenging in this function class.

Indeed, we later show non-convergence examples for other relevant settings. For stochastic SAM, we provide a non-convergence example (Theorem 4.4) in the same smooth and convex function class, showing that the suboptimality gap in terms of function values cannot have an upper bound, and therefore rendering convergence to global minima impossible. For deterministic SAM in nonsmooth Lipschitz convex functions, we show an example (Theorem 3.6) where convergence to the global minimum is possible only up to a certain distance proportional to $\rho$.

Given these examples, we suspect that there may also exist a non-convergence example for the deterministic SAM in this smooth convex setting. We leave settling this puzzle to future work.

### 3.3 Smooth and Nonconvex Functions

Existing studies prove that SAM (and its variants) with decaying or sufficiently small perturbation size $\rho$ converges to stationary points for smooth nonconvex functions [2, 18, 27, 31, 35]. Unfortunately, with constant perturbation size $\rho$, SAM exhibits a different convergence behavior: it does not converge all the way to stationary points.

**Theorem 3.4.** *Consider a $\beta$-smooth function $f$ satisfying $f^* = \inf_{\boldsymbol{x}} f(\boldsymbol{x}) > -\infty$. If we run deterministic SAM starting at $\boldsymbol{x}_0$ with any perturbation size $\rho > 0$ and step size $\eta = \frac{1}{\beta}$ to minimize $f$, we have*

$$\frac{1}{T} \sum_{t=0}^{T-1} \|\nabla f(\boldsymbol{x}_t)\|^2 \leq \mathcal{O}\left(\frac{\beta \Delta}{T}\right) + \beta^2 \rho^2.$$

Refer to the Appendix B.5 for the proof of Theorem 3.4. For a comparison, Theorem 9 of Andriushchenko and Flammarion [2] proves the convergence for this function class: $\frac{1}{T} \sum_{t=0}^{T} \|\nabla f(\boldsymbol{x}_t)\|^2 = \mathcal{O}\left(\frac{1}{T}\right)$, but again assuming SAM *without gradient normalization*, and boundedness of $\rho$. Our Theorem 3.4 guarantees $\mathcal{O}(\frac{1}{T})$ convergence up to an additive factor $\beta^2 \rho^2$.

One might speculate that the undesirable additive factor $\beta^2 \rho^2$ is an artifact of our analysis. The next theorem presents an example which proves that this extra term is in fact unavoidable.

**Theorem 3.5.** *For any $\rho > 0$ and $\eta \leq \frac{1}{\beta}$, there exists a $\beta$-smooth and $\Theta(\beta \rho)$-Lipschitz continuous function such that, if deterministic SAM is initialized at a point $\boldsymbol{x}_0$ sampled from a continuous probability distribution, then deterministic SAM converges to a nonstationary point, located at a distance of $\Omega(\rho)$ from a stationary point, with probability $1$.*

Here we present a brief outline for the proof of Theorem 3.5. For a given $\rho$, consider a one-dimensional function $f(x) = \frac{9\beta \rho^2}{25\pi^2} \sin\left(\frac{5\pi}{3\rho} x\right)$. Figure 2(a) demonstrates the virtual loss for this example. By examining the virtual loss $J_f$ and its stationary points, we observe that SAM iterates $x_t$ converge to a non-stationary point, located at a distance of $\Omega(\rho)$ from the stationary point. This means that the limit point of SAM will have gradient norm of order $\Omega(\beta \rho)$, thereby proving that the additive factor in Theorem 3.4 is tight in terms of $\rho$. A detailed analysis of Theorem 3.5 is provided in Appendix B.6.

**Remark: why do we ignore $\eta = \Omega(\frac{1}{\beta})$?** As the reader may have noticed, Theorem 3.5 only considers the case $\eta = \mathcal{O}(\frac{1}{\beta})$, and hence does not show that the additive factor is inevitable for *any* choice of $\eta > 0$. However, one can notice that if $\eta = \Omega(\frac{1}{\beta})$, then we can consider a one-dimensional $\Omega(\beta)$-strongly convex quadratic function and show that the SAM iterates blow up to infinity. For the same reason, in our other non-convergence results (Theorems 4.2 and 4.4), we only focus on $\eta = \mathcal{O}(\frac{1}{\beta})$.

### 3.4 Nonsmooth Lipschitz Convex Functions

Previous theorems study convergence of SAM assuming smoothness. The next theorem shows an example where SAM on a nonsmooth convex function converges only up to $\Omega(\rho)$ distance from the global minimum $\boldsymbol{x}^*$. This means that for constant perturbation size $\rho$, there exist *convex* functions that prevent SAM from converging to global minima. In Appendix B.7, we prove the following:

**Theorem 3.6.** *For any $\rho > 0$ and $\eta < \frac{7\rho}{4}$, there exists a nonsmooth Lipschitz convex function $f$ such that for some initialization, deterministic SAM converges to suboptimal points located at a distance of $\Omega(\rho)$ from the global minimum.*

## 4 Convergence Analysis Under Stochastic Settings

In this section, we present the main results on the convergence analysis of stochastic SAM, again with *time-invariant (constant)* perturbation size $\rho$ and *gradient normalization*. We consider both types of stochastic SAM: $n$-SAM and $m$-SAM, defined in Section 2.2. We study four types of function classes: smooth strongly convex, smooth convex, smooth nonconvex, and smooth Lipschitz nonconvex functions, under the assumption that the gradient oracle has bounded variance of $\sigma^2$ (Assumption 2.5). Our results in this section reveal that stochastic SAM exhibits different convergence properties compared to deterministic SAM.

### 4.1 Smooth and Strongly Convex Functions

Theorem 3.1 shows the convergence of deterministic SAM to global optima, for smooth strongly convex functions. Unlike this result, we find that under stochasticity and constant perturbation size $\rho$, both $n$-SAM and $m$-SAM ensure convergence *only up to an additive factor $\mathcal{O}(\rho^2)$*.

**Theorem 4.1.** *Consider a $\beta$-smooth, $\mu$-strongly convex function $f$, and assume Assumption 2.5. Under $n$-SAM, starting at $x_0$ with any perturbation size $\rho > 0$ and step size $\eta =$*

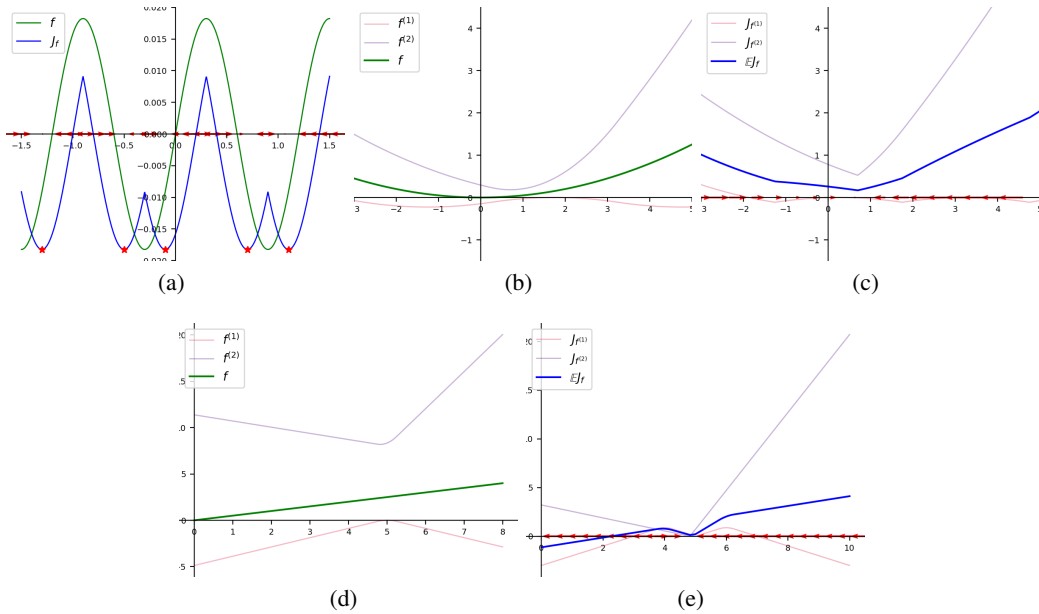

Figure 2: Examples of virtual loss plot for deterministic and stochastic SAM. The graph drawn in green indicates $f$, and the graph drawn in blue indicates $J_f$ (or $\mathbb{E}J_f$). Red arrows indicate the (expected) directions of SAM update. The (expected) updates are directed to red stars. (a) $f$ and $J_f$ in Theorem 3.5. (b) $f$ and its component functions $f^{(1)}$, $f^{(2)}$ in Theorem 4.2. (c) $\mathbb{E}J_f$ and its component functions $J_{f^{(1)}}$, $J_{f^{(2)}}$ in Theorem 4.2. (d) $f$ and its component functions $f^{(1)}$, $f^{(2)}$ in Theorem 4.4. (e) $\mathbb{E}J_f$ and its component functions $J_{f^{(1)}}$, $J_{f^{(2)}}$ in Theorem 4.4. For the simulation results of SAM trajectories on these functions, refer to Appendix D.

$\min\left\{\frac{1}{\mu T} \cdot \max\left\{1, \log\left(\frac{\mu^2 \Delta T}{\beta[\sigma^2 - \beta^2 \rho^2]_+}\right)\right\}, \frac{1}{2\beta}\right\}$ *to minimize $f$, we have*

$$\mathbb{E}f(\boldsymbol{x}_T) - f^* \leq \tilde{\mathcal{O}}\left(\exp\left(-\frac{\mu T}{2\beta}\right)\Delta + \frac{\beta[\sigma^2 - \beta^2 \rho^2]_+}{\mu^2 T}\right) + \frac{2\beta^2 \rho^2}{\mu}.$$

*Under $m$-SAM, additionally assuming $l(\cdot, \xi)$ is $\beta$-smooth for any $\xi$, the inequality continues to hold.*

For the proof, please refer to Appendix C.2. Theorem 4.1 provides a convergence rate of $\tilde{\mathcal{O}}(\frac{1}{T})$ to global minima, but only up to suboptimality gap $\frac{2\beta^2 \rho^2}{\mu}$. If $\sigma \leq \beta\rho$, then Theorem 4.1 becomes:

$$\mathbb{E}f(\boldsymbol{x}_T) - f^* \leq \tilde{\mathcal{O}}\left(\exp\left(-\frac{\mu T}{2\beta}\right)\Delta\right) + \frac{2\beta^2 \rho^2}{\mu},$$

thereby showing a convergence rate of $\mathcal{O}\left(\exp(-T)\right)$ modulo the additive factor.

For relevant existing studies, Theorem 2 of Andriushchenko and Flammarion [2] proves the convergence guarantee for smooth Polyak-Łojasiewicz functions: $\mathbb{E}f(\boldsymbol{x}_T) - f^* = \mathcal{O}\left(\frac{1}{T}\right)$, but assuming stochastic SAM *without gradient normalization*, and perturbation size $\rho$ decaying with $t$. In contrast, our analysis shows that the convergence property can be different with normalization and constant $\rho$.

The reader might be curious if the additional $\mathcal{O}(\rho^2)$ term can be removed. Our next theorem proves that in the case of high gradient noise ($\sigma > \beta\rho$), $m$-SAM with constant perturbation size $\rho$ cannot converge to global optima beyond a suboptimality gap $\Omega(\rho^2)$, when the component function $l(\boldsymbol{x};\xi)$ is smooth for any $\xi$. Hence, the additional term in Theorem 4.1 is *unavoidable*, at least for the more practical version $m$-SAM.

**Theorem 4.2.** *For any $\rho > 0, \beta > 0, \sigma > \beta\rho$ and $\eta \leq \frac{3}{10\beta}$, there exists a $\beta$-smooth and $\frac{\beta}{5}$-strongly convex function $f$ satisfying the following. (1) The function $f$ satisfies Assumption 2.5. (2) The component functions $l(\cdot;\xi)$ of $f$ are $\beta$-smooth for any $\xi$. (3) If we run $m$-SAM on $f$ initialized inside*

*a certain interval, then any arbitrary weighted average $\bar{x}$ of the iterates $x_0, x_1, \ldots$ must satisfy* $\mathbb{E}[f(\bar{x}) - f^*] \geq \Omega(\rho^2)$.

Here we provide a brief outline for Theorem 4.2. The in-depth analysis is provided in Appendix C.3. Given $\rho > 0$, $\beta > 0$, $\sigma > \beta\rho$, we consider a one-dimensional quadratic function $f(x)$ whose component function $l(x; \xi)$ is carefully chosen to satisfy the following for $x \in \left[\frac{\rho}{6}, \frac{13\rho}{6}\right]$:

$$f(x) = \mathbb{E}[l(x; \xi)] = \frac{\beta}{10}x^2, \quad l(x; \xi) = \begin{cases} -\frac{\beta}{10}\left(x - \frac{7\rho}{6}\right)^2, & \text{with probability } \frac{2}{3} \\ \frac{3\beta}{10}x^2 + \frac{\beta}{5}\left(x - \frac{7\rho}{6}\right)^2, & \text{otherwise.} \end{cases}$$

For values of $x$ outside this interval $\left[\frac{\rho}{6}, \frac{13\rho}{6}\right]$, each component function $l(x; \xi)$ takes the form of a strongly convex quadratic function. Figures 2(b) and 2(c) illustrate the original and virtual loss function plots of $l(x; \xi)$. The *local concavity* of component function plays a crucial role in making an attracting basin in the virtual loss, and this leads to an interval $\left[\frac{\rho}{6}, \frac{13\rho}{6}\right]$ bounded away from the global minimum 0, from which $m$-SAM iterates cannot escape.

For this scenario, we obtain $f(x_t) - f^* = \Omega(\rho^2)$ and $\|\nabla f(x_t)\|^2 = \Omega(\rho^2)$ for all iterates. From this, we can realize that the additive factor in Theorem 4.1 for $m$-SAM is unavoidable in the $\sigma > \beta\rho$ regime, and tight in terms of $\rho$. Moreover, the proof of Theorem 4.2 in Appendix C.3 reveals that even in the $\sigma \leq \beta\rho$ case, an analogous example gives $\|x_t - x^*\| = \Omega(\rho)$ for all iterates; hence, $m$-SAM fails to converge all the way to the global minimum in the small $\sigma$ regime as well.

## 4.2 Smooth and Convex Functions

We now move on to smooth convex functions and investigate the convergence guarantees of stochastic SAM for this function class. As can be guessed from Theorem 3.3, our convergence analysis in this section focuses on finding stationary points. Below, we provide a bound that ensures convergence to stationary points up to an additive factor.

**Theorem 4.3.** *Consider a $\beta$-smooth, convex function $f$, and assume Assumption 2.5. Under $n$-SAM, starting at $x_0$ with any perturbation size $\rho > 0$ and step size $\eta = \min\left\{\frac{\sqrt{\Delta}}{\sqrt{\beta[\sigma^2 - \beta^2\rho^2]_+ T}}, \frac{1}{2\beta}\right\}$ to minimize $f$, we have*

$$\frac{1}{T}\sum_{t=0}^{T-1}\mathbb{E}\|\nabla f(x_t)\|^2 = \mathcal{O}\left(\frac{\beta\Delta}{T} + \frac{\sqrt{\beta[\sigma^2 - \beta^2\rho^2]_+ \Delta}}{\sqrt{T}}\right) + 4\beta^2\rho^2.$$

*Under $m$-SAM, additionally assuming $l(\cdot, \xi)$ is $\beta$-smooth for any $\xi$, the inequality continues to hold.*

The proof of Theorem 4.3 can be found in Appendix C.4. Theorem 4.3 obtains a bound of $\mathcal{O}(\frac{1}{\sqrt{T}})$ modulo an additive factor $4\beta^2\rho^2$. Similar to Theorem 4.1, if $\sigma \leq \beta\rho$, then Theorem 4.3 reads

$$\frac{1}{T}\sum_{t=0}^{T-1}\mathbb{E}\|\nabla f(x_t)\|^2 = \mathcal{O}\left(\frac{\beta\Delta}{T}\right) + 4\beta^2\rho^2,$$

hence showing a convergence rate of $\mathcal{O}(\frac{1}{T})$ modulo the additive factor. Since the non-convergence example in Theorem 4.2 provides a scenario that $\mathbb{E}\|\nabla f(x_t)\|^2 = \Omega(\rho^2)$ for all $t$, we can see that the extra term is inevitable and also tight in terms of $\rho$.

Theorem 4.3 sounds quite weak, as it only proves convergence to a stationary point only up to an extra term. One could anticipate that stochastic SAM may actually converge to global minima of smooth convex functions modulo the unavoidable additive factor. However, as briefly mentioned in Section 3.2, the next theorem presents a counterexample illustrating that ensuring convergence to global minima, even up to an additive factor, is impossible for $m$-SAM.

**Theorem 4.4.** *For any $\rho > 0$, $\beta > 0$, $\sigma > 0$, and $\eta \leq \frac{1}{\beta}$, there exists a $\beta$-smooth and convex function $f$ satisfying the following. (1) The function $f$ satisfies Assumption 2.5. (2) The component functions $l(\cdot; \xi)$ of $f$ are $\beta$-smooth for any $\xi$. (3) If we run $m$-SAM on $f$ initialized inside a certain interval, then any arbitrary weighted average $\bar{x}$ of the iterates $x_0, x_1, \ldots$ must satisfy $\mathbb{E}[f(\bar{x}) - f^*] \geq C$, and the suboptimality gap $C$ can be made arbitrarily large and independent of the parameter $\rho$.*

Here, we present the intuitions of the proof for Theorem 4.4. As demonstrated in Figures 2(d) and 2(e), the *local concavity* of the component function significantly influences the formation of attracting

basins in its virtual loss, thereby creating a region from which the $m$-SAM updates get stuck inside the basin forever.

Also note that we can construct the function to form the basin at any point with arbitrary large function value (and hence large suboptimality gap). Therefore, establishing an upper bound on the convergence of function value becomes impossible in smooth convex functions. A detailed analysis for the non-convergence example is presented in Appendix C.5.

### 4.3 Smooth and Nonconvex Functions

We now study smooth nonconvex functions. Extending Theorem 3.4, we can show the following bound for stochastic $n$-SAM.

**Theorem 4.5.** *Consider a $\beta$-smooth function $f$ satisfying $f^* = \inf_x f(\boldsymbol{x}) > -\infty$, and assume Assumption 2.5. Under $n$-SAM, starting at $\boldsymbol{x}_0$ with any perturbation size $\rho > 0$ and step size $\eta = \min\left\{\frac{1}{2\beta}, \frac{\sqrt{\Delta}}{\sqrt{\beta\sigma^2 T}}\right\}$ to minimize $f$, we have*

$$\frac{1}{T}\sum\nolimits_{t=0}^{T-1}\mathbb{E}\|\nabla f(\boldsymbol{x}_t)\|^2 \leq \mathcal{O}\left(\frac{\beta\Delta}{T} + \frac{\sqrt{\beta\sigma^2\Delta}}{\sqrt{T}}\right) + \beta^2\rho^2.$$

The proof of Theorem 4.5 is provided in Appendix C.6. Notice that the non-convergence example presented in Theorem 3.5 (already) illustrates a scenario where $\mathbb{E}\|\nabla f(x_t)\|^2 = \Omega(\rho^2)$, thereby confirming the tightness of additive factor in terms of $\rho$.

The scope of applicability for Theorem 4.5 is limited to $n$-SAM. Compared to $n$-SAM, $m$-SAM employs a stronger assumption where $\xi = \tilde{\xi}$. By imposing an additional Lipschitzness condition on the function $f$, $m$-SAM leads to a similar but different convergence result.

**Theorem 4.6.** *Consider a $\beta$-smooth, $L$-Lipschitz continuous function $f$ satisfying $f^* = \inf_x f(\boldsymbol{x}) > -\infty$, and assume Assumption 2.5. Additionally assume $l(\cdot, \xi)$ is $\beta$-smooth for any $\xi$. Under $m$-SAM, starting at $\boldsymbol{x}_0$ with any perturbation size $\rho > 0$ and step size $\eta = \frac{\sqrt{\Delta}}{\sqrt{\beta(\sigma^2+L^2)T}}$ to minimize $f$, we have*

$$\frac{1}{T}\sum\nolimits_{t=0}^{T-1}\mathbb{E}\left[(\|\nabla f(\boldsymbol{x}_t)\| - \beta\rho)^2\right] \leq \mathcal{O}\left(\frac{\sqrt{\beta\Delta(\sigma^2+L^2)}}{\sqrt{T}}\right) + 5\beta^2\rho^2.$$

**Corollary 4.7.** *Under the setting of Theorem 4.6, we get*

$$\min_{t\in\{0,\ldots,T\}}\{\mathbb{E}\|\nabla f(\boldsymbol{x}_t)\|\} \leq \mathcal{O}\left(\frac{\left(\beta\Delta(\sigma^2+L^2)\right)^{1/4}}{T^{1/4}}\right) + \left(1+\sqrt{5}\right)\beta\rho.$$

The proofs for Theorem 4.6 and Corollary 4.7 are given in Appendix C.7. Since Theorem 3.5 presents an example where $\mathbb{E}\|\nabla f(\boldsymbol{x}_t)\| = \Omega(\rho)$ and $\mathbb{E}(\|\nabla f(\boldsymbol{x}_t)\| - \beta\rho)^2 = \Omega(\rho^2)$, we can verify that the additive factors in Theorem 4.6 and Corollary 4.7 are tight in terms of $\rho$.

As for previous studies, Theorems 2, 12, and 18 of Andriushchenko and Flammarion [2] prove the convergence of $n,m$-SAM for smooth nonconvex functions: $\frac{1}{T}\sum_{t=0}^{T}\mathbb{E}\|\nabla f(\boldsymbol{x}_t)\|^2 = \mathcal{O}(\frac{1}{\sqrt{T}})$, but assuming SAM *without gradient normalization*, and *sufficiently small $\rho$*. For $m$-SAM on smooth Lipschitz nonconvex functions, Theorem 1 of Mi et al. [27] proves $\frac{1}{T}\sum_{t=0}^{T}\mathbb{E}\|\nabla f(\boldsymbol{x}_t)\|^2 = \tilde{\mathcal{O}}(\frac{1}{\sqrt{T}})$, while assuming *decaying $\rho$*. From our results, we demonstrate that such full convergence results are impossible for practical versions of stochastic SAM.

## 5 Conclusions

This paper studies the convergence properties of SAM, under constant $\rho$ and with gradient normalization. We establish convergence guarantees of deterministic SAM for smooth and (strongly) convex functions. To our surprise, we discover scenarios in which deterministic SAM (for smooth nonconvex functions) and stochastic $m$-SAM (for all function class considered) converge only up to *unavoidable* additive factors proportional to $\rho^2$. Our findings emphasize the drastically different characteristics of SAM with vs. without decaying perturbation size. Establishing tighter bounds in terms of $\beta$ and $\mu$, or searching for a non-convergence example that applies to $n$-SAM might be interesting future research directions.

**Acknowledgments**

This paper was supported by Institute of Information & communications Technology Planning & Evaluation (IITP) grant (No. 2019-0-00075, Artificial Intelligence Graduate School Program (KAIST)) funded by the Korea government (MSIT), two National Research Foundation of Korea (NRF) grants (No. NRF-2019R1A5A1028324, RS-2023-00211352) funded by the Korea government (MSIT), and a grant funded by Samsung Electronics Co., Ltd.

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

# Contents

# A   More Details on the Related Works of SAM

**Sharpness and Generalization**   Hochreiter and Schmidhuber [17] propose an algorithm for finding low-complexity models with high generalization performance by finding flat minima. Keskar et al. [21] show that the poor generalization performance of large-batch training is due to the fact that large-batch training tends to converge to sharp minima. Jiang et al. [19] empirically show correlation between sharpness and generalization performance.

Motivated by prior studies, Foret et al. [16] propose Sharpness-Aware Minimization (SAM) that focuses on finding flat minima by aiming to (approximately) minimize $f^{\mathrm{SAM}}(\boldsymbol{x}) = \max_{\|\boldsymbol{\epsilon}\| \le \rho} f(\boldsymbol{x} + \boldsymbol{\epsilon})$ instead of $f$. Empirical results [3, 9, 16, 20, 25] show outstanding generalization performance of SAM on various tasks and models, including recent state-of-the-art models such as ViTs, MLP-Mixers, and T5.

**Other Theoretical Properties of SAM.**   Bartlett et al. [4] prove that for quadratic functions, SAM dynamics oscillate and align to the eigenvector corresponding to the largest eigenvalue of Hessian, and show interpretation of SAM dynamics as GD on a "virtual loss". Wen et al. [33] prove that for sufficiently small perturbation size $\rho$ of SAM, its trajectory follows a sharpness reduction flow which minimizes the maximum eigenvalue of Hessian. Dai et al. [13] examine the properties of SAM with and without normalization. They specifically focus on the stabilization and "drift-along-minima" effects observed in SAM with normalization, which are not present in SAM without normalization. Compagnoni et al. [12] derive a continuous time SDE model for SAM with and without normalization, and employ the model to explain why SAM has a preference for flat minima. Furthermore, they demonstrate that SAM with and without normalization exhibit distinct implicit regularization properties, especially for $\rho = \mathcal{O}(\sqrt{\eta})$.

Agarwala and Dauphin [1] prove that SAM produces an Edge Of Stability [11] stabilization effect, at a lower eigenvalue than gradient descent, under quadratic function settings. Kim et al. [22] show that saddle point acts as an attractor in SAM dynamics, and stochastic SAM takes more time to escape saddle point than stochastic gradient descent. Behdin and Mazumder [5] study the implicit regularization perspective of SAM, and provide a theoretical explanation of high generalization performance of SAM.

While Agarwala and Dauphin [1], Behdin and Mazumder [5], Kim et al. [22] provide useful insights on theoretical aspects of SAM, we point out that all proofs are restricted to SAM *without gradient normalization* in ascent steps. Considering the vastly different behaviors of SAM with and without normalization (Figure 1), it is not immediately clear whether these insights carry over to the practical version.

# B    Proofs for (Non-)Convergence of full-batch SAM

In this section, we show detailed proofs and explanations regarding convergence of SAM with constant $\rho$. The SAM algorithm we consider is defined as a two-step method:

$$\begin{cases} \boldsymbol{y}_t = \boldsymbol{x}_t + \rho \frac{\nabla f(\boldsymbol{x}_t)}{\|\nabla f(\boldsymbol{x}_t)\|}, \\ \boldsymbol{x}_{t+1} = \boldsymbol{x}_t - \eta \nabla f(\boldsymbol{y}_t). \end{cases}$$

## B.1    Additional Function Class and Important Lemmas

In this section, we define an additional function class to use in convergence proofs. We also state and prove a few lemmas that are used in our theorem proofs.

**Definition B.1** (Polyak-Lojasiewicz). A function $f : \mathbb{R}^d \to \mathbb{R}$ satisfies $\mu$-PL condition if there exists $\mu > 0$ such that,

$$\frac{1}{2}\|\nabla f(\boldsymbol{x})\|^2 \geq \mu(f(\boldsymbol{x}) - f^*) \tag{4}$$

for all $\boldsymbol{x} \in \mathbb{R}^d$.

It is well-known that $\mu$-strongly convex functions are $\mu$-PL, but not vice versa.

**Lemma B.2.** *For a differentiable and $\mu$-strongly convex function $f$, we have*

$$\langle \nabla f(\boldsymbol{x}_t), \nabla f(\boldsymbol{y}_t) - \nabla f(\boldsymbol{x}_t) \rangle \geq \mu\rho\|\nabla f(\boldsymbol{x}_t)\|.$$

*For a differentiable and convex function $f$, the inequality continues to hold with $\mu = 0$.*

*Proof.* If a differentiable function $f$ is $\mu$-strongly convex, then its gradient map $\boldsymbol{x} \mapsto \nabla f(\boldsymbol{x})$ is $\mu$-strongly monotone, i.e.,

$$\langle \boldsymbol{y} - \boldsymbol{x}, \nabla f(\boldsymbol{y}) - \nabla f(\boldsymbol{x}) \rangle \geq \mu\|\boldsymbol{y} - \boldsymbol{x}\|^2, \quad \forall \boldsymbol{x}, \boldsymbol{y} \in \mathbb{R}^d.$$

Using this fact, we have

$$\begin{aligned} \langle \nabla f(\boldsymbol{x}_t), \nabla f(\boldsymbol{y}_t) - \nabla f(\boldsymbol{x}_t) \rangle &= \frac{\|\nabla f(\boldsymbol{x}_t)\|}{\rho} \left\langle \rho \frac{\nabla f(\boldsymbol{x}_t)}{\|\nabla f(\boldsymbol{x}_t)\|}, \nabla f(\boldsymbol{y}_t) - \nabla f(\boldsymbol{x}_t) \right\rangle \\ &= \frac{\|\nabla f(\boldsymbol{x}_t)\|}{\rho} \langle \boldsymbol{y}_t - \boldsymbol{x}_t, \nabla f(\boldsymbol{y}_t) - \nabla f(\boldsymbol{x}_t) \rangle \\ &\geq \frac{\mu\|\nabla f(\boldsymbol{x}_t)\|}{\rho} \|\boldsymbol{y}_t - \boldsymbol{x}_t\|^2 = \mu\rho\|\nabla f(\boldsymbol{x}_t)\|, \end{aligned}$$

and this finishes the proof. $\qquad\square$

**Lemma B.3.** *For a $\beta$-smooth and $\mu$-strongly convex function, with step size $\eta \leq \frac{1}{2\beta}$, we have*

$$f(\boldsymbol{x}_{t+1}) \leq f(\boldsymbol{x}_t) - \frac{\eta}{2}\|\nabla f(\boldsymbol{x}_t)\|^2 - \frac{\eta\mu\rho}{2}\|\nabla f(\boldsymbol{x}_t)\| + \frac{\eta^2\beta^3\rho^2}{2}.$$

*For a $\beta$-smooth and convex function, the inequality continues to hold with $\mu = 0$.*

*Proof.* Starting from the definition of $\beta$-smoothness, we have

$$\begin{aligned} f(\boldsymbol{x}_{t+1}) &\leq f(\boldsymbol{x}_t) + \langle \nabla f(\boldsymbol{x}_t), \boldsymbol{x}_{t+1} - \boldsymbol{x}_t \rangle + \frac{\beta}{2}\|\boldsymbol{x}_{t+1} - \boldsymbol{x}_t\|^2 \\ &= f(\boldsymbol{x}_t) - \eta\langle \nabla f(\boldsymbol{x}_t), \nabla f(\boldsymbol{y}_t) \rangle + \frac{\eta^2\beta}{2}\|\nabla f(\boldsymbol{y}_t)\|^2 \\ &= f(\boldsymbol{x}_t) - \eta\langle \nabla f(\boldsymbol{x}_t), \nabla f(\boldsymbol{y}_t) - \nabla f(\boldsymbol{x}_t) + \nabla f(\boldsymbol{x}_t) \rangle \\ &\quad + \frac{\eta^2\beta}{2}\|\nabla f(\boldsymbol{y}_t) - \nabla f(\boldsymbol{x}_t) + \nabla f(\boldsymbol{x}_t)\|^2 \\ &= f(\boldsymbol{x}_t) - \eta\langle \nabla f(\boldsymbol{x}_t), \nabla f(\boldsymbol{y}_t) - \nabla f(\boldsymbol{x}_t) \rangle - \eta\|\nabla f(\boldsymbol{x}_t)\|^2 + \frac{\eta^2\beta}{2}\|\nabla f(\boldsymbol{y}_t) - \nabla f(\boldsymbol{x}_t)\|^2 \end{aligned}$$

$$+ \frac{\eta^2 \beta}{2} \|\nabla f(\boldsymbol{x}_t)\|^2 + \eta^2 \beta \langle \nabla f(\boldsymbol{x}_t), \nabla f(\boldsymbol{y}_t) - \nabla f(\boldsymbol{x}_t) \rangle$$

$$= f(\boldsymbol{x}_t) - \eta \left(1 - \frac{\eta\beta}{2}\right) \|\nabla f(\boldsymbol{x}_t)\|^2 - \eta(1 - \eta\beta)\langle \nabla f(\boldsymbol{x}_t), \nabla f(\boldsymbol{y}_t) - \nabla f(\boldsymbol{x}_t) \rangle$$

$$+ \frac{\eta^2 \beta}{2} \|\nabla f(\boldsymbol{y}_t) - \nabla f(\boldsymbol{x}_t)\|^2.$$

Since we assumed $\eta \leq \frac{1}{2\beta}$, both $\eta \left(1 - \frac{\eta\beta}{2}\right) \geq \frac{\eta}{2}$ and $\eta(1 - \eta\beta) \geq \frac{\eta}{2}$ hold. Applying Lemma B.2 to the above, we get

$$f(\boldsymbol{x}_{t+1}) \leq f(\boldsymbol{x}_t) - \frac{\eta}{2}\|\nabla f(\boldsymbol{x}_t)\|^2 - \frac{\eta\mu\rho}{2}\|\nabla f(\boldsymbol{x}_t)\| + \frac{\eta^2\beta}{2}\|\nabla f(\boldsymbol{y}_t) - \nabla f(\boldsymbol{x}_t)\|^2$$

$$\leq f(\boldsymbol{x}_t) - \frac{\eta}{2}\|\nabla f(\boldsymbol{x}_t)\|^2 - \frac{\eta\mu\rho}{2}\|\nabla f(\boldsymbol{x}_t)\| + \frac{\eta^2\beta}{2}(\beta\|\boldsymbol{y}_t - \boldsymbol{x}_t\|)^2$$

$$= f(\boldsymbol{x}_t) - \frac{\eta}{2}\|\nabla f(\boldsymbol{x}_t)\|^2 - \frac{\eta\mu\rho}{2}\|\nabla f(\boldsymbol{x}_t)\| + \frac{\eta^2\beta^3\rho^2}{2},$$

and this finishes the proof. □

## B.2 Convergence Proof for Smooth and Strongly Convex Functions (Proof of Theorem 3.1)

In this section, we prove Theorem 3.1, restated below for the sake of convenience.

**Theorem 3.1.** *Consider a $\beta$-smooth and $\mu$-strongly convex function $f$. If we run deterministic SAM starting at $\boldsymbol{x}_0$ with any perturbation size $\rho > 0$ and step size $\eta = \min \left\{ \frac{1}{\mu T} \max \left\{ 1, \log \left( \frac{\mu^5 \Delta T^2}{\beta^6 \rho^2} \right) \right\}, \frac{1}{2\beta} \right\}$ to minimize $f$, we have*

$$\min_{t \in \{0, \dots, T\}} f(\boldsymbol{x}_t) - f^* = \tilde{\mathcal{O}} \left( \exp \left( -\frac{\mu T}{2\beta} \right) \Delta + \frac{\beta^6 \rho^2}{\mu^5 T^2} \right).$$

*Proof.* The proof is divided into two cases, based on the $\|\nabla f(\boldsymbol{x}_t)\|$ observed throughout the entire optimization process. In the first case, the gradient norm $\|\nabla f(\boldsymbol{x}_t)\|$ at $\boldsymbol{x}_t$ is sufficiently large for all $t = 0, \dots, T-1$; in this case, we show that the SAM algorithm converges linearly. The other scenario corresponds to the case where there exists an iteration index $t \in \{0, \dots, T-1\}$ such that $\|\nabla f(\boldsymbol{x}_t)\|$ is smaller than a certain threshold. In this case, we can show from the PL inequality (4) that we are already close to global optimality.

From Lemma B.3, we have

$$f(\boldsymbol{x}_{t+1}) \leq f(\boldsymbol{x}_t) - \frac{\eta}{2}\|\nabla f(\boldsymbol{x}_t)\|^2 - \frac{\eta\mu\rho}{2}\|\nabla f(\boldsymbol{x}_t)\| + \frac{\eta^2\beta^3\rho^2}{2}.$$

Now, recall from Definition B.1 that $\mu$-strongly convex functions satisfy $\mu$-PL inequaltiy (4). From this, we get

$$f(\boldsymbol{x}_{t+1}) - f^* \leq (1 - \eta\mu)(f(\boldsymbol{x}_t) - f^*) - \frac{\eta\mu\rho}{2}\|\nabla f(\boldsymbol{x}_t)\| + \frac{\eta^2\beta^3\rho^2}{2}. \tag{5}$$

**Case 1: when $\|\nabla f(\boldsymbol{x}_t)\|$ remains large.** Let us now consider the first case, where $\|\nabla f(\boldsymbol{x}_t)\| \geq \frac{\eta\beta^3\rho}{\mu}$ holds for all $t = 0, \dots, T-1$. In this case, (5) becomes

$$f(\boldsymbol{x}_{t+1}) - f^* \leq (1 - \eta\mu)(f(\boldsymbol{x}_t) - f^*),$$

which holds for all $t = 0, \dots, T-1$. Unrolling the inequality, we obtain

$$f(\boldsymbol{x}_T) - f^* \leq (1 - \eta\mu)^T(f(\boldsymbol{x}_0) - f^*).$$

**Case 2: when some $\|\nabla f(\boldsymbol{x}_t)\|$ is small.** In the other case where there exists $t$ such that $\|\nabla f(\boldsymbol{x}_t)\| \leq \frac{\eta\beta^3\rho}{\mu}$, notice from $\mu$-PL inequality (4) that

$$f(\boldsymbol{x}_t) - f^* \leq \frac{1}{2\mu}\|\nabla f(\boldsymbol{x}_t)\|^2 \leq \frac{\eta^2\beta^6\rho^2}{2\mu^3}.$$

Therefore, combining the two cases, it is guaranteed that

$$\min_{t \in \{0,\dots,T\}} f(\boldsymbol{x}_t) - f^* \leq (1 - \eta\mu)^T \Delta + \frac{\eta^2 \beta^6 \rho^2}{2\mu^3}. \tag{6}$$

We now elaborate how the choice of step size $\eta = \min\left\{ \frac{1}{\mu T} \max\left\{ 1, \log\left( \frac{\mu^5 \Delta T^2}{\beta^6 \rho^2} \right) \right\}, \frac{1}{2\beta} \right\}$ results in the convergence rate in the theorem statement. Naturally, there are four cases, depending on the outcomes of the min and max operations.

**Case A:** $\log\left( \frac{\mu^5 \Delta T^2}{\beta^6 \rho^2} \right) \geq 1$ **and** $\frac{1}{\mu T} \log\left( \frac{\mu^5 \Delta T^2}{\beta^6 \rho^2} \right) \leq \frac{1}{2\beta}$. Putting $\eta = \frac{1}{\mu T} \log\left( \frac{\mu^5 \Delta T^2}{\beta^6 \rho^2} \right)$ into (6),

$$\min_{t \in \{0,\dots,T\}} f(\boldsymbol{x}_t) - f^* \leq \frac{\beta^6 \rho^2}{\mu^5 T^2} + \frac{\beta^6 \rho^2}{2\mu^5 T^2} \log^2\left( \frac{\mu^5 \Delta T^2}{\beta^6 \rho^2} \right).$$

**Case B:** $\log\left( \frac{\mu^5 \Delta T^2}{\beta^6 \rho^2} \right) \geq 1$ **and** $\frac{1}{\mu T} \log\left( \frac{\mu^5 \Delta T^2}{\beta^6 \rho^2} \right) \geq \frac{1}{2\beta}$. Substituting $\eta = \frac{1}{2\beta} \leq \frac{1}{\mu T} \log\left( \frac{\mu^5 \Delta T^2}{\beta^6 \rho^2} \right)$ to (6),

$$\min_{t \in \{0,\dots,T\}} f(\boldsymbol{x}_t) - f^* \leq \left( 1 - \frac{\mu}{2\beta} \right)^T \Delta + \frac{\beta^6 \rho^2}{2\mu^3} \cdot \left( \frac{1}{2\beta} \right)^2$$

$$\leq \exp\left( -\frac{\mu T}{2\beta} \right) \Delta + \frac{\beta^6 \rho^2}{2\mu^5 T^2} \log^2\left( \frac{\mu^5 \Delta T^2}{\beta^6 \rho^2} \right).$$

**Case C:** $\log\left( \frac{\mu^5 \Delta T^2}{\beta^6 \rho^2} \right) \leq 1$ **and** $\frac{1}{\mu T} \leq \frac{1}{2\beta}$. Putting $\eta = \frac{1}{\mu T} \geq \frac{1}{\mu T} \log\left( \frac{\mu^5 \Delta T^2}{\beta^6 \rho^2} \right)$ into (6),

$$\min_{t \in \{0,\dots,T\}} f(\boldsymbol{x}_t) - f^* \leq \left( 1 - \frac{1}{T} \right)^T \Delta + \frac{\beta^6 \rho^2}{2\mu^5 T^2}$$

$$\leq \left( 1 - \frac{1}{T} \log\left( \frac{\mu^5 \Delta T^2}{\beta^6 \rho^2} \right) \right)^T \Delta + \frac{\beta^6 \rho^2}{2\mu^5 T^2} \leq \frac{\beta^6 \rho^2}{\mu^5 T^2} + \frac{\beta^6 \rho^2}{2\mu^5 T^2}.$$

**Case D:** $\log\left( \frac{\mu^5 \Delta T^2}{\beta^6 \rho^2} \right) \leq 1$ **and** $\frac{1}{\mu T} \geq \frac{1}{2\beta}$. By substituting $\eta = \frac{1}{2\beta} \leq \frac{1}{\mu T}$ to (6) we obtain

$$\min_{t \in \{0,\dots,T\}} f(\boldsymbol{x}_t) - f^* \leq \left( 1 - \frac{\mu}{2\beta} \right)^T \Delta + \frac{\beta^6 \rho^2}{2\mu^3} \cdot \left( \frac{1}{2\beta} \right)^2$$

$$\leq \exp\left( -\frac{\mu T}{2\beta} \right) \Delta + \frac{\beta^6 \rho^2}{2\mu^5 T^2}.$$

Combining the four cases, we conclude

$$\min_{t \in \{0,\dots,T\}} f(\boldsymbol{x}_t) - f^* = \tilde{\mathcal{O}}\left( \exp\left( -\frac{\mu T}{2\beta} \right) \Delta + \frac{\beta^6 \rho^2}{\mu^5 T^2} \right),$$

thereby completing the proof. $\qquad\square$

### B.3 Lower Bound Proof for Smooth and Strongly Convex Functions (Proof of Theorem 3.2)

In this section, we prove Theorem 3.2, restated below for the sake of convenience.

**Theorem 3.2.** *Suppose* $\frac{\beta}{\mu} \geq 2$. *For any choice of perturbation size $\rho$, step size $\eta$, and initialization $\boldsymbol{x}_0$, there exists a differentiable, $\beta$-smooth, and $\mu$-strongly convex function $f$ such that*

$$\min_{t \in \{0,\dots,T\}} f(\boldsymbol{x}_t) - f^* = \Omega\left( \frac{\beta^3 \rho^2}{\mu^2 T^2} \right)$$

*holds for deterministic SAM iterates.*

The proof is divided into three cases, depending on the step size. For each case, we will define and analyze a one-dimensional function to show the lower bound. In doing so, without loss of generality we will fix an initialization $x_0$ because we can appropriately shift the function for different choices of $x_0$.

1. For $\eta \leq \frac{1}{2\mu T}$, we show that there exists a function $f$ such that

$$\min_{t \in \{0,\dots,T\}} f(x_t) - f^* = \Omega\left(\frac{\beta^4 \rho^2}{\mu^3}\right).$$

2. For $\frac{1}{2\mu T} \leq \eta \leq \frac{2}{\beta}$, we show existence of a function $f$ that satisfies

$$\min_{t \in \{0,\dots,T\}} f(x_t) - f^* = \Omega\left(\frac{\beta^3 \rho^2}{\mu^2 T^2}\right).$$

3. For $\eta \geq \frac{2}{\beta}$, we show that there exists a function $f$ such that

$$\min_{t \in \{0,\dots,T\}} f(x_t) - f^* = \Omega\left(\frac{\beta^3 \rho^2}{\mu^2}\right).$$

Combining the three results, we can see that the suboptimality gap of the best iterate is at least $\Omega\left(\frac{\beta^3 \rho^2}{\mu^2 T^2}\right)$, hence proving the theorem. Below, we prove the statements for the three intervals.

**Case 1: $\eta \leq \frac{1}{2\mu T}$.** Consider a function

$$f(x) = \frac{1}{2}\mu x^2 - \mu \rho x, \quad \nabla f(x) = \mu(x - \rho).$$

Suppose we start at initialization $x_0 = \frac{2\beta^2 \rho}{\mu^2} \geq 8\rho$. Note from the definition of $\nabla f$ that for any $x_t \geq \rho$, we have $y_t = x_t + \rho \frac{\nabla f(x_t)}{|\nabla f(x_t)|} = x_t + \rho$ and $\nabla f(y_t) = \mu(x_t + \rho - \rho) = \mu x_t$. Therefore, the first SAM update can be written as

$$x_1 = x_0 - \eta \nabla f(y_0) = (1 - \eta\mu) x_0.$$

Since $1 - \eta\mu \geq 1 - \frac{1}{2T} \geq \frac{1}{2}$, the inequality $x_1 \geq \rho$ still holds and the same argument can be repeated for the second update, yielding

$$x_2 = x_1 - \eta \nabla f(y_1) = (1 - \eta\mu) x_1 = (1 - \eta\mu)^2 x_0.$$

In fact, $(1 - \eta\mu)^t \geq (1 - \frac{1}{2T})^T \geq \frac{1}{2}$ for all $t \leq T$, so all the iterates up to $x_T$ stay above $\rho$. Thus, for any $t = 0, \dots, T$, the iterate $x_t$ can be written and bounded from below as

$$x_t = (1 - \eta\mu)^t x_0 \geq \frac{x_0}{2} = \frac{\beta^2 \rho}{\mu^2}.$$

Therefore, in this case, the suboptimality gap of the best iterate is at least

$$\min_{t \in \{0,\dots,T\}} f(x_t) - f^* \geq f\left(\frac{\beta^2 \rho}{\mu^2}\right) - f(\rho) = \frac{\beta^4 \rho^2}{2\mu^3} - \frac{\beta^2 \rho^2}{\mu} + \frac{\mu \rho^2}{2} = \Omega\left(\frac{\beta^4 \rho^2}{\mu^3}\right).$$

**Case 2: $\frac{1}{2\mu T} \leq \eta \leq \frac{2}{\beta}$.** Consider a function

$$f(x) = \frac{1}{4}\beta x^2, \quad \nabla f(x) = \frac{1}{2}\beta x.$$

Suppose we start at initialization $x_0 = \frac{\eta\beta\rho}{4 - \eta\beta}$. We are going to show that the SAM iterates oscillate between $\pm\frac{\eta\beta\rho}{4 - \eta\beta}$. Indeed, if $x_t = \frac{\eta\beta\rho}{4 - \eta\beta}$, then $\nabla f(x_t) > 0$ and

$$y_t = \frac{\eta\beta\rho}{4 - \eta\beta} + \rho = \frac{4\rho}{4 - \eta\beta}, \quad \text{and} \quad \nabla f(y_t) = \frac{2\beta\rho}{4 - \eta\beta}.$$

Then, after SAM update, we get

$$x_{t+1} = x_t - \eta \nabla f(y_t) = \frac{\eta \beta \rho}{4 - \eta \beta} - \eta \frac{2\beta \rho}{4 - \eta \beta} = -\frac{\eta \beta \rho}{4 - \eta \beta}.$$

The same argument can be repeated to show that $x_{t+2} = -x_{t+1} = x_t$. As a result, the iterates oscillate between $\pm \frac{\eta \beta \rho}{4 - \eta \beta}$ forever. In this case, the suboptimality gap is bounded from below by

$$\min_{t \in \{0, \dots, T\}} f(x_t) - f^* \geq \frac{\beta}{4} \left( \frac{\eta \beta \rho}{4 - \eta \beta} \right)^2 \geq \frac{\eta^2 \beta^3 \rho^2}{64}.$$

Applying $\eta \geq \frac{1}{2\mu T}$ yields

$$\min_{t \in \{0, \dots, T\}} f(x_t) - f^* = \Omega \left( \frac{\beta^3 \rho^2}{\mu^2 T^2} \right).$$

**Case 3:** $\eta \geq \frac{2}{\beta}$.  Consider a function

$$f(x) = \frac{1}{2} \beta x^2, \quad \nabla f(x) = \beta x.$$

Let the initialization be $x_0 = \frac{\beta \rho}{\mu}$. For any $x_t \geq 0$, we have $y_t = x_t + \rho$ and $\nabla f(y_t) = \beta(x_t + \rho)$. Then, the resulting SAM update becomes

$$x_{t+1} = x_t - \eta \nabla f(y_t) = (1 - \eta \beta)x_t - \eta \rho \leq (1 - \eta \beta)x_t.$$

Since we have $\eta \geq \frac{2}{\beta}$, we have $1 - \eta \beta \leq -1$ and $x_{t+1}$ has the opposite sign as $x_t$ and its absolute value is at least as large as $|x_t|$. We can similarly check that any further SAM update changes the sign and does not decrease the absolute value. Therefore, for all $t = 0, \dots, T$, we have $|x_t| \geq |x_0|$.

Consequently, the suboptimality gap of the best iterate is at least

$$\min_{t \in \{0, \dots, T\}} f(x_t) - f^* \geq f(x_0) - f(0) = \Omega \left( \frac{\beta^3 \rho^2}{\mu^2} \right).$$

**Remarks on validity of lower bound.**  Lastly, we comment on why we choose different functions and initialization for different choices of $\eta$ and why it suffices to provide a matching lower bound for Theorem 3.1. In convergence upper bounds in the form of Theorem 3.1, we aim to prove an upper bound on the following *minimax risk*:

$$\inf_{A \in \mathcal{A}} \sup_{f \in \mathcal{F}} \zeta(A(f)), \tag{7}$$

where $\mathcal{A}$ denotes the class of *algorithms*, $\mathcal{F}$ denotes the class of *functions*, and $\zeta(A(f))$ is the *suboptimality measure* for algorithm's output $A(f)$ for function $f$. In the context of Theorem 3.1, our algorithm class $\mathcal{A}$ corresponds to the choices of the "hyperparameters" $\eta$, $\rho$, and $x_0$ of SAM, and the function class $\mathcal{F}$ here is the class analyzed by the theorem: the collection of $\beta$-smooth and $\mu$-strongly convex functions. From this viewpoint, Theorem 3.1 can be thought of as an upper bound on (7), with the choice of $\zeta(A(f)) \triangleq \min_{t \in \{0, \dots, T\}} f(x_t) - f^*$.

Hence, showing a matching lower bound for Theorem 3.1 amounts to showing a matching lower bound for the minimax risk (7). For this purpose, it suffices to show that *for each $A \in \mathcal{A}$, there exists* a choice of $f \in \mathcal{F}$ such that a certain lower bound holds. Therefore, we are allowed to choose different choices of $f$ for each different choice of hyperparameters $\eta$, $\rho$, and $x_0$. In fact, in our proof, we choose different choices of $f$ and $x_0$ for each different choice of $\eta$; however, this is without loss of generality once we notice here that starting an algorithm at $x_0$ to minimize $f(x)$ is equivalent to starting at $0$ to minimize $f(x - x_0)$. Hence, even though we choose different $f$'s and $x_0$'s for different choices of $\eta$ in our proof of Theorem 3.2, this is sufficient for providing a matching upper bound for Theorem 3.1.

Admittedly, some authors show stronger versions than what we show, where a single function takes care of all possible hyperparameters. Indeed, such results provide lower bounds for $\sup_{f \in \mathcal{F}} \inf_{A \in \mathcal{A}} \zeta(A(f))$. Recalling that $\sup \inf \leq \inf \sup$, one can notice that these $\sup \inf$ lower bounds are in fact much stronger than what suffices.

## B.4  Convergence Proof for Smooth and Convex Functions (Proof of Theorem 3.3)

For smooth and convex function, we prove the convergence of gradient norm to zero. The theorem is restated for convenience.

**Theorem 3.3.** *Consider a $\beta$-smooth and convex function $f$. If we run deterministic SAM starting at $x_0$ with any perturbation size $\rho > 0$ and step size $\eta = \min\left\{\frac{\sqrt{2\Delta}}{\sqrt{\beta^3\rho^2 T}}, \frac{1}{2\beta}\right\}$ to minimize $f$, we have*

$$\frac{1}{T}\sum_{t=0}^{T-1}\|\nabla f(x_t)\|^2 = \mathcal{O}\left(\frac{\beta\Delta}{T} + \frac{\sqrt{\Delta\beta^3\rho^2}}{\sqrt{T}}\right).$$

*Proof.* Using Lemma B.3, for smooth and convex function $f$, we have

$$f(x_{t+1}) \le f(x_t) - \frac{\eta}{2}\|\nabla f(x_t)\|^2 + \frac{\eta^2\beta^3\rho^2}{2},$$

which can be rewritten as

$$\|\nabla f(x_t)\|^2 \le \frac{2}{\eta}(f(x_t) - f(x_{t+1})) + \eta\beta^3\rho^2.$$

Adding up the inequality for $t = 0, \ldots, T-1$, and the dividing both sides by $T$, we get

$$\frac{1}{T}\sum_{t=0}^{T-1}\|\nabla f(x_t)\|^2 \le \frac{2}{\eta T}(f(x_0) - f(x_T)) + \eta\beta^3\rho^2 \le \frac{2}{\eta T}\Delta + \eta\beta^3\rho^2. \tag{8}$$

We now spell out how the choice of step size $\eta = \min\left\{\sqrt{\frac{2\Delta}{\beta^3\rho^2 T}}, \frac{1}{2\beta}\right\}$ results in the convergence rate in the theorem statement. There are two cases to be considered, depending on the outcome of the $\min$ operation.

**Case A:** $\sqrt{\frac{2\Delta}{\beta^3\rho^2 T}} \le \frac{1}{2\beta}$.  Putting $\eta = \sqrt{\frac{2\Delta}{\beta^3\rho^2 T}}$ into (8), we get

$$\frac{1}{T}\sum_{t=0}^{T-1}\|\nabla f(x_t)\|^2 \le \frac{\sqrt{2\Delta\beta^3\rho^2}}{\sqrt{T}} + \frac{\sqrt{2\Delta\beta^3\rho^2}}{\sqrt{T}} = \frac{2\sqrt{2\Delta\beta^3\rho^2}}{\sqrt{T}}.$$

**Case B:** $\sqrt{\frac{2\Delta}{\beta^3\rho^2 T}} \ge \frac{1}{2\beta}$.  Substituting $\eta = \frac{1}{2\beta} \le \sqrt{\frac{2\Delta}{\beta^3\rho^2 T}}$ to (8) yields

$$\frac{1}{T}\sum_{t=0}^{T-1}\|\nabla f(x_t)\|^2 \le \frac{4\beta\Delta}{T} + \beta^3\rho^2 \cdot \frac{1}{2\beta} \le \frac{4\beta\Delta}{T} + \frac{\sqrt{2\Delta\beta^3\rho^2}}{\sqrt{T}}.$$

Merging the two cases, we conclude

$$\frac{1}{T}\sum_{t=0}^{T-1}\|\nabla f(x_t)\|^2 = \mathcal{O}\left(\frac{\beta\Delta}{T} + \frac{\sqrt{\Delta\beta^3\rho^2}}{\sqrt{T}}\right),$$

completing the proof. $\qquad\square$

## B.5  Convergence Proof for Smooth and Nonconvex Functions (Proof of Theorem 3.4)

In this section, we prove the convergence up to an additive factor $\beta^2\rho^2$ for smooth and nonconvex functions. The theorem is restated for convenience.

**Theorem 3.4.** *Consider a $\beta$-smooth function $f$ satisfying $f^* = \inf_x f(x) > -\infty$. If we run deterministic SAM starting at $x_0$ with any perturbation size $\rho > 0$ and step size $\eta = \frac{1}{\beta}$ to minimize $f$, we have*

$$\frac{1}{T}\sum_{t=0}^{T-1}\|\nabla f(x_t)\|^2 \le \mathcal{O}\left(\frac{\beta\Delta}{T}\right) + \beta^2\rho^2.$$

*Proof.* Starting from the definition of $\beta$-smoothness, we have

$$f(\boldsymbol{x}_{t+1}) \leq f(\boldsymbol{x}_t) - \eta\langle\nabla f(\boldsymbol{x}_t), \nabla f(\boldsymbol{y}_t)\rangle + \frac{\eta^2\beta}{2}\|\nabla f(\boldsymbol{y}_t)\|^2$$

$$= f(\boldsymbol{x}_t) - \frac{\eta}{2}\|\nabla f(\boldsymbol{x}_t)\|^2 - \frac{\eta}{2}\|\nabla f(\boldsymbol{y}_t)\|^2 + \frac{\eta}{2}\|\nabla f(\boldsymbol{x}_t) - \nabla f(\boldsymbol{y}_t)\|^2 + \frac{\eta^2\beta}{2}\|\nabla f(\boldsymbol{y}_t)\|^2$$

$$\leq f(\boldsymbol{x}_t) - \frac{\eta}{2}\|\nabla f(\boldsymbol{x}_t)\|^2 + \frac{\eta\beta^2}{2}\|\boldsymbol{x}_t - \boldsymbol{y}_t\|^2$$

$$= f(\boldsymbol{x}_t) - \frac{\eta}{2}\|\nabla f(\boldsymbol{x}_t)\|^2 + \frac{\eta\beta^2\rho^2}{2}.$$

The inequality can be rearranged as

$$\|\nabla f(\boldsymbol{x}_t)\|^2 \leq \frac{2}{\eta}\left(f(\boldsymbol{x}_t) - f(\boldsymbol{x}_{t+1})\right) + \beta^2\rho^2.$$

Adding up the inequality for $t = 0, \ldots, T-1$, and the dividing both sides by $T$, we get

$$\frac{1}{T}\sum_{t=0}^{T-1}\|\nabla f(\boldsymbol{x}_t)\|^2 \leq \frac{2}{\eta T}\left(f(\boldsymbol{x}_0) - f(\boldsymbol{x}_T)\right) + \beta^2\rho^2 \leq \frac{2\Delta}{\eta T} + \beta^2\rho^2. \tag{9}$$

Substituting $\eta = \frac{1}{\beta}$ to (9) yields

$$\frac{1}{T}\sum_{t=0}^{T-1}\|\nabla f(\boldsymbol{x}_t)\|^2 \leq \frac{2\beta\Delta}{T} + \beta^2\rho^2.$$

$\square$

## B.6 Non-convergence for a Smooth and Nonconvex Function (Proof of Theorem 3.5)

In this section, we spell out the proof of our counterexample that SAM with constant perturbation size $\rho$ provably fails to converge to a stationary point. Here we restate the theorem.

**Theorem 3.5.** *For any $\rho > 0$ and $\eta \leq \frac{1}{\beta}$, there exists a $\beta$-smooth and $\Theta(\beta\rho)$-Lipschitz continuous function such that, if deterministic SAM is initialized at a point $\boldsymbol{x}_0$ sampled from a continuous probability distribution, then deterministic SAM converges to a nonstationary point, located at a distance of $\Omega(\rho)$ from a stationary point, with probability $1$.*

*Proof.* For $x \in \mathbb{R}$, consider the following one-dimensional function: given $\rho$ and $\beta$,

$$f(x) = \frac{9\beta\rho^2}{25\pi^2}\sin\left(\frac{5\pi}{3\rho}x\right).$$

It is easy to check that this function is $\beta$-smooth and $\frac{3\beta\rho}{5\pi}$-Lipschitz continuous. Also, let

$$\mathbb{X} = \{x \mid x = (0.3 + 0.6k)\rho, k \in \mathbb{Z}\},$$

which is the set of points $x$ where $\nabla f(x) = 0$. For SAM with perturbation size $\rho$, given the current iterate $x_t$, the corresponding $y_t = x_t + \rho\frac{\nabla f(x_t)}{|\nabla f(x_t)|}$ is given by

$$y_t = x_t + \begin{cases} \rho & \text{if } (-0.3 + 1.2k)\rho < x_t < (0.3 + 1.2k)\rho \text{ for some } k \in \mathbb{Z}, \\ -\rho & \text{if } (0.3 + 1.2k)\rho < x_t < (0.9 + 1.2k)\rho \text{ for some } k \in \mathbb{Z}, \\ 0 & \text{if } x_t \in \mathbb{X}. \end{cases}$$

This leads to the following a virtual gradient map $G_f$:

$$G_f(x_t) = \nabla f(y_t) = \begin{cases} \frac{3\beta\rho}{5\pi}\cos\left(\frac{5\pi}{3\rho}x_t + \frac{5\pi}{3}\right) & \text{if } (-0.3 + 1.2k)\rho < x_t < (0.3 + 1.2k)\rho \text{ for some } k \in \mathbb{Z}, \\ \frac{3\beta\rho}{5\pi}\cos\left(\frac{5\pi}{3\rho}x_t - \frac{5\pi}{3}\right) & \text{if } (0.3 + 1.2k)\rho < x_t < (0.9 + 1.2k)\rho \text{ for some } k \in \mathbb{Z}, \\ 0 & \text{if } x_t \in \mathbb{X}. \end{cases}$$

From this virtual gradient map, we can define

$$\mathbb{Y} = \{x \mid x = (0.7 + 1.2k)\rho, k \in \mathbb{Z}\} \cup \{x \mid x = (-0.1 + 1.2k)\rho, k \in \mathbb{Z}\}$$

and $\mathbb{Y}$ is the set of all points $x$ where $G_f(x) = 0$ and $\nabla f(x) \neq 0$.

Since $f$ is a one-dimensional function, a virtual loss $J_f$ can be obtained by integrating $G_f$. One possible example is

$$J_f(x) = \begin{cases} \frac{9\beta\rho^2}{25\pi^2} \sin\left(\frac{5\pi}{3\rho}x + \frac{5\pi}{3}\right) & \text{if } (-0.3 + 1.2k)\rho \leq x \leq (0.3 + 1.2k)\rho \text{ for some } k \in \mathbb{Z}, \\ \frac{9\beta\rho^2}{25\pi^2} \sin\left(\frac{5\pi}{3\rho}x - \frac{5\pi}{3}\right) & \text{if } (0.3 + 1.2k)\rho \leq x \leq (0.9 + 1.2k)\rho \text{ for some } k \in \mathbb{Z}, \end{cases}$$

which is a piecewise $\beta$-smooth function that is minimized at points in $\mathbb{Y}$ and locally maximized at (non-differentiable) points in $\mathbb{X}$. Since $J_f$ is well-defined, we can view SAM as GD on $J_f$.

For sufficiently small $\eta \leq \frac{1}{\beta}$ and any initialization $x_0 \in \mathbb{R} \setminus \mathbb{X}$, the initialization belongs to one of the intervals $((-0.3 + 0.6k)\rho, (0.3 + 0.6k)\rho)$. For such small enough $\eta$, we can guarantee that the SAM iterates $x_t$ will stay in the interval $((-0.3 + 0.6k)\rho, (0.3 + 0.6k)\rho)$ for all $t \geq 0$. For example, suppose that $x_t \in (-0.3\rho, 0.3\rho)$. Then, the next iterate will stay in the same interval if and only if the image of the interval $(-0.3\rho, 0.3\rho)$ under a map $x \mapsto x - \frac{3\eta\beta\rho}{5\pi} \cos\left(\frac{5\pi}{3\rho}x + \frac{5\pi}{3}\right)$ is a subset of $(-0.3\rho, 0.3\rho)$.

$$\left\{x - \frac{3\eta\beta\rho}{5\pi} \cos\left(\frac{5\pi}{3\rho}x + \frac{5\pi}{3}\right) \mid x \in (-0.3\rho, 0.3\rho)\right\} \subset (-0.3\rho, 0.3\rho)$$

$$\iff \left\{z - \eta\beta \cos\left(z + \frac{5\pi}{3}\right) \mid z \in \left(-\frac{\pi}{2}, \frac{\pi}{2}\right)\right\} \subset \left(-\frac{\pi}{2}, \frac{\pi}{2}\right),$$

and the containment is true if $\eta\beta \leq 1$. By symmetry, the same argument can be applied to any intervals of the form $((-0.3 + 0.6k)\rho, (0.3 + 0.6k)\rho)$.

Thus, for any initialization $x_0 \in \mathbb{R} \setminus \mathbb{X}$, all SAM iterates stay inside the interval which $x_0$ belongs to. Inside the interval, by $\beta$-smoothness of $J_f$, the following descent lemma always holds:

$$J_f(x_{t+1}) \leq J_f(x_t) + \langle \nabla J_f(x_t), x_{t+1} - x_t \rangle + \frac{\beta}{2}\|x_{t+1} - x_t\|^2$$

$$= J_f(x_t) - \eta\left(1 - \frac{\eta\beta}{2}\right)\|G_f(x_t)\|^2$$

$$\leq J_f(x_t) - \frac{\eta}{2}\|G_f(x_t)\|^2.$$

Therefore, if we add the inequalities up for $t = 0, \ldots, T - 1$, we get

$$\sum_{t=0}^{T-1} \|G_f(x_t)\|^2 \leq \frac{2}{\eta}(J_f(x_0) - J_f(x_T)) \leq \frac{2}{\eta}(J_f(x_0) - J_f^*) < \infty,$$

where $J_f^* \triangleq \inf_x J_f(x) > -\infty$. Since the inequality holds for all $T$, the series is summable, which in turn implies that $\|G_f(x_t)\| \to 0$ as $t \to \infty$. However, the only point in the interval $((-0.3 + 0.6k)\rho, (0.3 + 0.6k)\rho)$ satisfying $\|G_f(x)\| = 0$ is the one in $\mathbb{Y} \cap ((-0.3 + 0.6k)\rho, (0.3 + 0.6k)\rho)$. Hence, if $x_0 \in \mathbb{R} \setminus \mathbb{X}$ and $\eta \leq \frac{1}{\beta}$, SAM must converge to a point in $\mathbb{Y}$. Since $x_0$ is drawn from a continuous probability distribution, $x_0 \in \mathbb{R} \setminus \mathbb{X}$ holds almost surely. This finishes the proof. $\square$

### B.7 Non-convergence for a Nonsmooth and Convex Functions (Proof of Theorem 3.6)

For nonsmooth convex function, depending on the initialization, SAM can converge to a suboptimal point with distance $\Omega(\rho)$ from the global minimum. The theorem is restated for convenience.

**Theorem 3.6.** *For any $\rho > 0$ and $\eta < \frac{7\rho}{4}$, there exists a nonsmooth Lipschitz convex function $f$ such that for some initialization, deterministic SAM converges to suboptimal points located at a distance of $\Omega(\rho)$ from the global minimum.*

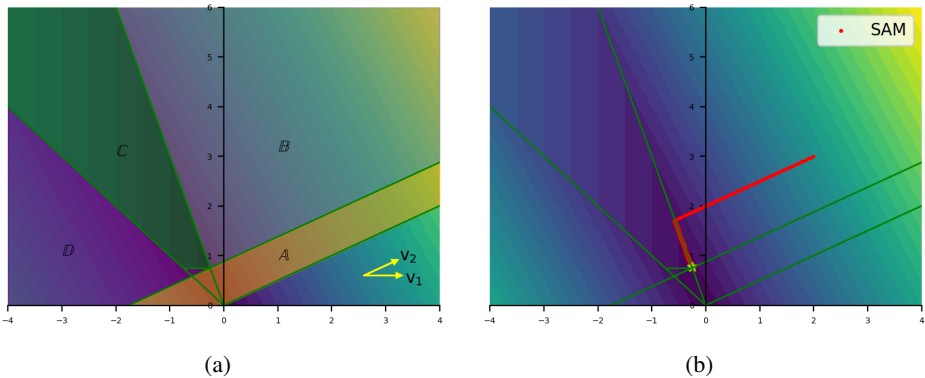

(a)                                            (b)

Figure 3: (a) The demonstration of four seperate regions, and basis vectors of example function. (b) The trajectory of SAM for example function. As SAM enters region $\mathbb{A}$, the update cannot get any closer to the global minima.

*Proof.* For $\boldsymbol{x} = (x^{(1)}, x^{(2)}) \in \mathbb{R}^2$, consider a 2-dimensional function,

$$f(\boldsymbol{x}) = \max\{|x^{(1)}|, |2x^{(1)} + x^{(2)}|\}.$$

A set $\{\boldsymbol{v}_1, \boldsymbol{v}_2\}$ can be established as a basis for $\mathbb{R}^2$, with $\boldsymbol{v}_1 = \boldsymbol{e}_1$, and $\boldsymbol{v}_2 = \frac{2}{\sqrt{5}}\boldsymbol{e}_1 + \frac{1}{\sqrt{5}}\boldsymbol{e}_2$. Any vector $\boldsymbol{x} \in \mathbb{R}^2$ can be uniquely expressed by $\boldsymbol{v}_1$ and $\boldsymbol{v}_2$. Consider the region: $\{\boldsymbol{x} = b^{(1)}\boldsymbol{v}_1 + b^{(2)}\boldsymbol{v}_2 \mid b^{(1)} < 0\}$.

This area can be partitioned into four separate regions.

$$\begin{cases} \mathbb{A} : \left\{\boldsymbol{x} = b^{(1)}\boldsymbol{v}_1 + b^{(2)}\boldsymbol{v}_2 \mid -\frac{7\rho}{2} < b^{(1)} < 0\right\}, \\ \mathbb{B} : \left\{\boldsymbol{x} = b^{(1)}\boldsymbol{v}_1 + b^{(2)}\boldsymbol{v}_2 \mid b^{(1)} < -\frac{7\rho}{2}\right\} \cap \left\{\boldsymbol{x} = b^{(1)}\boldsymbol{v}_1 + b^{(2)}\boldsymbol{v}_2 \mid b^{(1)} + \sqrt{5}b^{(2)} > 0\right\}, \\ \mathbb{C} : \left\{\boldsymbol{x} = b^{(1)}\boldsymbol{v}_1 + b^{(2)}\boldsymbol{v}_2 \mid b^{(1)} + \sqrt{5}b^{(2)} < 0\right\} \cap \left\{\boldsymbol{x} = b^{(1)}\boldsymbol{v}_1 + b^{(2)}\boldsymbol{v}_2 \mid -b^{(1)} + \sqrt{5}b^{(2)} > 0\right\} \\ \qquad \cap \left\{\boldsymbol{x} = b^{(1)}\boldsymbol{v}_1 + b^{(2)}\boldsymbol{v}_2 \mid -2b^{(1)} + \sqrt{5}b^{(2)} > \frac{3}{2}\rho\right\}, \\ \mathbb{D} : \left\{\boldsymbol{x} = b^{(1)}\boldsymbol{v}_1 + b^{(2)}\boldsymbol{v}_2 \mid b^{(1)} < 0\right\} - (\mathbb{A} \cup \mathbb{B} \cup \mathbb{C}). \end{cases}$$

Figure 3(a) demonstrates the regions $\mathbb{A}, \mathbb{B}, \mathbb{C}, \mathbb{D}$, as well as the vectors $\boldsymbol{v}_1$ and $\boldsymbol{v}_2$. When $\boldsymbol{x}$ belongs to the set $\boldsymbol{x} = b^{(1)}\boldsymbol{v}_1 + b^{(2)}\boldsymbol{v}_2 \mid b^{(1)} < 0$, we can examine four different scenarios.

**Case A:** $\boldsymbol{x} = b^{(1)}\boldsymbol{v}_1 + b^{(2)}\boldsymbol{v}_2 \in \mathbb{A}$. $\boldsymbol{x} \in \mathbb{A}$, so $\boldsymbol{y} = \boldsymbol{x} + \rho\frac{\nabla f(\boldsymbol{x})}{\|\nabla f(\boldsymbol{x})\|}$ and $-\nabla f(\boldsymbol{y})$ are as follows.

$$\boldsymbol{y} = \begin{cases} \boldsymbol{x} + \rho\boldsymbol{v}_2, & -b^{(1)} < \sqrt{5}b^{(2)} \\ \boldsymbol{x} - \rho\boldsymbol{v}_1, & b^{(1)} < \sqrt{5}b^{(2)} < -b^{(1)} \\ \boldsymbol{x} - \rho\boldsymbol{v}_2, & \sqrt{5}b^{(2)} < b^{(1)}. \end{cases}$$

$$-\nabla f(\boldsymbol{y}) = \begin{cases} -\sqrt{5}\boldsymbol{v}_2, & -b^{(1)} < \sqrt{5}b^{(2)} \\ \sqrt{5}\boldsymbol{v}_2, & b^{(1)} < \sqrt{5}b^{(2)} < -b^{(1)} \\ \sqrt{5}\boldsymbol{v}_2, & \sqrt{5}b^{(2)} < b^{(1)}. \end{cases}$$

As a result, the SAM updates $\nabla f(\boldsymbol{y})$ only affect $\boldsymbol{v}_2$, and does not affect on $\boldsymbol{v}_1$. Thus, if $\boldsymbol{x}_t \in \mathbb{A}$, the next SAM iterate $\boldsymbol{x}_{t+1}$ remains in $\mathbb{A}$.

**Case B:** $\boldsymbol{x} = b^{(1)}\boldsymbol{v}_1 + b^{(2)}\boldsymbol{v}_2 \in \mathbb{B}$. We can verify that $\boldsymbol{y} = \boldsymbol{x} + \rho\frac{\nabla f(\boldsymbol{x})}{\|\nabla f(\boldsymbol{x})\|} = \boldsymbol{x} + \rho\boldsymbol{v}_2$. Therefore, $-\nabla f(\boldsymbol{y}) = -\sqrt{5}\boldsymbol{v}_2$. Therefore, the SAM update shifts in the direction of $-\boldsymbol{v}_2$. Hence, if $\boldsymbol{x}_t \in \mathbb{B}$, the next SAM iterate will be $\boldsymbol{x}_{t+1} \in (\mathbb{B} \cup \mathbb{C} \cup \mathbb{D})$.

**Case C: $x = b^{(1)}v_1 + b^{(2)}v_2 \in \mathbb{C}$.** $y = x + \rho\frac{\nabla f(x)}{\|\nabla f(x)\|} = x - \rho v_1$, so

$$-\nabla f(y_t) = \begin{cases} \sqrt{5}v_2, & b^{(1)} < \sqrt{5}b^{(2)} < b^{(1)} + \rho \\ v_1, & b^{(1)} + \rho < \sqrt{5}b^{(2)} < -b^{(1)}. \end{cases}$$

For $x_t = b_t^{(1)}v_1 + b_t^{(2)}v_2 \in \mathbb{C}$, if $b_t^{(1)} < \sqrt{5}b_t^{(2)} < b_t^{(1)} + \rho$, then SAM update shifts in the direction of $v_2$, resulting in $x_{t+1} \in (\mathbb{B} \cup \mathbb{C} \cup \mathbb{D})$. Otherwise, the next SAM iterate $x_{t+1}$ shifts in the direction of $+v_1$. Consequently, $x_{t+1}$ can either remain in $\mathbb{B} \cup \mathbb{C} \cup \mathbb{D}$, or move to $\mathbb{A}$.

Assume that $x_{t+1}$ moves to $\mathbb{A}$. Given that $b_t^{(1)} < -\frac{7\rho}{2}$ for $x_t \in \mathbb{C}$, we can verify that $b_{t+1}^{(1)} < -\frac{7\rho}{2} + \eta$.Based on our previous observation in **Case A** where SAM updates only affect $v_2$ when $x \in \mathbb{A}$, we can conclude that for all subsequent iterates $x_i$ with $i > t+1$, the condition $b_i^{(1)} < -\frac{7\rho}{2} + \eta$ continues to hold.

**Case D: $x = b^{(1)}v_1 + b^{(2)}v_2 \in \mathbb{D}$.** $y = x + \rho\frac{\nabla f(x)}{\|\nabla f(x)\|}$ becomes

$$y = \begin{cases} x - \rho v_2, & \sqrt{5}b^{(2)} < b^{(1)} \\ x - \rho v_1, & \text{otherwise.} \end{cases}$$

We can check that $-\nabla f(y) = \sqrt{5}v_2$ for all cases. So the SAM update shifts in the direction of $+v_2$. As a result, if $x_t \in \mathbb{D}$, the next SAM iterate $x_{t+1}$ will fall into $x_{t+1} \in (\mathbb{B} \cup \mathbb{C} \cup \mathbb{D})$.

Furthermore, we can verify that for $x \in (\mathbb{B} \cup \mathbb{C} \cup \mathbb{D})$, $b^{(1)} < -\frac{7\rho}{2}$ holds. Therefore, summing up all the cases, we can conclude that if the initial iterate $x_0$ is chosen from $x_0 \in (\mathbb{B} \cup \mathbb{C} \cup \mathbb{D})$, then all subsequent SAM iterates $x_t$ will satisfy $b_t^{(1)} < -\frac{7\rho}{2} + \eta$. Since the global minimum is located at $x^* = (0, 0)$, it follows that $\|x_t - x^*\| > \frac{|7\rho/2 - \eta|}{\sqrt{5}}$ for every $t > 0$. Thus, for $\eta < \frac{7\rho}{4}$, we can ascertain that in this particular function, $\|x_t - x^*\| > \frac{7\rho}{4\sqrt{5}}$ for every $t > 0$, thereby proving that the distance from the global minimum is at least $\Omega(\rho)$. The trajectory plot of SAM on this example function is illustrated in Figure 3(b). $\qquad\square$

## C   Proofs for (Non-)Convergence of Stochastic SAM

In this section, we provide in-depth demonstrations and proofs regarding convergence of stochastic SAM with constant perturbation size $\rho$. The objective is defined as $f(\boldsymbol{x}) = \mathbb{E}_\xi[l(\boldsymbol{x}; \xi)]$, where $\xi$ represents a stochastic parameter (e.g., data sample) and $l(\boldsymbol{x}; \xi)$ represents the loss at point $\boldsymbol{x}$ with a random sample $\xi$. The update iteration for stochastic SAM is specified as follows:

$$\begin{cases} \boldsymbol{y}_t = \boldsymbol{x}_t + \rho \frac{g(\boldsymbol{x}_t)}{\|g(\boldsymbol{x}_t)\|}, \\ \boldsymbol{x}_{t+1} = \boldsymbol{x}_t - \eta \tilde{g}(\boldsymbol{y}_t). \end{cases}$$

We define $g(\boldsymbol{x}) = \nabla_{\boldsymbol{x}} l(\boldsymbol{x}; \xi)$ and $\tilde{g}(\boldsymbol{x}) = \nabla_{\boldsymbol{x}} l(\boldsymbol{x}; \tilde{\xi})$, where $\xi$ and $\tilde{\xi}$ are stochastic parameters. Stochastic SAM comes in two variations: $n$-SAM, where $\xi$ and $\tilde{\xi}$ are independent, and $m$-SAM, where $\xi$ is equal to $\tilde{\xi}$.

### C.1   Important Lemmas Regarding Stochastic SAM

In this section, we present a number of lemmas that are utilized in our theorem proofs regarding stochastic SAM. In order to do this, we also introduce extra notation, $\hat{\boldsymbol{y}}_t = \boldsymbol{x}_t + \rho \frac{\nabla f(\boldsymbol{x}_t)}{\|\nabla f(\boldsymbol{x}_t)\|}$, as a deterministically ascended parameter.

**Lemma C.1.** *For a differentiable and $\mu$-strongly convex function $f$, we have*

$$\mathbb{E}\langle \nabla f(\boldsymbol{x}_t), \tilde{g}(\boldsymbol{y}_t) - \nabla f(\boldsymbol{x}_t) \rangle \geq \mathbb{E}\langle \nabla f(\boldsymbol{x}_t), \tilde{g}(\boldsymbol{y}_t) - \tilde{g}(\hat{\boldsymbol{y}}_t) \rangle + \mu\rho\mathbb{E}\|\nabla f(\boldsymbol{x}_t)\|.$$

*For a differentiable and convex function $f$, the inequality continues to hold with $\mu = 0$.*

*Proof.*

$$\begin{aligned} \mathbb{E}\langle \nabla f(\boldsymbol{x}_t), \tilde{g}(\boldsymbol{y}_t) - \nabla f(\boldsymbol{x}_t) \rangle &= \mathbb{E}\langle \nabla f(\boldsymbol{x}_t), \tilde{g}(\boldsymbol{y}_t) - \tilde{g}(\hat{\boldsymbol{y}}_t) \rangle + \mathbb{E}\langle \nabla f(\boldsymbol{x}_t), \tilde{g}(\hat{\boldsymbol{y}}_t) - \nabla f(\boldsymbol{x}_t) \rangle \\ &= \mathbb{E}\langle \nabla f(\boldsymbol{x}_t), \tilde{g}(\boldsymbol{y}_t) - \tilde{g}(\hat{\boldsymbol{y}}_t) \rangle + \mathbb{E}\langle \nabla f(\boldsymbol{x}_t), \nabla f(\hat{\boldsymbol{y}}_t) - \nabla f(\boldsymbol{x}_t) \rangle \\ &\geq \mathbb{E}\langle \nabla f(\boldsymbol{x}_t), \tilde{g}(\boldsymbol{y}_t) - \tilde{g}(\hat{\boldsymbol{y}}_t) \rangle + \mu\rho\mathbb{E}\|\nabla f(\boldsymbol{x}_t)\|, \end{aligned}$$

where we use Lemma B.2 in the last inequality, thereby completing the proof. $\qquad\square$

**Lemma C.2.** *Under $n$-SAM, for a $\beta$-smooth and $\mu$-strongly convex function $f$, we have*

$$\mathbb{E}\langle \nabla f(\boldsymbol{x}_t), \tilde{g}(\boldsymbol{y}_t) - \nabla f(\boldsymbol{x}_t) \rangle \geq -\frac{1}{2}\mathbb{E}\|\nabla f(\boldsymbol{x}_t)\|^2 + \mu\rho\mathbb{E}\|\nabla f(\boldsymbol{x}_t)\| - 2\beta^2\rho^2.$$

*Under $m$-SAM, additionally assuming $l(\cdot, \xi)$ is $\beta$-smooth for any $\xi$, the inequality continues to hold.*

*Proof.* First we consider $n$-SAM. Starting from Lemma C.1, we have

$$\begin{aligned} \mathbb{E}\langle \nabla f(\boldsymbol{x}_t), \tilde{g}(\boldsymbol{y}_t) - \nabla f(\boldsymbol{x}_t) \rangle &\geq \mathbb{E}\langle \nabla f(\boldsymbol{x}_t), \tilde{g}(\boldsymbol{y}_t) - \tilde{g}(\hat{\boldsymbol{y}}_t) \rangle + \mu\rho\mathbb{E}\|\nabla f(\boldsymbol{x}_t)\| \\ &= \mathbb{E}\langle \nabla f(\boldsymbol{x}_t), \nabla f(\boldsymbol{y}_t) - \nabla f(\hat{\boldsymbol{y}}_t) \rangle + \mu\rho\mathbb{E}\|\nabla f(\boldsymbol{x}_t)\| \\ &\geq -\frac{1}{2}\mathbb{E}\|\nabla f(\boldsymbol{x}_t)\|^2 - \frac{1}{2}\mathbb{E}\|\nabla f(\boldsymbol{y}_t) - \nabla f(\hat{\boldsymbol{y}}_t)\|^2 + \mu\rho\mathbb{E}\|\nabla f(\boldsymbol{x}_t)\| \\ &\geq -\frac{1}{2}\mathbb{E}\|\nabla f(\boldsymbol{x}_t)\|^2 - \frac{\beta^2}{2}\mathbb{E}\|\boldsymbol{y}_t - \hat{\boldsymbol{y}}_t\|^2 + \mu\rho\mathbb{E}\|\nabla f(\boldsymbol{x}_t)\| \\ &= -\frac{1}{2}\mathbb{E}\|\nabla f(\boldsymbol{x}_t)\|^2 - \frac{\beta^2}{2}\mathbb{E}\left\|\rho\frac{g(\boldsymbol{x}_t)}{\|g(\boldsymbol{x}_t)\|} - \rho\frac{\nabla f(\boldsymbol{x}_t)}{\|\nabla f(\boldsymbol{x}_t)\|}\right\|^2 \\ &\quad + \mu\rho\mathbb{E}\|\nabla f(\boldsymbol{x}_t)\| \\ &\geq -\frac{1}{2}\mathbb{E}\|\nabla f(\boldsymbol{x}_t)\|^2 + \mu\rho\mathbb{E}\|\nabla f(\boldsymbol{x}_t)\| - 2\beta^2\rho^2. \end{aligned}$$

Next we consider $m$-SAM. additionally assuming $l(\cdot, \xi)$ is $\beta$-smooth for any $\xi$, starting from Lemma C.1, we have

$$\mathbb{E}\langle \nabla f(\boldsymbol{x}_t), \tilde{g}(\boldsymbol{y}_t) - \nabla f(\boldsymbol{x}_t) \rangle \geq \mathbb{E}\langle \nabla f(\boldsymbol{x}_t), \tilde{g}(\boldsymbol{y}_t) - \tilde{g}(\hat{\boldsymbol{y}}_t) \rangle + \mu\rho\|\nabla f(\boldsymbol{x}_t)\|$$

$$\geq -\frac{1}{2}\mathbb{E}\|\nabla f(\boldsymbol{x}_t)\|^2 - \frac{1}{2}\mathbb{E}\|\tilde{g}(\boldsymbol{y}_t) - \tilde{g}(\hat{\boldsymbol{y}}_t)\|^2 + \mu\rho\mathbb{E}\|\nabla f(\boldsymbol{x}_t)\|$$

$$\geq -\frac{1}{2}\mathbb{E}\|\nabla f(\boldsymbol{x}_t)\|^2 - \frac{\beta^2}{2}\mathbb{E}\|\boldsymbol{y}_t - \hat{\boldsymbol{y}}_t\|^2 + \mu\rho\mathbb{E}\|\nabla f(\boldsymbol{x}_t)\|$$

$$= -\frac{1}{2}\mathbb{E}\|\nabla f(\boldsymbol{x}_t)\|^2 - \frac{\beta^2}{2}\mathbb{E}\left\|\rho\frac{g(\boldsymbol{x}_t)}{\|g(\boldsymbol{x}_t)\|} - \rho\frac{\nabla f(\boldsymbol{x}_t)}{\|\nabla f(\boldsymbol{x}_t)\|}\right\|^2$$
$$+ \mu\rho\mathbb{E}\|\nabla f(\boldsymbol{x}_t)\|$$

$$\geq -\frac{1}{2}\mathbb{E}\|\nabla f(\boldsymbol{x}_t)\|^2 + \mu\rho\mathbb{E}\|\nabla f(\boldsymbol{x}_t)\| - 2\beta^2\rho^2,$$

completing the proof. $\qquad\square$

**Lemma C.3.** *Consider a $\beta$-smooth, $\mu$-strongly convex function $f$, and assume Assumption 2.5. Under $n$-SAM, with step size $\eta \leq \frac{1}{2\beta}$, we have*

$$\mathbb{E}f(\boldsymbol{x}_{t+1}) \leq \mathbb{E}f(\boldsymbol{x}_t) - \frac{\eta}{2}\mathbb{E}\|\nabla f(\boldsymbol{x}_t)\|^2 - \frac{\eta\mu\rho}{2}\mathbb{E}\|\nabla f(\boldsymbol{x}_t)\| + 2\eta\beta^2\rho^2 - \eta^2\beta(\beta^2\rho^2 - \sigma^2).$$

*Under $m$-SAM, additionally assuming $l(\cdot, \xi)$ is $\beta$-smooth for any $\xi$, the inequality continues to hold.*

*Proof.* Starting from the definition of $\beta$-smoothness, we have

$$\mathbb{E}f(\boldsymbol{x}_{t+1}) \leq \mathbb{E}f(\boldsymbol{x}_t) - \eta\mathbb{E}\langle\nabla f(\boldsymbol{x}_t), \tilde{g}(\boldsymbol{y}_t)\rangle + \frac{\eta^2\beta}{2}\mathbb{E}\|\tilde{g}(\boldsymbol{y}_t)\|^2$$

$$= \mathbb{E}f(\boldsymbol{x}_t) - \eta\mathbb{E}\langle\nabla f(\boldsymbol{x}_t), \tilde{g}(\boldsymbol{y}_t)\rangle + \frac{\eta^2\beta}{2}\mathbb{E}\|\tilde{g}(\boldsymbol{y}_t) - \nabla f(\boldsymbol{x}_t)\|^2 + \frac{\eta^2\beta}{2}\mathbb{E}\|\nabla f(\boldsymbol{x}_t)\|^2$$
$$+ \eta^2\beta\mathbb{E}\langle\nabla f(\boldsymbol{x}_t), \tilde{g}(\boldsymbol{y}_t) - \nabla f(\boldsymbol{x}_t)\rangle$$

$$= \mathbb{E}f(\boldsymbol{x}_t) - \eta\mathbb{E}\langle\nabla f(\boldsymbol{x}_t), \tilde{g}(\boldsymbol{y}_t) - \nabla f(\boldsymbol{x}_t)\rangle - \eta\mathbb{E}\|\nabla f(\boldsymbol{x}_t)\|^2$$
$$+ \frac{\eta^2\beta}{2}\mathbb{E}\|\tilde{g}(\boldsymbol{y}_t) - \nabla f(\boldsymbol{x}_t)\|^2 + \frac{\eta^2\beta}{2}\mathbb{E}\|\nabla f(\boldsymbol{x}_t)\|^2$$
$$+ \eta^2\beta\mathbb{E}\langle\nabla f(\boldsymbol{x}_t), \tilde{g}(\boldsymbol{y}_t) - \nabla f(\boldsymbol{x}_t)\rangle$$

$$\leq \mathbb{E}f(\boldsymbol{x}_t) - \eta(1 - \eta\beta)\mathbb{E}\langle\nabla f(\boldsymbol{x}_t), \tilde{g}(\boldsymbol{y}_t) - \nabla f(\boldsymbol{x}_t)\rangle - \eta\left(1 - \frac{\eta\beta}{2}\right)\mathbb{E}\|\nabla f(\boldsymbol{x}_t)\|^2$$
$$+ \eta^2\beta\mathbb{E}\|\tilde{g}(\boldsymbol{y}_t) - \nabla f(\boldsymbol{y}_t)\|^2 + \eta^2\beta\mathbb{E}\|\nabla f(\boldsymbol{y}_t) - \nabla f(\boldsymbol{x}_t)\|^2$$

$$\leq \mathbb{E}f(\boldsymbol{x}_t) - \eta(1 - \eta\beta)\mathbb{E}\langle\nabla f(\boldsymbol{x}_t), \tilde{g}(\boldsymbol{y}_t) - \nabla f(\boldsymbol{x}_t)\rangle - \eta\left(1 - \frac{\eta\beta}{2}\right)\mathbb{E}\|\nabla f(\boldsymbol{x}_t)\|^2$$
$$+ \eta^2\beta(\sigma^2 + \beta^2\rho^2).$$

Since we assumed $\eta \leq \frac{1}{2\beta}$, $\eta(1 - \eta\beta) \geq \frac{\eta}{2}$ hold. Applying Lemma C.2, we get

$$\mathbb{E}f(\boldsymbol{x}_{t+1}) \leq \mathbb{E}f(\boldsymbol{x}_t) - \eta(1 - \eta\beta)\left(-\frac{1}{2}\mathbb{E}\|\nabla f(\boldsymbol{x}_t)\|^2 + \mu\rho\mathbb{E}\|\nabla f(\boldsymbol{x}_t)\| - 2\beta^2\rho^2\right)$$

$$- \eta\left(1 - \frac{\eta\beta}{2}\right)\mathbb{E}\|\nabla f(\boldsymbol{x}_t)\|^2 + \eta^2\beta(\sigma^2 + \beta^2\rho^2)$$

$$= \mathbb{E}f(\boldsymbol{x}_t) - \frac{\eta}{2}\mathbb{E}\|\nabla f(\boldsymbol{x}_t)\|^2 - \eta(1 - \eta\beta)\mu\rho\mathbb{E}\|\nabla f(\boldsymbol{x}_t)\| + 2\eta\beta^2\rho^2 - \eta^2\beta(\beta^2\rho^2 - \sigma^2)$$

$$\leq \mathbb{E}f(\boldsymbol{x}_t) - \frac{\eta}{2}\mathbb{E}\|\nabla f(\boldsymbol{x}_t)\|^2 - \frac{\eta\mu\rho}{2}\mathbb{E}\|\nabla f(\boldsymbol{x}_t)\| + 2\eta\beta^2\rho^2 - \eta^2\beta(\beta^2\rho^2 - \sigma^2),$$

completing the proof. $\qquad\square$

## C.2  Convergence Proof for Smooth and Strongly Convex Functions (Proof of Theorem 4.1)

In this section, we demonstrate the convergence result of stochastic SAM for smooth and strongly convex functions. For convenience, we restate the theorem here.

**Theorem 4.1.** *Consider a $\beta$-smooth, $\mu$-strongly convex function $f$, and assume Assumption 2.5. Under $n$-SAM, starting at $x_0$ with any perturbation size $\rho > 0$ and step size $\eta = \min\left\{\frac{1}{\mu T} \cdot \max\left\{1, \log\left(\frac{\mu^2 \Delta T}{\beta[\sigma^2 - \beta^2\rho^2]_+}\right)\right\}, \frac{1}{2\beta}\right\}$ to minimize $f$, we have*

$$\mathbb{E}f(\boldsymbol{x}_T) - f^* \leq \tilde{\mathcal{O}}\left(\exp\left(-\frac{\mu T}{2\beta}\right)\Delta + \frac{\beta[\sigma^2 - \beta^2\rho^2]_+}{\mu^2 T}\right) + \frac{2\beta^2\rho^2}{\mu}.$$

*Under $m$-SAM, additionally assuming $l(\cdot, \xi)$ is $\beta$-smooth for any $\xi$, the inequality continues to hold.*

*Proof.* We start the proof from Lemma C.3; in order to apply the lemma, additionally assuming $\beta$-smoothness for component functions $l(\cdot, \xi)$ is necessary for $m$-SAM.

$$\mathbb{E}f(\boldsymbol{x}_{t+1}) \leq \mathbb{E}f(\boldsymbol{x}_t) - \frac{\eta}{2}\mathbb{E}\|\nabla f(\boldsymbol{x}_t)\|^2 - \frac{\eta\mu\rho}{2}\mathbb{E}\|\nabla f(\boldsymbol{x}_t)\| + 2\eta\beta^2\rho^2 - \eta^2\beta(\beta^2\rho^2 - \sigma^2).$$

Since $\mu$-strongly convex functions satisfy $\mu$-PL inequality (B.1), we get

$$\mathbb{E}f(\boldsymbol{x}_{t+1}) - f^* \leq (1 - \eta\mu)(\mathbb{E}f(\boldsymbol{x}_t) - f^*) - \frac{\eta\mu\rho}{2}\mathbb{E}\|\nabla f(\boldsymbol{x}_t)\| + 2\eta\beta^2\rho^2 - \eta^2\beta(\beta^2\rho^2 - \sigma^2).$$

Depending on the value of $\sigma$, there are two cases in which the convergence rate varies.

**Case A: $\sigma \leq \beta\rho$.** In this case, we have

$$\mathbb{E}f(\boldsymbol{x}_{t+1}) - f^* \leq (1 - \eta\mu)(\mathbb{E}f(\boldsymbol{x}_t) - f^*) - \frac{\eta\mu\rho}{2}\mathbb{E}\|\nabla f(\boldsymbol{x}_t)\| + 2\eta\beta^2\rho^2$$

$$\leq (1 - \eta\mu)(\mathbb{E}f(\boldsymbol{x}_t) - f^*) + 2\eta\beta^2\rho^2,$$

and our choice of $\eta$ must be $\frac{1}{2\beta}$. Unrolling the inequality and substituting $\eta = \frac{1}{2\beta}$ draws out

$$\mathbb{E}f(\boldsymbol{x}_T) - f^* \leq (1 - \eta\mu)^T(\mathbb{E}f(\boldsymbol{x}_0) - f^*) + \frac{2\beta^2\rho^2}{\mu}$$

$$= \left(1 - \frac{\mu}{2\beta}\right)^T \Delta + \frac{2\beta^2\rho^2}{\mu}$$

$$\leq \exp\left(-\frac{\mu T}{2\beta}\right)\Delta + \frac{2\beta^2\rho^2}{\mu}.$$

**Case B: $\sigma > \beta\rho$.** In this case, we have

$$\mathbb{E}f(\boldsymbol{x}_{t+1}) - f^* \leq (1 - \eta\mu)(\mathbb{E}f(\boldsymbol{x}_t) - f^*) - \frac{\eta\mu\rho}{2}\mathbb{E}\|\nabla f(\boldsymbol{x}_t)\| + 2\eta\beta^2\rho^2 + \eta^2\beta(\sigma^2 - \beta^2\rho^2)$$

$$\leq (1 - \eta\mu)(\mathbb{E}f(\boldsymbol{x}_t) - f^*) + 2\eta\beta^2\rho^2 + \eta^2\beta(\sigma^2 - \beta^2\rho^2).$$

Again, unrolling the inequality draws out

$$\mathbb{E}f(\boldsymbol{x}_T) - f^* \leq (1 - \eta\mu)^T\Delta + \frac{2\beta^2\rho^2}{\mu} + \frac{\eta\beta(\sigma^2 - \beta^2\rho^2)}{\mu}. \tag{10}$$

Similar to Section B.2, substituting $\eta = \min\left\{\frac{1}{\mu T} \cdot \max\left\{1, \log\left(\frac{\mu^2 \Delta T}{\beta(\sigma^2 - \beta^2\rho^2)}\right)\right\}, \frac{1}{2\beta}\right\}$ can result in four cases.

**Case B-1:** $\log\left(\frac{\mu^2 \Delta T}{\beta(\sigma^2 - \beta^2\rho^2)}\right) \geq 1$, **and** $\frac{1}{2\beta} \geq \frac{1}{\mu T}\log\left(\frac{\mu^2 \Delta T}{\beta(\sigma^2 - \beta^2\rho^2)}\right)$. Setting $\eta = \frac{1}{\mu T}\log\left(\frac{\mu^2 \Delta T}{\beta(\sigma^2 - \beta^2\rho^2)}\right)$,

$$\mathbb{E}f(\boldsymbol{x}_T) - f^* \leq \frac{\beta(\sigma^2 - \beta^2\rho^2)}{\mu^2 T} + \frac{\beta(\sigma^2 - \beta^2\rho^2)}{\mu^2 T} \cdot \log\left(\frac{\mu^2 \Delta T}{\beta(\sigma^2 - \beta^2\rho^2)}\right) + \frac{2\beta^2\rho^2}{\mu}.$$

**Case B-2:** $\log\left(\frac{\mu^2 \Delta T}{\beta(\sigma^2 - \beta^2\rho^2)}\right) \geq 1$, **and** $\frac{1}{2\beta} \leq \frac{1}{\mu T}\log\left(\frac{\mu^2 \Delta T}{\beta(\sigma^2 - \beta^2\rho^2)}\right)$. Setting $\eta = \frac{1}{2\beta}$,

$$\mathbb{E}f(\boldsymbol{x}_T) - f^* \leq \exp\left(-\frac{\mu T}{2\beta}\right)\Delta + \frac{\beta(\sigma^2 - \beta^2\rho^2)}{\mu} \cdot \frac{1}{2\beta} + \frac{2\beta^2\rho^2}{\mu}$$

$$\leq \exp\left(-\frac{\mu T}{2\beta}\right)\Delta + \frac{\beta(\sigma^2 - \beta^2\rho^2)}{\mu^2 T} \cdot \log\left(\frac{\mu^2 \Delta T}{\beta(\sigma^2 - \beta^2\rho^2)}\right) + \frac{2\beta^2\rho^2}{\mu}.$$

**Case B-3:** $\log\left(\frac{\mu^2\Delta T}{\beta(\sigma^2-\beta^2\rho^2)}\right) \leq 1$**, and** $\frac{1}{2\beta} \geq \frac{1}{\mu T}$**.** Setting $\eta = \frac{1}{\mu T}$,

$$
\begin{aligned}
\mathbb{E}f(\boldsymbol{x}_T) - f^* &\leq \left(1-\frac{1}{T}\right)^T \Delta + \frac{\beta(\sigma^2-\beta^2\rho^2)}{\mu^2 T} + \frac{2\beta^2\rho^2}{\mu} \\
&\leq \left(1-\frac{1}{T}\log\left(\frac{\mu^2\Delta T}{\beta(\sigma^2-\beta^2\rho^2)}\right)\right)^T \Delta + \frac{\beta(\sigma^2-\beta^2\rho^2)}{\mu^2 T} + \frac{2\beta^2\rho^2}{\mu} \\
&\leq \frac{\beta(\sigma^2-\beta^2\rho^2)}{\mu^2 T} + \frac{\beta(\sigma^2-\beta^2\rho^2)}{\mu^2 T} + \frac{2\beta^2\rho^2}{\mu}.
\end{aligned}
$$

**Case B-4:** $\log\left(\frac{\mu^2\Delta T}{\beta(\sigma^2-\beta^2\rho^2)}\right) \leq 1$**, and** $\frac{1}{2\beta} \leq \frac{1}{\mu T}$**.** Setting $\eta = \frac{1}{2\beta}$,

$$
\begin{aligned}
\mathbb{E}f(\boldsymbol{x}_T) - f^* &\leq \exp\left(-\frac{\mu T}{2\beta}\right)\Delta + \frac{\beta(\sigma^2-\beta^2\rho^2)}{\mu}\cdot\frac{1}{2\beta} + \frac{2\beta^2\rho^2}{\mu} \\
&\leq \exp\left(-\frac{\mu T}{2\beta}\right)\Delta + \frac{\beta(\sigma^2-\beta^2\rho^2)}{\mu^2 T} + \frac{2\beta^2\rho^2}{\mu}.
\end{aligned}
$$

Merging all four cases, we get

$$
\mathbb{E}f(\boldsymbol{x}_T) - f^* = \tilde{\mathcal{O}}\left(\exp\left(-\frac{\mu T}{2\beta}\right)\Delta + \frac{\beta(\sigma^2-\beta^2\rho^2)}{\mu^2 T}\right) + \frac{2\beta^2\rho^2}{\mu},
$$

thereby finishing the proof. $\qquad\square$

### C.3 Non-Convergence of $m$-SAM for a Smooth and Strongly Convex Function (Proof of Theorem 4.2)

In this section, we present a formal analysis of our counterexample, which shows that stochastic $m$-SAM with constant perturbation size $\rho$ may fail to fully converge to a global minimum, and can get close to the global minimum by only $\Omega(\rho)$. This counterexample indicates that the $\mathcal{O}(\rho^2)$ term in Theorem 4.1 is unavoidable. For readers' convenience, we restate the theorem.

**Theorem 4.2.** *For any $\rho > 0, \beta > 0, \sigma > \beta\rho$ and $\eta \leq \frac{3}{10\beta}$, there exists a $\beta$-smooth and $\frac{\beta}{5}$-strongly convex function $f$ satisfying the following. (1) The function $f$ satisfies Assumption 2.5. (2) The component functions $l(\cdot;\xi)$ of $f$ are $\beta$-smooth for any $\xi$. (3) If we run $m$-SAM on $f$ initialized inside a certain interval, then any arbitrary weighted average $\bar{\boldsymbol{x}}$ of the iterates $\boldsymbol{x}_0, \boldsymbol{x}_1, \ldots$ must satisfy $\mathbb{E}[f(\bar{\boldsymbol{x}}) - f^*] \geq \Omega(\rho^2)$.*

*Proof.* Given $\rho > 0$, $\beta > 0$, $\sigma \geq 0$, and $\eta \leq \frac{3}{10\beta}$, we choose $p \triangleq \frac{2}{3}$, $c \triangleq \left(1+\frac{p}{4}\right)\rho$. For $x \in \mathbb{R}$, and a constant $a > 0$ which will be chosen later. Using these $p$, $c$, and $a$, we construct the counterexample. We consider a one-dimensional smooth strongly convex function

$$
f(x) = \frac{a}{2}x^2,
$$

and the stochastic function $l(x;\xi)$ can be given as

$$
l(x;\xi) = \begin{cases} f^{(1)}(x), & \text{with probability } p \\ f^{(2)}(x), & \text{otherwise,} \end{cases}
$$

where each component functions are as described below:

$$
f^{(1)}(x) = \begin{cases} \frac{a}{2}(x-c+2\rho)^2 - a\rho^2, & x \leq c-\rho \\ -\frac{a}{2}(x-c)^2, & c-\rho \leq x \leq c+\rho \\ \frac{a}{2}(x-c-2\rho)^2 - a\rho^2, & c+\rho \leq x. \end{cases}
$$

$$
f^{(2)}(x) = \begin{cases} \frac{1}{1-p}\left(\frac{a}{2}x^2 - \frac{pa}{2}(x-c+2\rho)^2 + pa\rho^2\right), & x \leq c-\rho \\ \frac{1}{1-p}\left(\frac{a}{2}x^2 + \frac{pa}{2}(x-c)^2\right), & c-\rho \leq x \leq c+\rho \\ \frac{1}{1-p}\left(\frac{a}{2}x^2 - \frac{pa}{2}(x-c-2\rho)^2 + pa\rho^2\right), & c+\rho \leq x. \end{cases}
$$

First, it is easy to verify that $\mathbb{E}l(x;\xi) = f(x)$. We can also confirm that $f$, $f^{(1)}$ are $a$-smooth, and $f^{(2)}$ is $\left(\frac{1+p}{1-p}a\right)$-smooth. Furthermore, $\|\nabla f(x) - \nabla f^{(1)}(x)\|^2 \leq a^2(2\rho + c)^2$ holds, along with $\|\nabla f(x) - \nabla f^{(2)}(x)\|^2 \leq \left(\frac{pa}{1-p}\right)^2 (2\rho + c)^2$. Consequently, we have $\mathbb{E}\|\nabla f(x) - \nabla l(x;\xi)\|^2 \leq \frac{pa^2}{1-p}(2\rho + c)^2 = 2a^2 \cdot \left(\frac{19\rho}{6}\right)^2$, where we used $p = \frac{2}{3}$ and $c = (1 + \frac{p}{4})\rho$.

Now choose $a = \min\left\{\frac{\beta}{5}, \frac{\sigma}{5\rho}\right\} \leq \frac{\beta}{5}$. This choice ensures that $f$, $f^{(1)}$, and $f^{(2)}$ are all $\beta$-smooth, as desired. Also notice that $2a^2 \cdot \left(\frac{19\rho}{6}\right)^2 \leq 2 \cdot \left(\frac{19}{5\cdot 6}\right)^2 \sigma^2 < \sigma^2$, so the function satisfies Assumption 2.5.

For the remaining of the proof, we investigate the virtual gradient maps $G_{f^{(1)}}$ and $G_{f^{(2)}}$ within the specified region of interest: $c - \rho \leq x \leq c + \rho$. We will then demonstrate that if we initialize $m$-SAM in this interval, all subsequent iterations of $m$-SAM will remain inside $[c - \rho, c + \rho]$, as required above.

Within this interval, the perturbed iterate $y = \rho\frac{\nabla f^{(1)}(x)}{\|\nabla f^{(1)}(x)\|}$ and the virtual gradient map $G_{f^{(1)}}$ of $f^{(1)}$ can be described as follows.

$$y = \begin{cases} x + \rho, & x < c \\ x, & x = c \\ x - \rho, & x > c, \end{cases} \qquad G_{f^{(1)}}(x) = \begin{cases} -a(x + \rho - c), & x < c \\ 0, & x = c \\ -a(x - \rho - c), & x > c. \end{cases}$$

Additionally defining $c' \triangleq \frac{p}{1+p}c$, we now calculate $y = \rho\frac{\nabla f^{(2)}(x)}{\|\nabla f^{(2)}(x)\|}$ and the virtual gradient map $G_{f^{(2)}}$ of $f^{(2)}$.

$$y = \begin{cases} x - \rho, & x < c', \\ x, & x = c', \\ x + \rho, & x > c' \end{cases} \qquad G_{f^{(2)}}(x) = \begin{cases} \frac{a}{1-p}\left(x - \rho + p(x - \rho - c)\right), & x < c' \\ 0, & x = c' \\ \frac{a}{1-p}\left(x + \rho + p(x + \rho - c)\right), & x > c' \end{cases}$$

Here, given $c = \left(1 + \frac{p}{4}\right)\rho$, we can verify that

$$0 < c - \rho \leq c' \leq c \leq c + \rho,$$

where $0$, $c'$, and $c$ are the global minimum (or maximum) of $f$, $f^{(2)}$, and $f^{(1)}$, respectively.

Recall that $a \leq \frac{\beta}{5}$. Since $\eta \leq \frac{3}{10\beta}$, we have $\eta a \leq \frac{3}{50}$, which will be useful for the rest of the proof. Now, we analyze $x_{t+1}$, the next iterate of $m$-SAM. For $c - \rho \leq x_t \leq c + \rho$, the next iterate $x_{t+1}$ of $m$-SAM using $G_{f^{(1)}}(x_t)$ is

$$x_{t+1} = \begin{cases} x_t + \eta a(x_t + \rho - c), & x_t < c \\ x_t, & x_t = c \\ x_t + \eta a(x_t - \rho - c), & x_t > c. \end{cases}$$

We will show that $x_{t+1}$ must stay within the interval $[c - \rho, c + \rho]$. For the case $x_t = c$, the inclusion is trivial. To analyze the rest, we divide into two cases.

**Case A-1:** $c - \rho \leq x_t < c$. Since $\eta a \leq \frac{3}{50}$, we have

$$c - \rho \leq x_t \leq x_{t+1} = x_t + \eta a(x_t + \rho - c) \leq c + \eta a(c + \rho - c)$$
$$= c + \eta a\rho \leq c + \frac{3\rho}{50}.$$

**Case A-2:** $c < x_t \leq c + \rho$. Since $\eta a \leq \frac{3}{50}$, we have

$$c + \rho \geq x_t \geq x_{t+1} = x_t + \eta a(x_t - \rho - c) \geq c + \eta a(c - \rho - c)$$
$$= c - \eta a\rho \geq c - \frac{3\rho}{50}.$$

Cases **A-1** and **A-2** show that $x_{t+1}$ also remains in $[c - \rho, c + \rho]$, when we update $m$-SAM using $G_{f^{(1)}}(x_t)$.

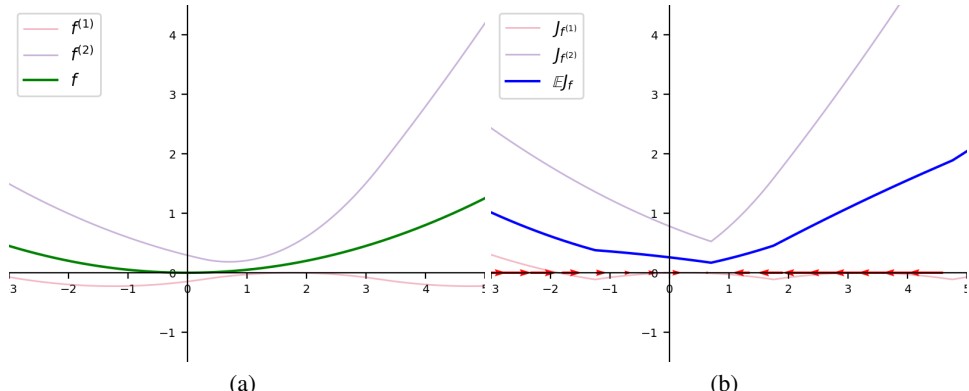

Figure 4: The original and virtual loss plot for the example function in Theorem 4.2. The graph drawn in purple and red are the original/virtual loss of component functions. The graph drawn in green indicates $f$, and the graph drawn in blue indicates $\mathbb{E}J_f$. (a) $f$ and its component functions $f^{(1)}$, $f^{(2)}$. (b)$\mathbb{E}J_f$ and its component functions $J_{f^{(1)}}$, $J_{f^{(2)}}$.

We next examine the case involving $G_{f^{(2)}}$. The next iterate $x_{t+1}$ of $m$-SAM using $G_{f^{(2)}}(x_t)$ is

$$
x_{t+1} = \begin{cases} x_t - \eta\frac{a}{1-p}\left(x_t - \rho + p(x_t - \rho - c)\right), & x_t < c' \\ x_t, & x_t = c' \\ x_t - \eta\frac{a}{1-p}\left(x_t + \rho + p(x_t + \rho - c)\right), & x_t > c'. \end{cases}
$$

Again, we divide it into two cases, since the $x_t = c'$ case is obvious.

**Case B-1:** $c - \rho \le x_t < c'$. Since $(1+p)x_t - (1+p)\rho - pc < 0$ for $x_t \in [c - \rho, c')$, we have

$$
c - \rho \le x_t \le x_{t+1} = x_t - \eta\frac{a}{1-p}\left((1+p)x_t - (1+p)\rho - pc\right)
$$

$$
= \left(1 - \eta\frac{a(1+p)}{1-p}\right)x_t + \eta\frac{a(1+p)}{1-p}\rho + \eta\frac{ap}{1-p}c.
$$

Since $\eta a \le \frac{3}{50}$, we have $1 - \eta\frac{a(1+p)}{1-p} = 1 - 5\eta a \ge 0$, so

$$
\left(1 - \eta\frac{a(1+p)}{1-p}\right)x_t + \eta\frac{a(1+p)}{1-p}\rho + \eta\frac{ap}{1-p}c
$$

$$
\le \left(1 - \eta\frac{a(1+p)}{1-p}\right)c' + \eta\frac{a(1+p)}{1-p}\rho + \eta\frac{ap}{1-p}c
$$

$$
= c' + \eta\frac{a(1+p)}{1-p}\rho = c' + 5\eta a\rho \le c' + \rho \le c + \rho.
$$

**Case B-2:** $c' < x_t \le c + \rho$. Since $(1+p)x_t + (1+p)\rho - pc > 0$ for $x_t \in (c', c + \rho]$, we have

$$
c + \rho \ge x_t \ge x_{t+1} = x_t - \eta\frac{a}{1-p}\left((1+p)x_t + (1+p)\rho - pc\right)
$$

$$
\ge c' - \eta\frac{a}{1-p}\left((1+p)c' + (1+p)\rho - pc\right)
$$

$$
= \frac{p}{1+p}c - \eta\frac{a(1+p)}{1-p}\rho = c - \frac{1}{1+p}c - 5\eta a\rho
$$

$$
= c - \frac{1}{1+p}\cdot\left(1 + \frac{p}{4}\right)\rho - 5\eta a\rho
$$

$$
\ge c - \frac{3}{5}\cdot\left(\frac{7}{6}\right)\rho - \frac{3}{10}\rho = c - \rho,
$$

where the last inequality used $\eta a \leq \frac{3}{50}$.

Cases **B-1** and **B-2** demonstrate that $x_{t+1}$ also remains within the interval $[c - \rho, c + \rho]$ when we update $m$-SAM using $G_{f^{(2)}}(x_t)$.

To gain a better intuitive understanding, we provide Figures 4(a), 4(b), demonstrating the original and virtual loss functions of $f^{(1)}$ and $f^{(2)}$. In Figure 4(b), we can examine that $m$-SAM updates generate an attraction basin in $J_{f^{(1)}}$, thereby making a region that $m$-SAM cannot escape.

The aforementioned case analyses indicate that when the initial point $x_0$ falls within the interval $[c - \rho, c + \rho]$, all subsequent iterations of $m$-SAM will also remain within this interval, regardless of the selected component function for updating. Given $c = \left(1 + \frac{p}{4}\right)\rho = \frac{7}{6}\rho$, we can conclude that all subsequent iterations of $m$-SAM are points located at a distance of at least $\frac{1}{6}\rho$ from the global optimum, for any $\beta > 0$, $\rho > 0$, and $\sigma > 0$.

Moreover, in case of $\sigma > \beta\rho$, it follows that $f(x) = \frac{a}{2}x^2 = \frac{\beta}{10}x^2$. Consequently, the suboptimality gap at any timestep is at least

$$f(x_t) - f^* = f(x_t) \geq f\left(\frac{\rho}{6}\right) = \Omega(\beta\rho^2),$$

thereby finishing the proof. $\qquad\square$

### C.4 Convergence Proof for Smooth and Convex Functions (Proof of Theorem 4.3)

In this section, we establish the convergence of stochastic SAM for smooth and convex functions to near-stationary points. For ease of understanding, we provide the theorem statement here.

**Theorem 4.3.** *Consider a $\beta$-smooth, convex function $f$, and assume Assumption 2.5. Under $n$-SAM, starting at $x_0$ with any perturbation size $\rho > 0$ and step size $\eta = \min\left\{\frac{\sqrt{\Delta}}{\sqrt{\beta[\sigma^2 - \beta^2\rho^2] + T}}, \frac{1}{2\beta}\right\}$ to minimize $f$, we have*

$$\frac{1}{T}\sum_{t=0}^{T-1}\mathbb{E}\|\nabla f(\boldsymbol{x}_t)\|^2 = \mathcal{O}\left(\frac{\beta\Delta}{T} + \frac{\sqrt{\beta[\sigma^2 - \beta^2\rho^2]_+ \Delta}}{\sqrt{T}}\right) + 4\beta^2\rho^2.$$

*Under $m$-SAM, additionally assuming $l(\cdot, \xi)$ is $\beta$-smooth for any $\xi$, the inequality continues to hold.*

*Proof.* We start from Lemma C.3 with $\mu = 0$. In order to do this, additionally assuming $\beta$-smoothness of component functions $l(\cdot, \xi)$ is necessary for $m$-SAM.

$$\mathbb{E}f(\boldsymbol{x}_{t+1}) \leq \mathbb{E}f(\boldsymbol{x}_t) - \frac{\eta}{2}\mathbb{E}\|\nabla f(\boldsymbol{x}_t)\|^2 + 2\beta^2\rho^2\eta + \beta\eta^2(\sigma^2 - \beta^2\rho^2),$$

which can be rewritten as

$$\mathbb{E}\|\nabla f(\boldsymbol{x}_t)\|^2 \leq \frac{2}{\eta}(\mathbb{E}f(\boldsymbol{x}_t) - \mathbb{E}f(\boldsymbol{x}_{t+1})) + 4\beta^2\rho^2 + 2\beta\eta(\sigma^2 - \beta^2\rho^2).$$

Adding up the inequality for $t = 0, \cdots, T - 1$, and dividing both sides by $T$, we get

$$\frac{1}{T}\sum_{t=0}^{T-1}\mathbb{E}\|\nabla f(\boldsymbol{x}_t)\|^2 \leq \frac{2}{\eta T}(\mathbb{E}f(\boldsymbol{x}_0) - \mathbb{E}f(\boldsymbol{x}_T)) + 4\beta^2\rho^2 + 2\beta\eta(\sigma^2 - \beta^2\rho^2)$$

$$\leq \frac{2}{\eta T}\Delta + 4\beta^2\rho^2 + 2\eta\beta(\sigma^2 - \beta^2\rho^2). \tag{11}$$

The convergence rate varies depending on two different cases determined by the value of $\sigma$.

**Case A: $\sigma \leq \beta\rho$.** In this case, it must hold that $\eta = \frac{1}{2\beta}$. By this choice of $\eta$, we have

$$\frac{1}{T}\sum_{t=0}^{T-1}\mathbb{E}\|\nabla f(\boldsymbol{x}_t)\|^2 \leq \frac{2\Delta}{\eta T} + 4\beta^2\rho^2$$

$$= \frac{4\beta\Delta}{T} + 4\beta^2\rho^2.$$

**Case B: $\sigma > \beta\rho$.** Setting $\eta = \min\left\{ \sqrt{\frac{\Delta}{\beta(\sigma^2 - \beta^2\rho^2)T}}, \frac{1}{2\beta} \right\}$, we consider two cases.

**Case B-1:** $\sqrt{\frac{\Delta}{\beta(\sigma^2 - \beta^2\rho^2)T}} \leq \frac{1}{2\beta}$. Putting $\eta = \sqrt{\frac{\Delta}{\beta(\sigma^2 - \beta^2\rho^2)T}}$ into (11), we get

$$\frac{1}{T}\sum_{t=0}^{T-1} \mathbb{E}\|\nabla f(\boldsymbol{x}_t)\|^2 \leq 4\sqrt{\frac{\beta(\sigma^2 - \beta^2\rho^2)\Delta}{T}} + 4\beta^2\rho^2.$$

**Case B-2:** $\sqrt{\frac{\Delta}{\beta(\sigma^2 - \beta^2\rho^2)T}} \geq \frac{1}{2\beta}$. Placing $\eta = \frac{1}{2\beta}$ into (11), we get

$$\frac{1}{T}\sum_{t=0}^{T-1} \mathbb{E}\|\nabla f(\boldsymbol{x}_t)\|^2 \leq \frac{4\beta\Delta}{T} + 4\beta^2\rho^2 + 2 \cdot \frac{1}{2\beta} \cdot \beta(\sigma^2 - \beta^2\rho^2)$$

$$\leq \frac{4\beta\Delta}{T} + 4\beta^2\rho^2 + 2 \cdot \sqrt{\frac{\Delta}{\beta(\sigma^2 - \beta^2\rho^2)T}} \cdot \beta(\sigma^2 - \beta^2\rho^2)$$

$$= \frac{4\beta\Delta}{T} + 2\sqrt{\frac{\beta(\sigma^2 - \beta^2\rho^2)\Delta}{T}} + 4\beta^2\rho^2.$$

Merging the two cases, we conclude

$$\frac{1}{T}\sum_{t=0}^{T-1} \mathbb{E}\|\nabla f(\boldsymbol{x}_t)\|^2 = \mathcal{O}\left( \frac{\beta\Delta}{T} + \frac{\sqrt{\beta(\sigma^2 - \beta^2\rho^2)\Delta}}{\sqrt{T}} \right) + 4\beta^2\rho^2,$$

thereby completing the proof. $\qquad\square$

### C.5 Non-Convergence of $m$-SAM for a Smooth and Convex Function (Proof of Theorem 4.4)

In this section, we provide an in-depth analysis of our counterexample, which demonstrates that stochastic $m$-SAM with constant perturbation size $\rho$ can provably converge to a point that is not a global minimum. Furthermore, the suboptimality gap in terms of the function value can be made arbitrarily large, hence indicating that proving convergence of $m$-SAM to global minima (modulo some additive factors $\mathcal{O}(\rho^2)$) is impossible. The theorem is restated for convenience.

**Theorem 4.4.** *For any $\rho > 0$, $\beta > 0$, $\sigma > 0$, and $\eta \leq \frac{1}{\beta}$, there exists a $\beta$-smooth and convex function $f$ satisfying the following. (1) The function $f$ satisfies Assumption 2.5. (2) The component functions $l(\cdot; \xi)$ of $f$ are $\beta$-smooth for any $\xi$. (3) If we run $m$-SAM on $f$ initialized inside a certain interval, then any arbitrary weighted average $\bar{\boldsymbol{x}}$ of the iterates $\boldsymbol{x}_0, \boldsymbol{x}_1, \ldots$ must satisfy $\mathbb{E}[f(\bar{\boldsymbol{x}}) - f^*] \geq C$, and the suboptimality gap $C$ can be made arbitrarily large and independent of the parameter $\rho$.*

*Proof.* Given $\rho > 0$, $\beta > 0$, $\sigma > 0$, we set an arbitrary constant $c > \frac{5\rho}{4}$, and a parameter $a > 0$ which will be chosen later. For $x \in \mathbb{R}$, consider a one-dimensional smooth convex function,

$$f(x) = \begin{cases} ax + \frac{\beta}{2}x^2, & x \leq 0 \\ ax, & x \geq 0, \end{cases}$$

and $l(x; \xi)$ can be given as

$$l(x; \xi) = \begin{cases} f^{(1)}(x), & \text{with probability } p \\ f^{(2)}(x), & \text{otherwise,} \end{cases}$$

where the functions $f^{(1)}$ and $f^{(2)}$ are given by the following definitions.

$$f^{(1)}(x) = \begin{cases} 2a\left(x - c + \frac{\rho}{8}\right) + \frac{\beta}{2}x^2, & x \leq 0 \\ 2a\left(x - c + \frac{\rho}{8}\right), & 0 \leq x \leq c - \frac{\rho}{4} \\ -\frac{4a}{\rho}(x - c)^2, & c - \frac{\rho}{4} \leq x \leq c + \frac{\rho}{4} \\ -2a\left(x - c - \frac{\rho}{8}\right), & c + \frac{\rho}{4} \leq x, \end{cases}$$

$$f^{(2)}(x) = \begin{cases} \frac{1}{1-p}\left(ax - 2pa\left(x - c + \frac{\rho}{8}\right)\right) + \frac{\beta}{2}x^2, & x \le 0 \\ \frac{1}{1-p}\left(ax - 2pa\left(x - c + \frac{\rho}{8}\right)\right), & 0 \le x \le c - \frac{\rho}{4} \\ \frac{1}{1-p}\left(ax + \frac{4pa}{\rho}(x - c)^2\right), & c - \frac{\rho}{4} \le x \le c + \frac{\rho}{4} \\ \frac{1}{1-p}\left(ax + 2pa\left(x - c - \frac{\rho}{8}\right)\right), & c + \frac{\rho}{4} \le x. \end{cases}$$

It can be verified that $f^{(1)}$ is $\left(\max\left\{\frac{8a}{\rho}, \beta\right\}\right)$-smooth, and $f^{(2)}$ is $\left(\max\left\{\frac{8a}{\rho} \cdot \frac{p}{1-p}, \beta\right\}\right)$-smooth. Moreover, $\|\nabla f(x) - \nabla f^{(1)}(x)\|^2 \le 9a^2$ holds, as well as $\|\nabla f(x) - \nabla f^{(2)}(x)\|^2 \le 9a^2 \cdot \frac{p^2}{(1-p)^2}$. Consequently, $\mathbb{E}\|\nabla f(x) - g(x)\|^2 \le 9a^2 \cdot \frac{p}{1-p}$.

By selecting $p > \frac{1}{2}$, and setting $a = \min\left\{\frac{\beta\rho(1-p)}{8p}, \frac{\sigma\sqrt{1-p}}{3\sqrt{p}}\right\}$, we can check that $\mathbb{E}l(x;\xi) = f(x)$, and the component functions are $\beta$-smooth. Additionally, $f$ satisfies Assumption 2.5.

In the following analysis, we examine the virtual gradient maps $G_{f^{(1)}}$ and $G_{f^{(2)}}$ in the specified region of interest: $c - \rho \le x \le c + \rho$. For this specific interval, we are going to show that if we start $m$-SAM inside this interval, then all the subsequent iterates of $m$-SAM must stay inside the same interval $[c - \rho, c + \rho]$.

In this region, $y = x + \rho\frac{\nabla f^{(1)}(x)}{\|\nabla f^{(1)}(x)\|}$ and $G_{f^{(1)}}$ are as follows.

$$y = \begin{cases} x + \rho, & c - \rho \le x < c \\ x, & x = c \\ x - \rho, & c < x \le c + \rho. \end{cases}$$

$$G_{f^{(1)}}(x) = \begin{cases} -\frac{8a}{\rho}(x + \rho - c), & c - \rho \le x \le c - \frac{3\rho}{4} \\ -2a, & c - \frac{3\rho}{4} \le x < c \\ 0, & x = c \\ 2a, & c < x \le c + \frac{3\rho}{4} \\ -\frac{8a}{\rho}(x - \rho - c), & c + \frac{3\rho}{4} \le x \le c + \rho. \end{cases}$$

Additionally defining $c' \triangleq c - \frac{\rho}{8p}$, we can compute $y = x + \rho\frac{\nabla f^{(2)}(x)}{\|\nabla f^{(2)}(x)\|}$ and $G_{f^{(2)}}$ as follows.

$$y = \begin{cases} x - \rho, & c - \rho \le x < c' \\ x, & x = c' \\ x + \rho, & c' < x \le c + \rho. \end{cases}$$

$$G_{f^{(2)}}(x) = \begin{cases} \frac{a - 2ap}{1-p}, & c - \rho \le x < c' \\ 0, & x = c' \\ \frac{a + 2ap}{1-p}, & c' < x \le c + \rho. \end{cases}$$

Now recall that $\eta \le \frac{1}{\beta}$. Given that $a \le \frac{\beta\rho(1-p)}{8p}$, we can derive $\eta \le \frac{1}{\beta} \le \frac{\rho}{a} \cdot \frac{1-p}{8p}$. Furthermore, the next iterate $x_{t+1}$ of $m$-SAM using $G_{f^{(1)}}(x_t)$ for $c - \rho \le x_t \le c + \rho$ is

$$x_{t+1} = x_t - \eta G_{f^{(1)}}(x_t) = \begin{cases} x_t + \eta \cdot \frac{8a}{\rho}(x + \rho - c), & c - \rho \le x_t \le c - \frac{3\rho}{4} \\ x_t + 2\eta a, & c - \frac{3\rho}{4} \le x_t < c \\ x_t, & x_t = c \\ x_t - 2\eta a, & c < x_t \le c + \frac{3\rho}{4} \\ x_t + \eta \cdot \frac{8a}{\rho}(x - \rho - c), & c + \frac{3\rho}{4} \le x_t \le c + \rho. \end{cases}$$

We will now show that $x_{t+1}$ must remain in the interval $[c - \rho, c + \rho]$. For the case $x_t = c$, it is obvious. Dividing the rest into three cases,

**Case A-1:** $c - \rho \le x_t \le c - \frac{3\rho}{4}$. Since $0 \le \eta \cdot \frac{8a}{\rho}(x_t + \rho - c) \le 2\eta a \le \rho \cdot \frac{1-p}{4p} \le \frac{\rho}{4}$, we have

$$c - \rho \le x_t \le x_{t+1} \le x_t + \frac{\rho}{4} \le c + \rho,$$

which proves that $x_{t+1}$ also remains in $[c - \rho, c + \rho]$.

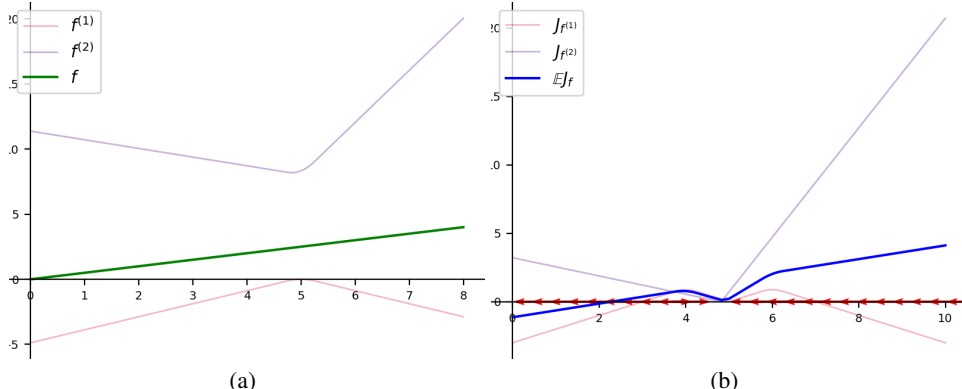

Figure 5: The original and virtual loss plot for the example function in Theorem 4.4. The graph drawn in purple and red are the original/virtual loss of component functions. The graph drawn in green indicates $f$, and the graph drawn in blue indicates $\mathbb{E}J_f$. (a) $f$ and its component functions $f^{(1)}$, $f^{(2)}$. (b) $\mathbb{E}J_f$ and its component functions $J_{f^{(1)}}$, $J_{f^{(2)}}$.

**Case A-2:** $c - \frac{3\rho}{4} \leq x_t < c$, $c < x_t \leq c + \frac{3\rho}{4}$. Since $0 \leq 2\eta a \leq \rho \cdot \frac{1-p}{4p} \leq \frac{\rho}{4}$, we have

$$c - \rho \leq x_t - \frac{\rho}{4} \leq x_{t+1} \leq x_t + \frac{\rho}{4} \leq c + \rho,$$

thereby proving that $x_{t+1}$ also remains in $[c - \rho, c + \rho]$.

**Case A-3:** $c + \frac{3\rho}{4} \leq x_t \leq c + \rho$. Since $0 \geq \eta \cdot \frac{8a}{\rho}(x_t - \rho - c) \geq -2\eta a \geq -\rho \cdot \frac{1-p}{4p} \geq -\frac{\rho}{4}$, we have

$$c - \rho \leq x_t - \frac{\rho}{4} \leq x_{t+1} \leq x_t \leq c + \rho,$$

which shows that $x_{t+1}$ also remains in $[c - \rho, c + \rho]$.

We next consider the case of $G_{f^{(2)}}$. The next iterate $x_{t+1}$ of $m$-SAM using $G_{f^{(2)}}(x_t)$ for $c - \rho \leq x_t \leq c + \rho$ is given by

$$x_{t+1} = x_t - \eta G_{f^{(2)}}(x_t) = \begin{cases} x_t - \eta a \cdot \frac{1-2p}{1-p}, & c - \rho \leq x_t < c' \\ x_t, & x_t = c' \\ x_t - \eta a \cdot \frac{1+2p}{1-p}, & c' < x_t \leq c + \rho. \end{cases}$$

Again, the $x_t = c'$ case is trivial. Considering the remaining two cases,

**Case B-1:** $c - \rho \leq x_t < c'$. Since $0 \geq \eta a \cdot \frac{1-2p}{1-p} \geq \rho \cdot \frac{1-2p}{8p} \geq -(c + \rho - c')$, we get

$$c - \rho \leq x_t \leq x_{t+1} \leq x_t + (c + \rho - c') \leq c + \rho,$$

which indicates that $x_{t+1}$ also remains in $[c - \rho, c + \rho]$.

**Case B-2:** $c' < x_t \leq c + \rho$. Since $0 \leq \eta a \cdot \frac{1+2p}{1-p} \leq \rho \cdot \frac{1+2p}{8p} \leq c' - (c - \rho)$, we have

$$c - \rho \leq x_t - c' + (c - \rho) \leq x_{t+1} \leq x_t \leq c + \rho,$$

which shows that $x_{t+1}$ also remains in $[c - \rho, c + \rho]$.

The case analyses above indicate that if $x_0$ is initialized in $[c - \rho, c + \rho]$, the subsequent iterates of $m$-SAM also remain in $[c - \rho, c + \rho]$, regardless of the component function chosen by $m$-SAM for update. Figures 5(a) and 5(b) demonstrate the original loss function and virtual loss function of component functions $f^{(1)}$ and $f^{(2)}$. Due to the concavity of $f^{(1)}$, a basin of attraction is formed in $J_{f^{(1)}}$, thereby creating a region where the iterations of $m$-SAM cannot escape.

Therefore, the suboptimality gap at any timestep is at least

$$f(x_t) - f^* = f(x_t) - f\left(-\frac{a}{\beta}\right) \geq f(c - \rho) - f\left(-\frac{a}{\beta}\right) = a(c - \rho) + \frac{a^2}{2\beta}.$$

Therefore, regardless of iterate averaging scheme, the suboptimality gap in terms of function value will stay above $a(c - \rho) + \frac{a^2}{2\beta}$.

Moreover, this suboptimality gap can be made arbitrarily large if we choose larger values of $c$; notice that $c > \frac{5\rho}{4}$ is the only requirement on $c$. Consequently, it is impossible to guarantee convergence to global minima, even up to an additive factor, for $m$-SAM. This finishes the proof. □

### C.6 Convergence Proof for Smooth and Nonconvex Function under $n$-SAM (Proof of Theorem 4.5)

In this section, we establish the convergence result of stochastic $n$-SAM for smooth and nonconvex functions. For convenience, we restate the theorem.

**Theorem 4.5.** *Consider a $\beta$-smooth function $f$ satisfying $f^* = \inf_x f(x) > -\infty$, and assume Assumption 2.5. Under $n$-SAM, starting at $x_0$ with any perturbation size $\rho > 0$ and step size $\eta = \min\left\{\frac{1}{2\beta}, \frac{\sqrt{\Delta}}{\sqrt{\beta\sigma^2 T}}\right\}$ to minimize $f$, we have*

$$\frac{1}{T}\sum_{t=0}^{T-1}\mathbb{E}\|\nabla f(x_t)\|^2 \leq \mathcal{O}\left(\frac{\beta\Delta}{T} + \frac{\sqrt{\beta\sigma^2\Delta}}{\sqrt{T}}\right) + \beta^2\rho^2.$$

*Proof.* Starting from the definition of $\beta$-smoothness, we have

$$\begin{aligned}
\mathbb{E}f(x_{t+1}) &\leq \mathbb{E}f(x_t) - \eta\mathbb{E}\langle\nabla f(x_t), \tilde{g}(y_t)\rangle + \frac{\beta\eta^2}{2}\mathbb{E}\|\tilde{g}(y_t)\|^2 \\
&\leq \mathbb{E}f(x_t) - \eta\mathbb{E}\langle\nabla f(x_t), \nabla f(y_t)\rangle + \beta\eta^2\left(\mathbb{E}\|\nabla f(y_t)\|^2 + \mathbb{E}\|\tilde{g}(y_t) - \nabla f(y_t)\|^2\right) \\
&\leq \mathbb{E}f(x_t) - \eta\mathbb{E}\langle\nabla f(x_t), \nabla f(y_t)\rangle + \beta\eta^2(\mathbb{E}\|\nabla f(y_t)\|^2 + \sigma^2) \\
&= \mathbb{E}f(x_t) - \frac{\eta}{2}\mathbb{E}\|\nabla f(x_t)\|^2 - \frac{\eta}{2}\mathbb{E}\|\nabla f(y_t)\|^2 + \frac{\eta}{2}\mathbb{E}\|\nabla f(x_t) - \nabla f(y_t)\|^2 \\
&\quad + \beta\eta^2(\mathbb{E}\|\nabla f(y_t)\|^2 + \sigma^2) \\
&\leq \mathbb{E}f(x_t) - \frac{\eta}{2}\mathbb{E}\|\nabla f(x_t)\|^2 + \frac{\beta^2\eta}{2}\mathbb{E}\|x_t - y_t\|^2 + \beta\sigma^2\eta^2 \\
&= \mathbb{E}f(x_t) - \frac{\eta}{2}\mathbb{E}\|\nabla f(x_t)\|^2 + \frac{\beta^2\rho^2\eta}{2} + \beta\sigma^2\eta^2.
\end{aligned}$$

Rearranging the inequality, we get

$$\mathbb{E}\|\nabla f(x_t)\|^2 \leq \frac{2}{\eta}(\mathbb{E}f(x_t) - \mathbb{E}f(x_{t+1})) + \beta^2\rho^2 + 2\beta\sigma^2\eta.$$

Adding up the inequality for $t = 0, \cdots, T-1$, and dividing both sides by $T$, we get

$$\frac{1}{T}\sum_{t=0}^{T-1}\mathbb{E}\|\nabla f(x_t)\|^2 \leq \frac{2}{\eta T}\left(\mathbb{E}f(x_0) - \mathbb{E}f(x_T)\right) + \beta^2\rho^2 + 2\beta\sigma^2\eta$$

$$\leq \frac{2\Delta}{\eta T} + \beta^2\rho^2 + 2\beta\sigma^2\eta. \tag{12}$$

Substituting $\eta = \min\left\{\frac{1}{2\beta}, \frac{\sqrt{\Delta}}{\sqrt{T\beta\sigma^2}}\right\}$ to (12), we get two cases.

**Case A:** $\frac{1}{2\beta} \leq \frac{\sqrt{\Delta}}{\sqrt{T\beta\sigma^2}}$. $\eta = \frac{1}{2\beta}$, so we have

$$\frac{1}{T}\sum_{t=0}^{T-1}\mathbb{E}\|\nabla f(x_t)\|^2 \leq \frac{2\Delta}{\eta T} + \beta^2\rho^2 + 2\beta\sigma^2\eta$$

$$\leq \frac{4\beta\Delta}{T} + 2\beta\sigma^2 \cdot \frac{\sqrt{\Delta}}{\sqrt{T\beta\sigma^2}} + \beta^2\rho^2$$

$$= \frac{4\beta\Delta}{T} + \frac{2\sqrt{\beta\sigma^2\Delta}}{\sqrt{T}} + \beta^2\rho^2.$$

**Case B:** $\frac{\sqrt{\Delta}}{\sqrt{T\beta\sigma^2}} \leq \frac{1}{2\beta}$. $\eta = \frac{\sqrt{\Delta}}{\sqrt{T\beta\sigma^2}}$, and it leads to

$$\frac{1}{T}\sum_{t=0}^{T-1} \mathbb{E}\|\nabla f(\boldsymbol{x}_t)\|^2 \leq \frac{2\Delta}{\eta T} + \beta^2\rho^2 + 2\beta\sigma^2\eta$$

$$\leq \frac{2\sqrt{\Delta}}{T} \cdot \sqrt{T\beta\sigma^2} + \frac{2\sqrt{\beta\sigma^2\Delta}}{\sqrt{T}} + \beta^2\rho^2$$

$$= \frac{4\sqrt{\beta\sigma^2\Delta}}{\sqrt{T}} + \beta^2\rho^2.$$

Merging two cases, we can conclude that

$$\frac{1}{T}\sum_{t=0}^{T-1} \mathbb{E}\|\nabla f(\boldsymbol{x}_t)\|^2 \leq \mathcal{O}\left(\frac{\beta\Delta}{T} + \frac{\sqrt{\beta\sigma^2\Delta}}{\sqrt{T}}\right) + \beta^2\rho^2,$$

thereby completing the proof. $\qquad\square$

## C.7 Convergence Proof for Smooth Lipschitz Nonconvex Function under $m$-SAM (Proof of Theorem 4.6 and Corollary 4.7)

In this section, the convergence proof for stochastic $m$-SAM for smooth, Lipschitz, and nonconvex functions is presented. The notation $\hat{\boldsymbol{y}}_t = \boldsymbol{x}_t + \rho\frac{\nabla f(\boldsymbol{x}_t)}{\|\nabla f(\boldsymbol{x}_t)\|}$ is used here once again. The theorem is restated for convenience.

**Theorem 4.6.** *Consider a $\beta$-smooth, $L$-Lipschitz continuous function $f$ satisfying $f^* = \inf_x f(\boldsymbol{x}) > -\infty$, and assume Assumption 2.5. Additionally assume $l(\cdot, \xi)$ is $\beta$-smooth for any $\xi$. Under $m$-SAM, starting at $\boldsymbol{x}_0$ with any perturbation size $\rho > 0$ and step size $\eta = \frac{\sqrt{\Delta}}{\sqrt{\beta(\sigma^2+L^2)T}}$ to minimize $f$, we have*

$$\frac{1}{T}\sum_{t=0}^{T-1} \mathbb{E}\left[(\|\nabla f(\boldsymbol{x}_t)\| - \beta\rho)^2\right] \leq \mathcal{O}\left(\frac{\sqrt{\beta\Delta(\sigma^2+L^2)}}{\sqrt{T}}\right) + 5\beta^2\rho^2.$$

*Proof.* The proof technique resembles Mi et al. [27]. Starting from the definition of $\beta$-smoothness, we have

$$\mathbb{E}f(\boldsymbol{x}_{t+1}) \leq \mathbb{E}f(\boldsymbol{x}_t) - \eta\mathbb{E}\langle\nabla f(\boldsymbol{x}_t), \tilde{g}(\boldsymbol{y}_t)\rangle + \frac{\beta\eta^2}{2}\mathbb{E}\|\tilde{g}(\boldsymbol{y}_t)\|^2$$

$$\leq \mathbb{E}f(\boldsymbol{x}_t) - \eta\mathbb{E}\|\nabla f(\boldsymbol{x}_t)\|^2 - \eta\mathbb{E}\langle\nabla f(\boldsymbol{x}_t), \tilde{g}(\boldsymbol{y}_t) - \nabla f(\boldsymbol{x}_t)\rangle + \frac{\beta\eta^2}{2}\mathbb{E}\|\tilde{g}(\boldsymbol{y}_t)\|^2$$

$$= \mathbb{E}f(\boldsymbol{x}_t) - \eta\mathbb{E}\|\nabla f(\boldsymbol{x}_t)\|^2 - \eta\mathbb{E}\langle\nabla f(\boldsymbol{x}_t), \tilde{g}(\boldsymbol{y}_t) - \tilde{g}(\hat{\boldsymbol{y}}_t)\rangle - \eta\mathbb{E}\langle\nabla f(\boldsymbol{x}_t), \nabla f(\hat{\boldsymbol{y}}_t) - \nabla f(\boldsymbol{x}_t)\rangle$$

$$+ \frac{\beta\eta^2}{2}\mathbb{E}\|\tilde{g}(\boldsymbol{y}_t)\|^2$$

$$\leq \mathbb{E}f(\boldsymbol{x}_t) - \eta\mathbb{E}\|\nabla f(\boldsymbol{x}_t)\|^2 + \frac{\eta}{2}\mathbb{E}\|\nabla f(\boldsymbol{x}_t)\|^2 + \frac{\eta}{2}\mathbb{E}\|\tilde{g}(\boldsymbol{y}_t) - \tilde{g}(\hat{\boldsymbol{y}}_t)\|^2$$

$$- \eta\mathbb{E}\left\langle\frac{\|\nabla f(\boldsymbol{x}_t)\|}{\rho}\cdot(\hat{\boldsymbol{y}}_t - \boldsymbol{x}_t), \nabla f(\hat{\boldsymbol{y}}_t) - \nabla f(\boldsymbol{x}_t)\right\rangle + \frac{\beta\eta^2}{2}\mathbb{E}\|\tilde{g}(\boldsymbol{y}_t)\|^2$$

$$\leq \mathbb{E}f(\boldsymbol{x}_t) - \frac{\eta}{2}\mathbb{E}\|\nabla f(\boldsymbol{x}_t)\|^2 + \frac{\beta^2\eta}{2}\mathbb{E}\|\boldsymbol{y}_t - \hat{\boldsymbol{y}}_t\|^2 + \eta\mathbb{E}\left[\frac{\beta\|\nabla f(\boldsymbol{x}_t)\|}{\rho}\cdot\|\hat{\boldsymbol{y}}_t - \boldsymbol{x}_t\|^2\right] + \frac{\beta\eta^2}{2}\mathbb{E}\|\tilde{g}(\boldsymbol{y}_t)\|^2$$

$$\leq \mathbb{E}f(\boldsymbol{x}_t) - \frac{\eta}{2}\mathbb{E}\|\nabla f(\boldsymbol{x}_t)\|^2 + \frac{\beta^2\rho^2\eta}{2}\mathbb{E}\left\|\frac{\tilde{g}(\boldsymbol{x}_t)}{\|\tilde{g}(\boldsymbol{x}_t)\|} - \frac{\nabla f(\boldsymbol{x}_t)}{\|\nabla f(\boldsymbol{x}_t)\|}\right\|^2 + \beta\rho\eta\mathbb{E}\|\nabla f(\boldsymbol{x}_t)\|$$

$$+ \frac{\beta \eta^2}{2} \mathbb{E} \| \tilde{g}(\boldsymbol{y}_t) \|^2$$

$$\leq \mathbb{E} f(\boldsymbol{x}_t) - \frac{\eta}{2} \mathbb{E} \| \nabla f(\boldsymbol{x}_t) \|^2 + 2\beta^2 \rho^2 \eta + \beta \rho \eta \mathbb{E} \| \nabla f(\boldsymbol{x}_t) \| + \frac{\beta \eta^2}{2} \mathbb{E} \| \tilde{g}(\boldsymbol{y}_t) \|^2$$

$$\leq \mathbb{E} f(\boldsymbol{x}_t) - \frac{\eta}{2} \mathbb{E} (\| \nabla f(\boldsymbol{x}_t) \| - \beta \rho)^2 + \frac{5}{2} \beta^2 \rho^2 \eta + \beta \eta^2 (\mathbb{E} \| \nabla f(\boldsymbol{y}_t) \|^2 + \mathbb{E} \| \tilde{g}(\boldsymbol{y}_t) - \nabla f(\boldsymbol{y}_t) \|^2)$$

$$\leq \mathbb{E} f(\boldsymbol{x}_t) - \frac{\eta}{2} \mathbb{E} (\| \nabla f(\boldsymbol{x}_t) \| - \beta \rho)^2 + \frac{5}{2} \beta^2 \rho^2 \eta + \beta \eta^2 (\sigma^2 + L^2).$$

Rearranging the inequality, we get

$$\mathbb{E} \left[ (\| \nabla f(\boldsymbol{x}_t) \| - \beta \rho)^2 \right] \leq \frac{2}{\eta} (\mathbb{E} f(\boldsymbol{x}_t) - \mathbb{E} f(\boldsymbol{x}_{t+1})) + 5\beta^2 \rho^2 + 2\beta \eta (\sigma^2 + L^2).$$

Adding up the inequality for $t = 0, \cdots, T - 1$, and dividing both sides by $T$, we get

$$\frac{1}{T} \sum_{t=0}^{T-1} \mathbb{E} \left[ (\| \nabla f(\boldsymbol{x}_t) \| - \beta \rho)^2 \right] \leq \frac{2}{\eta T} (\mathbb{E} f(\boldsymbol{x}_0) - \mathbb{E} f(\boldsymbol{x}_T)) + 5\beta^2 \rho^2 + 2\beta \eta (\sigma^2 + L^2)$$

$$\leq \frac{2\Delta}{\eta T} + 5\beta^2 \rho^2 + 2\beta \eta (\sigma^2 + L^2). \tag{13}$$

Substituting $\eta = \frac{\sqrt{\Delta}}{\sqrt{\beta(\sigma^2+L^2)T}}$ to (13) yields

$$\frac{1}{T} \sum_{t=0}^{T-1} \mathbb{E} \left[ (\| \nabla f(\boldsymbol{x}_t) \| - \beta \rho)^2 \right] \leq \frac{4\sqrt{\beta \Delta (\sigma^2 + L^2)}}{\sqrt{T}} + 5\beta^2 \rho^2,$$

thereby completing the proof. $\qquad \square$

From the result of Theorem 4.6, Corollary 4.7 can be derived. The corollary is restated for the ease of reference.

**Corollary 4.7.** *Under the setting of Theorem 4.6, we get*

$$\min_{t \in \{0,...,T\}} \{ \mathbb{E} \| \nabla f(\boldsymbol{x}_t) \| \} \leq \mathcal{O} \left( \frac{(\beta \Delta (\sigma^2 + L^2))^{1/4}}{T^{1/4}} \right) + \left( 1 + \sqrt{5} \right) \beta \rho.$$

*Proof.* Starting from the result of Theorem 4.6, we have

$$\min_{t \in \{0,...,T\}} \left\{ (\mathbb{E} \| \nabla f(\boldsymbol{x}_t) \| - \beta \rho)^2 \right\} \leq \min_{t \in \{0,...,T\}} \mathbb{E} \left[ (\| \nabla f(\boldsymbol{x}_t) \| - \beta \rho)^2 \right]$$

$$\leq \frac{1}{T} \sum_{t=0}^{T-1} \mathbb{E} \left[ (\| \nabla f(\boldsymbol{x}_t) \| - \beta \rho)^2 \right]$$

$$\leq \mathcal{O} \left( \frac{\sqrt{\beta \Delta (\sigma^2 + L^2)}}{\sqrt{T}} \right) + 5\beta^2 \rho^2.$$

Taking the square root on both sides,

$$\min_{t \in \{0,...,T\}} \left\{ \left| \mathbb{E} \| \nabla f(\boldsymbol{x}_t) \| - \beta \rho \right| \right\} \leq \mathcal{O} \left( \frac{(\beta \Delta (\sigma^2 + L^2))^{1/4}}{T^{1/4}} \right) + \sqrt{5} \beta \rho.$$

Rearranging the inequality, we obtain

$$\min_{t \in \{0,...,T\}} \{ \mathbb{E} \| \nabla f(\boldsymbol{x}_t) \| \} \leq \mathcal{O} \left( \frac{(\beta \Delta (\sigma^2 + L^2))^{1/4}}{T^{1/4}} \right) + \left( 1 + \sqrt{5} \right) \beta \rho,$$

thereby completing the proof. $\qquad \square$

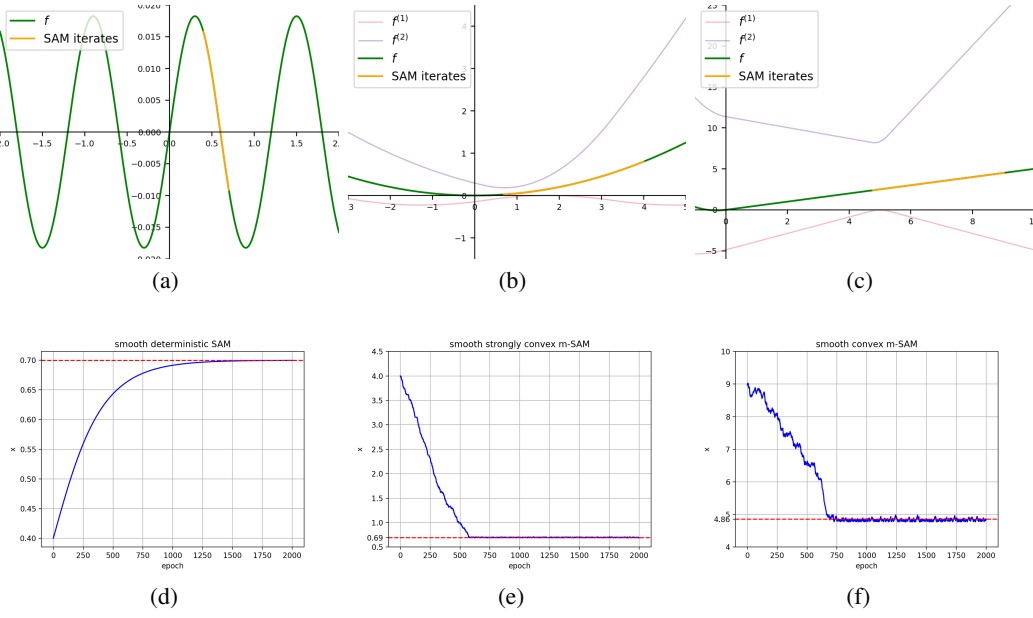

Figure 6: The results of the SAM simulations on the example functions. The yellow line indicates the trajectory of SAM iterates. (a) and (d) display deterministic SAM iterates (with initialization $x_0 = 0.4$) and the plot of x-coordinate values over epochs, for a smooth nonconvex function as shown in Figure 2(a) under settings in Theorem 3.5. (b) and (e) show $m$-SAM iterates (with initialization $x_0 = 4$) and the plot of x-coordinate values over epochs, for the smooth strongly convex function as shown in Figure 2(b) under settings in Theorem 4.2. (c) and (f) demonstrate $m$-SAM iterates (with initialization $x_0 = 9$) and the plot of x-coordinate values over epochs, for the smooth convex function as shown in Figure 2(d) under settings in Theorem 4.4. All plots empirically verify that practical SAM cannot converge all the way to optima. Instead, the iterates get trapped in certain regions.

## D  Simulation Results of SAM on Example Functions

Figure 6 shows the results of SAM simulations on the example functions considered in Sections 3 and 4. It demonstrates that SAM indeed does not converge in our worst-case example constructions.

