# OpenReview forum: "Practical Sharpness-Aware Minimization Cannot Converge All the Way to Optima"
_NeurIPS.cc/2023/Conference — NeurIPS 2023 spotlight_

### Official Review · Reviewer_y8UR · 2023-06-23

**Soundness:** 2 fair
**Presentation:** 3 good
**Contribution:** 2 fair
**Rating:** 6
**Confidence:** 4

**Summary:**

This paper analyzes the convergence properties of Sharpness-Aware Minimization (SAM) with **constant** perturbation size $\rho$ and gradient normalization applied to the updates; this has not been done in prior works. Both deterministic and stochastic settings are considered. The authors show the following:

1. For the strongly convex + smooth case in the deterministic setting, SAM converges to the global minimum. A matching lower bound for the rate is also presented although I'm not sure if it's correct; please see Weakness #2. For the general convex + smooth case in the deterministic setting, the authors are only able to show convergence to a stationary point (in terms of gradient norm).

2. More importantly, the authors show that in all other settings considered in the paper, SAM cannot converge to the global/local minimum. Specifically, this includes the smooth non-convex and non-smooth Lipschitz convex cases in the deterministic setting and smooth + (strongly) convex and smooth non-convex cases in the stochastic setting. Some lower bounds are also presented to complement the upper bounds.

In summary, compared to the case of decaying perturbation size $\rho$ analyzed in prior theoretical works on SAM, this work shows that using a constant $\rho$ can inhibit convergence.

**Strengths:**

**1.** Unlike prior theoretical works on SAM, this paper considers the use of constant perturbation size and gradient normalization in the SAM update which is more aligned with practice.

**2.** The theory is comprehensive in the sense that the common function classes are covered and there is analysis for both the deterministic and stochastic settings.

**3.** In some cases, matching lower bounds are also presented to support the upper bounds. Some of the constructions and techniques for the lower bounds seemed novel to me (for e.g., Theorem 3.6) but I'm not an expert on lower bounds.

**4.** The paper is also written more or less clearly.

**Weaknesses:**

**1.** In the abstract, the authors write: "*Perhaps surprisingly, in many scenarios, we find out that SAM has limited capability to converge to global minima or stationary points*". But I don't find the non-convergence of SAM **with constant $\rho$** (and constant step-size $\eta$) very surprising; in fact, the asymptotic convergence in Theorems 3.1 and 3.3 surprises me. I say this because of the following reason. Suppose deterministic SAM converges to some point $\tilde{x}$. Then, we must have $\nabla f\Big(\tilde{x} + \rho \frac{\nabla f(\tilde{x})}{||\nabla f(\tilde{x})||}\Big) = 0$. This means that we must have $\tilde{x} + \rho \frac{\nabla f(\tilde{x})}{||\nabla f(\tilde{x})||} = x^{\ast}$, where $x^{\ast}$ is some stationary point of $f$. So, $|| \tilde{x} -  x^{\ast}|| = \rho$, i.e., if SAM with constant $\rho$ converges to some point, then that point must be $\rho$ distance away from a stationary point of $f$. Notice that if $\rho$ is a decreasing function of $t$, this (apparent) issue of non-convergence won't arise.

**2.** There seems to be an issue in Theorem 3.2 which states that $\frac{\beta}{\mu} \geq 2$. But in the proof of Theorem 3.2, one-dimensional quadratics are used to obtain lower bounds; for these one-dimensional quadratics the **tightest possible** smoothness constant = **tightest possible** strong convexity parameter. So actually, $\beta = \mu$ in all 3 cases. Now in Case 1, $x_0 = 2\rho$. But then, we get $x_t \geq \frac{x_0}{2} = \rho$. Unfortunately, this yields a vacuous lower bound of 0. Can the authors comment on this? It seems that if the current proof strategy is to be used, we would need at least 2-d quadratics where $\beta > \mu$.

**3.** In Theorem 3.4, the dependence of the convergence bound w.r.t. $T$ seems loose and I don’t think the Lipschitz assumption is required. In the second equation of the proof, there is a $-\frac{\eta}{2}||\nabla f(y_t)||^2$ term and a $\frac{\eta^2 \beta}{2}||\nabla f(y_t)||^2$ term; these two terms can be combined together and as long as $\eta \beta < 1$, the coefficient of $||\nabla f(y_t)||^2$ is negative. So I think we can get $O(1/T)$ convergence to a $O(\beta^2 \rho^2)$ stationary point with a fixed step-size independent of $T$. Also, if we follow this approach, we don’t even Lipschitzness.

**4.** It is unfortunate that the lower bounds for the stochastic case needed to be corrected in the supplementary material.

**5.** (Minor weakness but a limitation nevertheless) SAM was proposed to improve generalization. But this paper has no results illustrating how SAM improves generalization such as by converging to points that are *approximately* flat minima (or because it inhibits convergence preventing over-fitting).

A missed reference (just for the author's information): https://openreview.net/pdf?id=IcDTYTI0Nx shows the convergence of SAM with non-constant perturbation sizes.

**Questions:**

Please address Weaknesses 1, 2 and 3. My current rating is mainly due to Weakness 1; if the authors can point out some fallacy in my understanding or convince me that the *non-convergence* of SAM isn't all that obvious, I can increase my score.

**Limitations:**

Some minor limitations have been discussed such as the loose dependence on $\beta$ and $\mu$ in Theorem 3.1. But important limitations such as Weakness #5 have not been discussed. No foreseeable negative societal impacts.

---

> ### Author Rebuttal · Authors · 2023-08-10
>
> We appreciate the reviewer for dedicating their valuable time to evaluate our paper. Below are our responses to the questions raised.
>
> > **Weakness 1. Non-convergence seems obvious.**
>
> As the reviewer correctly points out, if deterministic SAM finds a fixed point $\tilde{x}$, it must hold that $\nabla f(\tilde{x}+\rho \frac{\nabla f(\tilde{x})}{\lVert\nabla f(\tilde{x})\rVert})=0$. However, this does not necessarily indicate that $\tilde{x} + \rho\frac{\nabla f(\tilde{x})}{\lVert\nabla f(\tilde{x})\rVert} = x^*\neq \tilde{x}$, where $x^*$ represents a stationary point or a global minimum. This is because we treat $\frac{\nabla f(x)}{\lVert\nabla f(x)\rVert}$ as $0$ whenever $\nabla f(x)=0$ (see lines 123-125), which means that the converged fixed point $\tilde{x}$ can also be $\tilde{x} = x^*$ itself, not necessarily a point satisfying $\lVert\tilde{x}-x^*\rVert=\rho$. In light of this point, we'll clarify why the equation $\lVert\tilde{x}-x^*\rVert=\rho$ is not always true, by presenting examples in convex and nonconvex functions.
>
> - **Convex functions.**
>
> By utilizing Lemma B.2, we get $\langle\nabla f(\tilde{x}),\nabla f(\tilde{y})-\nabla f(\tilde{x})\rangle\geq 0$, where $\tilde{y}=\tilde{x}+\rho\frac{\nabla f(\tilde{x})}{\lVert \nabla f(\tilde{x})\rVert}$. Rearranging, we get $\langle \nabla f(\tilde{x}),\nabla f(\tilde{y})\rangle\geq\lVert \nabla f(\tilde{x})\rVert^2$. Consequently, $\lVert \nabla f(\tilde{y})\rVert ^2 \geq \lVert \nabla f(\tilde{x})\rVert^2$, which in turn indicates that $\nabla f(\tilde{y})=0$ if and only if $\tilde{x}=x^*$, where $x^*$ is the global minimum. This implies that if SAM converges to $\tilde{x}$, then $\tilde{x}$ **must be** $x^*$.
>
> - **Nonconvex functions.**
>
> Here, we present scenarios wherein SAM tends to converge in proximity to a stationary point $x^*$, rather than a point located $\rho$-away from $x^*$.
>
> **Scenario 1**: Consider a "local maxima function," represented by $f(x)=-\frac{1}{2}x^2$. We can quickly verify that a possible virtual loss (recall Definition 2.7) can be defined as
> $$
> J_f(x)=\\begin{cases}
>     -\\frac{1}{2}(x+\rho)^2,  &x\leq0 \\\\
>     -\frac{1}{2}(x-\rho)^2,  &x>0.
> \\end{cases}
> $$
> Therefore, when $x_t\in[-\rho,\rho]$, SAM updates tend to converge towards $x^*=0$. Precisely, we can check that for $\eta<1$, if $x_t\in[-\eta\rho,\eta\rho]$, the subsequent iterates remain within $[-\eta\rho,\eta\rho]$, implying $|x_t-x^*|\leq\eta\rho$. This means that for sufficiently small $\eta$, the SAM iterates stay in proximity to $x^*$ at a distance **much closer than** $\rho$.
>
> **Scenario 2**: Consider a "saddle function," denoted as $f(x,y)=x^2-y^2$. Analogous to scenario 1, the SAM iterates near the saddle point tend to converge to that saddle point. For the detailed illustration, refer to Figure 5 in [1].
>
> From these scenarios, we can check that for nonconvex functions, local maxima and saddle points can serve as attractors within certain regions. Consequently, there are many scenarios that SAM iterates converge to stationary points $x^*$, and we can conclude that the convergence of SAM to points $\lVert \tilde{x}-x^*\rVert=\rho$ **does not always happen** even in nonconvex settings.
>
> > **Weakness 2. There seems to be an issue in Theorem 3.2.**
>
> As outlined in our discussion on the "Remarks on the validity of lower bound" (lines 634-656), our objective is to establish a lower bound for Equation (7). Specifically, we aim to demonstrate the existence of lower bound for each $A \in \mathcal A$, wherein we select $f \in \mathcal F$ accordingly. In the context of Theorem 3.2, the function class $\mathcal{F}$ represents the collection of all $\beta$-smooth $\mu$-strongly convex functions, where the constants satisfy $\frac{\beta}{\mu} \geq 2$. In order to prove a lower bound, we get to choose ``worst-case'' functions from $\mathcal F$, and this function class naturally includes 1-D quadratic functions: $f(x)=\frac{a}{2}x^2+bx+c$, for any $a$, $b$, and $c$ satisfying $\mu\leq a\leq\beta$. It is easy to verify that the functions we have employed in Cases 1, 2, and 3 of Section B.3 belong to this particular category $\mathcal F$. Therefore, using 1-D quadratic examples does not harm the correctness of our derivation of lower bounds.
>
> Additionally, as for the issue raised for Case 1, even in the case of $\beta=\mu$, by scaling $x_0$ (in line 612) up properly and starting at $x_0 = 4\rho$, we can get $x_t \geq \frac{x_0}{2} = 2\rho$ and immediately resolve the issue.
>
> > **Weakness 3. Theorem 3.4 seems loose.**
>
> Setting $\eta = \frac{1}{\beta}$ and following the same steps as in Theorem 3.4 while also combining the two terms as suggested by the reviewer, we can verify that $\frac{1}{T}\sum_{t=0}^{T-1}\lVert\nabla f(x_t)\rVert^2\leq \frac{2\beta\Delta}{T}+\beta^2\rho^2$. This result shows that we can achieve a convergence rate of $O(1/T)$ to a $O(\beta^2\rho^2)$ stationary point without the Lipschitzness assumption. The same approach applies to stochastic $n$, $m$-SAM as well (with $\eta=\mathrm{min}\\\{\frac{1}{2\beta},\frac{\sqrt{\Delta}}{\sqrt{T\beta\sigma^2}}\\\}$ or $\eta=\frac{1}{\beta}$). We greatly appreciate the reviewer's valuable input which improved our findings, and we will make sure to incorporate these improvements in the revised manuscript.
>
> > **Weakness 5. No results regarding generalization**
>
> We would appreciate it if you could refer to our response of Question 1 in the ``global response".
>
> We hope that our response successfully addresses the issues raised by the reviewer, and we would greatly appreciate it if the reviewer could consider reassessing our paper. Thank you again for the insightful review.
>
> ---
> [1] Kim, H., Park, J., Choi, Y., \& Lee, J. (2023). Stability Analysis of Sharpness-Aware Minimization. arXiv preprint arXiv:2301.06308.

---

> > ### Comment · Reviewer_y8UR · 2023-08-13
> > **Reply to Rebuttal**
> >
> > I thank the authors for the detailed rebuttal!
> >
> > **Weakness 1**: Thank you for the explanation. Gradient normalization becoming 0 at stationary points is somewhat disconcerting to me. I see that under this seemingly unrealistic definition of normalization, non-convergence is not all that obvious. But I feel this definition is misaligned with reality and hard to reconcile. And perhaps that is why it would be meaningful to incorporate the small corrective $\epsilon$ term in the denominator of normalization. I will raise my score to 6 but I strongly encourage the authors to discuss this limitation and include all the discussion here in the next version of the paper.
> >
> > **Weakness 2**: I agree that the bound can be corrected **w.r.t. $\rho$** by properly up-scaling $x_0$ but the factors of $\beta$ and $\mu$ in the bound don't seem right to me because the functions are 1D quadratics with $\beta = \mu$; I suggest removing the $\frac{\beta}{\mu} \geq 2$ condition in the theorem statement as well as the $\beta$ and $\mu$ terms in the bound.

---

> > > ### Author Response · Authors · 2023-08-15
> > > **Additional response to Reviewer y8UR**
> > >
> > > We appreciate the reviewer for providing thorough insights, and we are glad to know that the reviewer decided to offer a positive evaluation; we find it highly encouraging.
> > >
> > > As for Weakness 1, we agree to the reviewer that setting $\frac{\nabla f(x)}{\lVert \nabla f(x) \rVert} = 0$ when $\nabla f(x) = 0$ could be deemed a non-standard way of normalization. However, we believe it is reasonable to adopt this approach when it comes to analyzing practical version of SAM. This is because the practical implementations of SAM in fact use $\frac{\nabla f(x)}{\lVert \nabla f(x) \rVert + \epsilon}$, where a small $\epsilon > 0$ is introduced for numerical stability (e.g., the official code of SAM in Tensorflow). We will thoroughly address this discrepancy in our next revision to ensure it is not overlooked.
> > >
> > > Regarding the issue raised in Weakness 2, we would like to re-emphasize that the $\beta$ and $\mu$ serve as constants defining the **function class** $\mathcal F$, rather than **individual functions** chosen from the class. It is important to note that a 1-d quadratic function $f(x) = \frac{1}{2} a x^2 + b x + c$ is by definition $\beta$-smooth and $\mu$-strongly convex for **any** $\beta$ and $\mu$ satisfying $\mu \leq a \leq \beta$. For instance, the function $f(x) = \frac{1}{2}\cdot\frac{3}{4}x^2$ is $\frac{1}{2}$-strongly convex and $1$-smooth, although these constants are obviously not the tightest. In the proof of Theorem 3.2, our goal is to find the worst-case examples out of the function class $\mathcal F$. This class contains all $\beta$-smooth and $\mu$-strongly convex functions, where $\frac{\beta}{\mu} \geq 2$. Since the **worst-case** functions chosen in our proof are 1-d quadratic functions, with coefficients of quadratic term being $a = \mu$, $\frac{\beta}{2}$, and $\beta$ for Cases 1, 2, and 3, respectively, it becomes evident that $\mu \leq a \leq \beta$ holds for all three cases and they are indeed valid members of $\mathcal F$. Having said that, we acknowledge that it is quite confusing, and it is ideal not to have the assumption $\frac{\beta}{\mu} \geq 2$. We will carefully reconsider it and make necessary adjustments to improve our result and minimize any possible confusion.
> > >
> > > Again, we appreciate your valuable feedback. We will make sure to incorporate all the discussions into the next version of the paper.

---

### Official Review · Reviewer_T74n · 2023-06-25

**Soundness:** 3 good
**Presentation:** 2 fair
**Contribution:** 3 good
**Rating:** 7
**Confidence:** 3

**Summary:**

SAM is a very practical algorithm for improving generalization in deep learning, however, even the convergence properties of SAM are not well understood. The paper studies the convergence/non-convergence of SAM under various standard setups in optimization, including smooth/nonsmooth, convex/strongly convex/nonconvex, deterministic/stochastic optimization problems. The results of this paper provide a detailed and complete description of the convergence of SAM for these problems.

**Strengths:**

1. Compared with existing works, the paper analyzes SAM with fixed $\rho$ and normalization steps, which is the algorithm that people use in practice. Under these practical configurations, this paper provides different characteristics of SAM.
2. The upper bounds give sufficient conditions for the convergence of SAM and the convergence rate of the algorithm under these conditions. Additionally, this paper also provides lower bounds to show the tightness of their upper bounds in the dependency of certain important problem parameters such as $\rho$ and $T$.
3. I found the lower bounds very interesting and they provide many insights about SAM. In contrast to gradient descent (GD), the result in this paper shows that deterministic SAM (full-batch SAM) fails to converge to a  stationary point even for smooth nonconvex objectives. In contrast to stochastic gradient descent (SGD), stochastic SAM (m-SAM) even fails to converge to a global minimum for smooth strongly convex objectives. These results justify the limited capability of SAM in optimization and suggest the differences between SAM and other common optimizers such as SGD/Adam/RMSProp.


**Weaknesses:**

See the questions part.

**Questions:**

1. It seems that we only have the non-convergence result of stochastic SAM for smooth and strongly convex functions for m-SAM. Can we prove similar lower bounds for n-SAM? If not, I hope the authors can explain the underlying intuition. By the way, it is proper to say "stochastic SAM for every function classes we consider, fail to converge properly" in Line 359?
2. I can not fully understand the purpose of Sec. 3.4 as well as Thm. 3.6. It seems that for merely convex objectives, gradient descent also fails to converge to a global minima? See the second inequality in Thm. 2.1.7 in Nesterov's celebrated book Lectures
on Convex Optimization", Second Edition.

---

> ### Author Rebuttal · Authors · 2023-08-10
>
> We appreciate the reviewer for their detailed evaluation of our paper. We are truly grateful for your interest in our lower bound results. The responses addressing the questions raised are outlined below.
>
> > **Question 1. Can we prove similar lower bounds for $n$-SAM?**
>
> Indeed, we have conducted empirical investigations using toy scenarios under $n$-SAM, and observed some scenarios that manifest the same non-convergence phenomenon observed in $m$-SAM. However, the current proof technique employed for establishing the lower bound of $m$-SAM does not readily extend to $n$-SAM.
>
> To elaborate on the underlying intuition, in $m$-SAM, the same component function is employed during both the ascending and descending steps (for formal definitions of $n$-SAM and $m$-SAM, please refer to Section 2.2). This symmetry enables us to leverage the concept of virtual loss associated with each component function (component function within the context of stochastic optimization), illustrating how iterations can be trapped within a specific interval. In contrast, $n$-SAM can employ distinct component functions for its ascending and descending steps, rendering the virtual loss of each component function less applicable. Consequently, the existing proof technique cannot be seamlessly transferred to $n$-SAM, prompting the need for an alternative approach. Developing an alternative technique for theoretically establishing the lower bound of $n$-SAM could be a viable direction for future research.
>
> One important thing to note, though, is that $n$-SAM is a theoretical variant of SAM, which have never been used in practice. In practical scenarios, it is predominantly $m$-SAM that has gained widespread adoption, as one can check from open-source implementations of SAM on GitHub (we wish we could provide pointers here, but links are not allowed!). Since our focus is centered on the practical settings of SAM, we believe that the absence of lower bound results under $n$-SAM does not significantly harm our paper.
>
> Also, thank you for pointing out the subtle issue in line 359. The sentence should indeed be revised to "stochastic $m$-SAM for every function classes we consider, fail to converge properly."
>
> > **Question 2. The purpose of Sec 3.4.**
>
> In the preceding Sections 3.1-3.3, we study the convergence properties of deterministic SAM for smooth functions under different convexity assumptions. This leads us to wonder whether SAM can similarly provide convergence guarantees for nonsmooth functions. Following this curiosity, we investigate whether SAM can find global minima of nonsmooth convex functions; unfortunately, the answer is on the negative. In Section 3.4, we present a lower bound of $\Omega(\rho)$ on the distance of SAM iterates to the global minimum. This result highlights that, unlike smooth convex functions, SAM is unable to achieve global convergence in nonsmooth convex functions.
>
> Indeed, as noted by the reviewer, Theorem 2.1.7 in [1] shows a lower bound on the distance to the global minimum for a smooth convex function. However, the cited theorem has a fundamental difference in comparison to our Theorem 3.6. Importantly, the lower bound in [1] for gradient-based methods exclusively applies to early iterates: specifically, it applies to the range of iterates where $1 \leq t \leq \frac{1}{2}(n-1)$, with $t$ is the iteration index and $n$ represents the dimension of the problem. On the contrary, our established lower bound for SAM holds true for all iterations, even when $t$ goes to infinity.
>
> ---
> [1] Nesterov, Y. (2018). Lectures on convex optimization (Vol. 137, p. 576). Berlin: Springer.

---

> > ### Comment · Reviewer_T74n · 2023-08-11
> > **Repsonse to the Rebuttal**
> >
> > The author's rebuttal addressed my concern well. I think it is a very interesting paper, and I decide to raise the score to 7.

---

> > > ### Author Response · Authors · 2023-08-12
> > > **Additional response to Reviewer T74n**
> > >
> > > We express our gratitude to the reviewer for the valuable discussion and positive evaluation. We are glad to hear that our discussion has cleared your concerns. Please let us know if there are any further inquiries.
> > >
> > > Best regards,
> > >
> > > Authors

---

### Official Review · Reviewer_NH1R · 2023-07-02

**Soundness:** 3 good
**Presentation:** 3 good
**Contribution:** 2 fair
**Rating:** 5
**Confidence:** 3

**Summary:**

This paper studies the convergence properties of sharpness-aware minimization in a specific setting with the use of gradient normalization and arbitrary constant perturbation. The paper established convergence rates for both deterministic and stochastic SAM with various assumptions on the convexity of the objective function. The authors argue the term $\mathcal{O}(\rho^2)$ appearing multiple times in their bounds is in fact unavoidable by establishing corresponding lower bounds.

**Strengths:**

- The paper is well-written and easy to follow.
- The analysis covers different levels of convexity.
- The author proves lower bounds to justify their proposed convergence rate.

**Weaknesses:**

- The proof of the upper bounds seems to have followed that of [1].
- Several cases are left out, including the lower bound for the smooth and convex case of deterministic SAM, and the lower bound for the smooth and strongly-convex case of stochastic SAM with small variance.
- Generalization is not considered.

**Questions:**

- General questions:
  -  The paper considers SAM with constant stepsizes and gradient normalization and this setting seems to be a common choice in practice. However, I do think more justification for this setting is needed, especially when [1] derived faster convergence rates for SAM without gradient normalization.
  - It seems no methodological conclusions for the implementation of SAM are provided in the paper. Based on the results in this paper, (how) should we decrease the perturbation $\rho$ in the algorithm so that the additive term $\mathcal{O}(\rho^2)$ would be resolved and thus the convergence be faster?

- Small issues:
  - There is only one lower bound result in Table 1, which looks weird to me.
  - In line 830, does "Assumption 4.2" refer to Theorem 4.2?

[1] M. Andriushchenko and N. Flammarion. Towards understanding sharpness-aware minimization. In International Conference on Machine Learning, pages 639–668. PMLR, 2022.

---

> ### Author Rebuttal · Authors · 2023-08-10
>
> We sincerely thank the reviewer for devoting their time to evaluate our paper. Here are our responses to the weaknesses raised.
>
> > **Weakness 1. Comparison with [1].**
>
> Our proofs might share similarities with those in [1]. However, our analyses overcome unique difficulties and hence different from [1]; allow us to elaborate below.
>
> First, our paper takes the normalization term into consideration, which is not addressed in [1]. This introduces certain unfavorable terms throughout the proof procedure. One obvious example can be found in line 772. If we ignore normalization, we would have gotten the term $\lVert\rho g(x_t) -\rho \nabla f(x_t)\rVert^2$, which can be easily bounded using Assumption 2.5. However, our paper incorporates the normalization term, leading to a completely different expression: $\lVert\rho\frac{g(x_t)}{\lVert g(x_t)\rVert}-\rho\frac{\nabla f(x_t)}{\lVert\nabla f(x_t)\rVert}\rVert^2$, thereby rendering the analysis more challenging.
> As a result, employing different techniques (rather than approaches found in [1]) becomes crucial to address these unfavorable terms.
>
> Due to these distinctive terms that specifically emerge in our study, many techniques from [1], such as the Lyapunov function method in Theorem 11, cannot be seamlessly transferred to our framework. In cases where these methods are not applicable, we construct an alternative strategy, such as a finely-tailored multi-case analysis based on the values of $\rho$, $\eta$, or $\lVert\nabla f(x_t)\rVert$ (One of the examples that use an alternative method is showcased in Section B.2).
>
> Second, the upper bounds presented in our work hold true for all $\rho>0$. In contrast, every theorem in [1] relies on critical assumptions, like bounding $\rho$, decaying $\rho$, or requiring $\rho$ to be sufficiently small. Establishing the results under less restrictive assumptions on $\rho$ naturally requires employing distinct proof techniques.
>
> > **Weakness 2. Two cases of lower bound are left out.**
>
> We acknowledge that the lower bound for smooth and convex functions in deterministic SAM has not been addressed in this work and is reserved for future investigations.
>
> Regarding the case of lower bound for smooth and strongly-convex functions in $m$-SAM with small variance ($\sigma\leq\beta\rho$), a modification to the proof of Theorem 4.2 allows us to at least establish a lower bound for the distance to the global minimum, denoted as $\lVert x_t-x^* \rVert$.
>
> In line 827, we can modify the selection of $a = \frac{\beta}{5}$ and instead choose $a = \frac{\sigma}{5\rho} \leq\frac{\beta}{5}$. The validity of the proof remains intact with this revised choice of $a$, even when $\sigma \leq \beta\rho$. Consequently, when the initial point $x_0$ falls within the interval $[c-\rho,c+\rho]$ (where $c=\frac{7}{6}\rho$), all subsequent iterations of $m$-SAM remain within this interval. This observation indicates that even under the condition $\sigma\leq\beta\rho$, an analogous example demonstrates $\lVert x_t-x^* \rVert=\Omega(\rho)$ for all iterates.
>
> Of course, this lower bound leads to a similar lower bound on the function value: $f(x_t)-f^* = \Omega(\sigma \rho)$. Nonetheless, we omitted this lower bound, as it does not tightly match the factor $O(\rho^2)$ in the upper bound. However, it is worth highlighting that at least, even in the regime of small $\sigma$, $m$-SAM does not achieve convergence all the way to the global minimum.
>
> > **Weakness 3. Generalization not considered.**
>
> We would appreciate it if you could refer to our response of Question 1 in the "global response".
>
> > **Question 1. Justification for this setting.**
>
> SAM is commonly employed in real-world applications, while USAM (SAM without gradient normalization) serves as a theoretical variant designed for theoretical analysis and isn't used practically. As our paper focuses on practical settings, it is natural that we study SAM instead of USAM.
>
> Furthermore, USAM shows drastically different behaviors compared to SAM. This distinction is illustrated in Figure 1, where we can observe that USAM has a trajectory closer to GD than SAM. Also, [2] investigates the differences between SAM and USAM, such as the stabilization property (SAM can converge with a wider range of $\eta$ than USAM) or the "drift-along-minima" phenomena (SAM tends to drift along a manifold of minima, while USAM gets stuck at a minimum). So, it is reasonable to regard USAM as an entirely different optimizer when compared to SAM, and hence natural to expect different convergence properties.
>
> Hence, it is crucial to notice that the faster convergence rates in [1] for USAM do not imply similar convergence properties for SAM. Our paper specifically addresses the relatively unexplored convergence behavior of SAM under practical setups.
>
> > **Question 2. (How) Should we decrease $\rho$?**
>
> We would appreciate it if you could refer to our response of Question 2 in the "global response".
>
> > **Issues.**
>
> For Table 1, some lower bound results have been excluded as they do not provide rates with respect to time $T$. Instead, they demonstrate that the additive factor of $O(\rho^2)$ in upper bounds are unavoidable, thereby having different characteristics compared to the lower bound result (Theorem 3.2) presented in Table 1.
>
> Thank you for pointing out the typo in line 830. The correct reference should be "Assumption 2.5", which indeed holds within the interval $[c-\rho,c+\rho]$.
>
> We sincerely hope that the reviewer finds our response convincing, and we would appreciate it if the reviewer could consider re-evaluating our score. Thank you again for the helpful feedback.
>
> ---
> [1] Andriushchenko, M., \& Flammarion, N. (2022, June). Towards understanding sharpness-aware minimization. In International Conference on Machine Learning (pp. 639-668). PMLR.
>
> [2] Dai, Y., Ahn, K., \& Sra, S. (2023). The Crucial Role of Normalization in Sharpness-Aware Minimization. arXiv preprint arXiv:2305.15287.

---

> > ### Comment · Reviewer_NH1R · 2023-08-15
> >
> > Thanks for the response. I am raising my score to (5), subject to the authors adding the above comments in their paper.

---

> > > ### Author Response · Authors · 2023-08-15
> > > **Additional response to Reviewer NH1R**
> > >
> > > We sincerely appreciate the reviewer for their thorough discussions and for offering a positive evaluation. We will ensure that all the above discussions are incorporated in the next version of the paper.
> > >
> > > Best regards,
> > >
> > > Authors

---

### Official Review · Reviewer_AJAE · 2023-07-10

**Soundness:** 3 good
**Presentation:** 2 fair
**Contribution:** 3 good
**Rating:** 7
**Confidence:** 3

**Summary:**

This paper examines convergence properties of sharpness-aware minimization, which is an empirically-popular algorithm for training deep neural networks, yet whose theoretical properties are poorly understood. This paper makes several contributions to the analysis of sharpness-aware minimization, which are novel in this growing literature.

For smooth and convex functions, this paper proves that the best iterate of SAM converges to the global minimizer. This is expected as, for such functions, there exists a global minimizer; thus, even after perturbations, the convergent point should still be the global minimizer.

For smooth and nonconvex functions, this paper proves that there is an additional bias term in the convergent point compared with a local minimizer. Furthermore, a lower bound example is constructed, showing that this bias term is necessary.

**Strengths:**

- The paper is nicely written and provides enough background knowledge for readers to understand the topic. For each theorem, the authors accompany their results with a sketch of the high-level proof arguments.

- The lower-bound examples are especially interesting in terms of complementing the convergence rates.

**Weaknesses:**

- The results in the main text are a bit dense. It would be easier to read if the paper is formatted in a more friendly manner.

- There are no experiments/simulations to complement the theoretical results. I consider this a limitation since SAM is primarily motivated as an empirically-successful algorithm, so a connection between theoretical results to their practical implications would be important for this audience.

**Questions:**

- I wonder if the authors could discuss the practical implications of their findings. For example, how does the extra bias term affect the empirical performance of SAM; Does it inject some kind of regularizer that is actually helpful for the empirical performance, or is it possible to design a better algorithm that eliminates this bias term altogether?

- There is a concurrent paper that studies a very related question:

Dai, Y., Ahn, K., & Sra, S. (2023). The Crucial Role of Normalization in Sharpness-Aware Minimization. arXiv preprint arXiv:2305.15287.

It would be helpful if the authors could discuss the relation to this paper.

**Limitations:**

The authors discuss the limitations of their results along with presentations in the main text. The potential societal impact of their work should be minimal, given the technical nature of the paper.

---

> ### Author Rebuttal · Authors · 2023-08-10
>
> We are grateful to the reviewer for dedicating their time in evaluating our paper. Your interest in our lower bound examples is deeply appreciated. Below, we provide our responses addressing the raised questions.
>
> > **Weakness 1. The results in the main text are a bit dense.**
>
> Thank you for providing insights on enhancing readability. We are making an effort to address this matter to improve the quality of our paper.
>
> > **Weakness 2. There are no simulations to complement the theoretical results.**
>
> We have conducted supplementary simulations for the toy functions utilized in the non-convergence theorems. These simulations corroborate the validity of our theoretical proofs through empirical evidence. The results of these simulations are presented in the PDF file attached in the "global response". Moreover, the simulation result for the non-convergence in nonsmooth convex functions can be found in Figure 4 of Section B.6.
>
> We would greatly appreciate it if the reviewer could take a look at these results. If these supplementary experiments are deemed insufficient, we are open to any valuable suggestions for further experimental verification.
>
> > **Question 1. Practical implications of our findings.**
>
> We would appreciate it if you could refer to our response of Question 2 in the "global response".
>
> > **Question 2. Relation to [1].**
>
> [1] delves into an examination of the contrasting characteristics between SAM and USAM (SAM without normalization). They particularly focus on the stabilization property of SAM and USAM, i.e., SAM can converge with a wider range of $\eta$ than USAM. Additionally, they study on the "drift-along-minima" phenomena observed in SAM, wherein SAM tends to continue drifting along a manifold of minima, while USAM easily gets stuck at a minimum. In contrast to the results in [1], our paper focuses more on the (non-)convergence properties of SAM pertaining to finding global minima or stationary points.
>
> Having said that, the two papers indeed have a couple of directly comparable results too. In the discussion of stabilization (Theorem 1) in [1], the authors show that for $\mu$-strongly convex $\beta$-smooth functions, the iterates of SAM converge to a local neighborhood around the global minimum $x^*$. This theorem can be adapted to establish a convergence guarantee of $f(x_T)-f^* = \tilde{O}(\frac{1}{T})$, achieved by selecting step size $\eta = \mathrm{min}\\\{\frac{1}{\mu T} \mathrm{max}\\\{1, \log(\frac{\mu^2 \Delta T}{\beta^3\rho^2})\\\}, \frac{1}{\beta}\\\}$ and employing a proof technique similar to ours. It is worth highlighting that Theorem 3.1 in our paper achieves a **faster** convergence rate for the same function class: $\tilde{O}(\frac{1}{T^2})$.
>
> ---
> [1] Dai, Y., Ahn, K., \& Sra, S. (2023). The Crucial Role of Normalization in Sharpness-Aware Minimization. arXiv preprint arXiv:2305.15287.

---

### Author Rebuttal · Authors · 2023-08-10

Dear reviewers,

We sincerely thank all the reviewers for their careful evaluation of our paper and their valuable questions and comments. Your reviews have been tremendously helpful in improving our manuscript. We are glad that many reviewers recognized the comprehensiveness of our upper and lower bounds as well as their relevance to practical settings.

Below, we would like to start by addressing some concerns that were raised by multiple reviewers. We will address the remaining questions in the individual responses below.

> **Question 1. SAM was proposed to improve generalization. But this paper has no results illustrating how SAM improves generalization. (Reviewers y8UR, NH1R)**

We definitely agree that the main reason for SAM's widespread adoption is its superior generalization performance. We believe that understanding this characteristic of SAM from a theoretical perspective is of great importance and interest.

Indeed, prior researches [1,2] proved SAM's tendency to approach flat minima, but under the crucial assumption that SAM should initially converge near the global minima manifold and/or with a sufficiently small $\rho$. Yet, the question whether practical SAM really converges in the first place still remains open. Our paper tackles this gap by introducing scenarios where SAM can (or cannot) converge, under practical settings.

Our paper focuses on the convergence properties of SAM, thus the investigation of SAM's tendency to find flat minima and generalization property is outside the scope of our paper. We wish to tackle the generalization aspect of SAM in the future.

> **Question 2. Regarding the practical implication of our findings, How does the extra bias term affect the empirical performance of SAM? Also, is it possible to design a better algorithm that eliminates the extra bias term? (Reviewers AJAE, NH1R)**

Although SAM has gained much attention due to the superior generalization ability of models trained by the algorithm, our findings reveal that the capability of SAM as an optimizer is rather limited, which is quite surprising.

One straightforward method to eliminate the bias term $O(\rho^2)$ is to use USAM (SAM without gradient normalization) with a sufficiently small (or decaying) $\rho$; as demonstrated in [3], convergence guarantees for USAM have been established under such conditions. Additionally, employing SAM with a sufficiently small constant $\rho$ (which appropriately decays with $T$) can also address the extra bias term.

However, we are in fact skeptical if these modifications can be seen as "improved" algorithms, as the modified algorithms will be closer to GD (as outlined in lines 63-89 of our paper), thereby **losing** the distinct features of SAM.

The presence of an extra unavoidable bias term in the convergence bounds could potentially relate to SAM's generalization property; however, further investigation is required. As of now, no definitive conclusions can be drawn, making it a promising avenue for future research.

> **Additional experiments**

As per Reviewer AJAE's suggestion, we conducted additional experiments which demonstrate that SAM indeed does not converge in our worst-case example constructions. The results can be found in the attached pdf file below.

---
[1] Bartlett, P. L., Long, P. M., & Bousquet, O. (2022). The dynamics of sharpness-aware minimization: Bouncing across ravines and drifting towards wide minima. arXiv preprint arXiv:2210.01513.

[2] Wen, K., Ma, T., & Li, Z. (2022). How does sharpness-aware minimization minimize sharpness?. arXiv preprint arXiv:2211.05729.

[3] Andriushchenko, M., & Flammarion, N. (2022, June). Towards understanding sharpness-aware minimization. In International Conference on Machine Learning (pp. 639-668). PMLR.

---

### Decision · Program_Chairs · 2023-09-21

**Decision:**

Accept (spotlight)

**Comment:**

The paper analyzes the convergence of the sharpness-aware minimization (SAM) algorithm. It provides both upper and lower bounds of convergence when the function is smooth and strongly convex, whereas in the case of smooth and nonconvex functions, it shows that convergence to a stationary point is possible only up to a certain neighborhood, and that furthermore this is unavoidable. Based on these contributions, along with the reviewers' having reached a consensus, I recommend acceptance of this paper.